# Quantitative optical nanophysiology of Ca$^{2+}$ signaling at inner hair cell active zones

Jakob Neef [1,2,3,4], Nicolai T. Urban [5,6], Tzu-Lun Ohn[1,2,3], Thomas Frank [1,9], Philippe Jean[1], Stefan W. Hell [5,6], Katrin I. Willig[2,5,6,7] & Tobias Moser [1,2,3,4,6,8]

Ca$^{2+}$ influx triggers the release of synaptic vesicles at the presynaptic active zone (AZ). A quantitative characterization of presynaptic Ca$^{2+}$ signaling is critical for understanding synaptic transmission. However, this has remained challenging to establish at the required resolution. Here, we employ confocal and stimulated emission depletion (STED) microscopy to quantify the number (20–330) and arrangement (mostly linear 70 nm × 100–600 nm clusters) of Ca$^{2+}$ channels at AZs of mouse cochlear inner hair cells (IHCs). Establishing STED Ca$^{2+}$ imaging, we analyze presynaptic Ca$^{2+}$ signals at the nanometer scale and find confined elongated Ca$^{2+}$ domains at normal IHC AZs, whereas Ca$^{2+}$ domains are spatially spread out at the AZs of bassoon-deficient IHCs. Performing 2D-STED fluorescence lifetime analysis, we arrive at estimates of the Ca$^{2+}$ concentrations at stimulated IHC AZs of on average 25 μM. We propose that IHCs form bassoon-dependent presynaptic Ca$^{2+}$-channel clusters of similar density but scalable length, thereby varying the number of Ca$^{2+}$ channels amongst individual AZs.

[1] Institute for Auditory Neuroscience and InnerEarLab, University Medical Center Göttingen, 37099 Göttingen, Germany. [2] Collaborative Research Center 889, University of Göttingen, 37075 Göttingen, Germany. [3] Bernstein Focus for Neurotechnology, University of Göttingen, 37075 Göttingen, Germany. [4] Synaptic Nanophysiology Group, Max Planck Institute for Biophysical Chemistry, 37077 Göttingen, Germany. [5] Department of Nanobiophotonics, Max Planck Institute for Biophysical Chemistry, 37077 Göttingen, Germany. [6] Center for Nanoscale Microscopy and Molecular Physiology of the Brain, University Medical Center Göttingen, 37099 Göttingen, Germany. [7] Optical Nanoscopy in Neuroscience, University Medical Center Göttingen, 37099 Göttingen, Germany. [8] Bernstein Center for Computational Neuroscience, University of Göttingen, 37075 Göttingen, Germany. [9] Present address: Friedrich Miescher Institute for Biomedical Research, 4058 Basel, Switzerland. Jakob Neef, Nicolai T. Urban and Tzu-Lun Ohn contributed equally to this work. Correspondence and requests for materials should be addressed to N.T.U. (email: nicolai.urban@mpfi.org) or to K.I.W. (email: kwillig@em.mpg.de) or to T.M. (email: tmoser@gwdg.de)

Voltage-gated $Ca^{2+}$ influx mediates stimulus–secretion coupling at the presynaptic active zone (AZ) and the ensuing transmission of information forms the basis for sensory, neural, and motor function[1]. For a few decades, synaptic neuroscience has aimed to establish a quantitative understanding of AZ $Ca^{2+}$ signaling[2–10]. Progress has been limited, however, by technical challenges such as the resolution limit of conventional light microscopy, which has precluded direct visualization of nanoscale $Ca^{2+}$ domains. Therefore, indirect approaches analyzing $Ca^{2+}$-dependent processes such as transmitter release and the gating of presynaptic $Ca^{2+}$-activated potassium channels[3,5,6] as well as mathematical modeling[3,4,8–11] have been employed to elucidate spatiotemporal $Ca^{2+}$-concentration profiles at the AZ. But understanding synaptic transmission requires information on the number, biophysical properties, and topography relative to vesicular release sites of the presynaptic $Ca^{2+}$ channels that determine the AZ $Ca^{2+}$ signaling. Indeed, localization and quantification of presynaptic $Ca^{2+}$ channels by electron microscopy[9,12,13] or cell-attached recordings[14,15] has recently fueled the progress.

Here, we studied the quantitative nanophysiology of presynaptic $Ca^{2+}$ influx using AZs of sensory inner hair cells (IHCs) as a model system. IHCs are well-suited for studies of $Ca^{2+}$ signaling at individual AZs because their AZs are relatively large and functionally separated due to µm-scale nearest-neighbor distance and strong $Ca^{2+}$ buffering[16–18]. Moreover, voltage-gated $Ca^{2+}$ influx in IHCs is almost exclusively mediated by a single type of $Ca^{2+}$ channel ($Ca_V1.3$)[19], which has also been characterized at the single-channel level[15,20]. Freeze-fracture electron microscopy[21] and two-dimensional stimulated emission depletion (2D-STED) microscopy of immunolabeled $Ca^{2+}$ channels[8,22] suggest a stripe-like clustering of $Ca^{2+}$ channels at the AZ, at least in fixed hair cells, similar to observations of the presynaptic protein bassoon at conventional synapses[23]. Based on analysis of the whole-cell $Ca^{2+}$ current and the average number of AZs, the mean number of $Ca^{2+}$ channels per AZ has been approximated to 80–100[15,21,24]. $Ca^{2+}$ nanodomains with $[Ca^{2+}]$ up to hundreds of µM were postulated from modeling, readout of $[Ca^{2+}]$ by large conductance $Ca^{2+}$-activated $K^+$ channels[3], or synaptic exocytosis[6,8,18,24]. Direct nanoscopic imaging of presynaptic $[Ca^{2+}]$, however, has yet to be established for AZs of IHCs or other presynaptic cells. Along with information on the number and topography of the $Ca^{2+}$ channels and vesicular release sites[8,9,11,12,14], such measurements will advance our understanding of stimulus–secretion coupling at the AZ. Moreover, a major heterogeneity of presynaptic $Ca^{2+}$ signaling was observed among the AZs of a given cell[12,14,17], which in IHCs likely contributes to the response diversity observed for postsynaptic neurons[25,26]. This calls for analysis beyond the average properties of AZs.

Here, we make use of the good experimental accessibility of individual AZs in IHCs to analyze the number, distribution, and activity of $Ca^{2+}$ channels at individual synapses using innovative methods. We combine confocal $Ca^{2+}$ imaging with selective inhibition of $Ca^{2+}$ influx at individual AZs and optical fluctuation analysis for estimating the number of $Ca^{2+}$ channels per AZ. Employing 3D-STED, we quantify the structure of AZ $Ca^{2+}$-channel clusters in IHCs. We establish 2D-STED $Ca^{2+}$ imaging and STED fluorescence lifetime measurements of AZ $[Ca^{2+}]$ and validate the method by mathematical modeling based on quantitative structural and functional characterization of presynaptic $Ca^{2+}$ channels.

## Results
**Nanoscale anatomy of synaptic $Ca^{2+}$-channel clusters in IHCs.** We analyzed the nanoscale layout of presynaptic $Ca^{2+}$ channels located at IHC AZs using super-resolution STED microscopy. AZs were immunolabeled against both the $Ca_V1.3\alpha_1$ subunit of the $Ca_V1.3$ $Ca^{2+}$ channel as well as against the presynaptic protein bassoon, which marks the presynaptic density at IHC AZs[22,27]. Using 2D-STED, we imaged AZs that were located at the basal membrane of an IHC, selecting those that seemed to lie parallel to the imaging plane (Fig. 1a–j). Most of the AZs (~80%) showed linear arrangements of both $Ca^{2+}$ channels and bassoon "clusters", whereby the $Ca^{2+}$-channel cluster co-aligned with the bassoon-labeled presynaptic density (Fig. 1a–e). We introduced a subjective classification of the apparent shapes of $Ca^{2+}$-channel and bassoon clusters that yielded comparable results when performed by different observers. About 60% of the channel clusters formed narrow lines of an average full-width at half-maximum (FWHM) of ~70 nm (Fig. 1a–c), whereas another ~20% were somewhat wider (FWHM ~100 nm, Fig. 1d, e), with some of them forming what appeared to be double stripes of $Ca_V1.3$- or bassoon-immunofluorescence sandwiching the respective other type of cluster (Fig. 1d, e). At about 15% of the synapses, larger and more complex arrangements of $Ca^{2+}$-channel and bassoon immunofluorescence were observed (Fig. 1f, g) and less than 5% of the synapses exhibited small, spot-like $Ca_V1.3$- and bassoon-immunofluorescence (Fig. 1h). The latter might represent spot-like clusters or stripe-like clusters that were aligned perpendicular to the imaging plane of the microscope.

For the most faithful 2D-STED measurements of the clusters, we used a secondary antibody tagged with the dye providing the highest resolution (Abberior STAR635P) for either $Ca_V1.3$ $Ca^{2+}$ channels or bassoon. We then approximated the apparent size of the clusters by fitting a 2D Gaussian function to the raw STED images (Supplementary Fig. 1). This revealed that the FWHMs of the long and short axes for linear clusters of $Ca_V1.3$ channels and bassoon were very similar (Fig. 1j), with the short axis averaging to $73 \pm 7$ nm for $Ca_V1.3$ channel clusters and $65 \pm 3$ nm for bassoon (mean ± SEM; medians: 67 and 61 nm, respectively). The apparent length (i.e., long axis) of the $Ca_V1.3$ channel clusters averaged $239 \pm 20$ nm (mean ± SEM; median: 206 nm) and varied between approximately 100 and 600 nm, likely reflecting differences in AZ size[16,26,28]. Similarly, bassoon clusters showed an average apparent length of $235 \pm 31$ nm (mean ± SEM; median: 227 nm). When relating the length of $Ca^{2+}$-channel and bassoon clusters with the lower resolution estimate of the length of their corresponding bassoon and $Ca^{2+}$-channel clusters (labeled with Abberior STAR580-conjugated secondary antibody), respectively, we found a high degree of correlation (Fig. 2g; Pearson's correlation coefficient of 0.71 for the entire data set including 3D-STED, see below; 0.58 for the 2D-data). Together with the identical length estimate, this indicates that $Ca^{2+}$-channel clusters and corresponding presynaptic densities (marked by bassoon) scale with each other.

One major drawback of a 2D-STED analysis is that the orientation of a given AZ relative to the imaging plane can have a marked effect on its apparent shape and size. If we assume that the imaged AZs were not oriented perfectly parallel to the imaging plane, but at an angle, then measuring the cluster sizes using 2D-STED (providing only lateral ($xy$) and no axial ($z$) super-resolution) would lead to an underestimation of the true cluster dimensions. We therefore turned to 3D-STED imaging[29] to complement our 2D-STED data. While sacrificing some lateral resolution for increased axial resolution, we could obtain a near-isotropic effective point spread function (PSF), i.e., probing region. We acquired stacks of images (Fig. 2a–d) and, to reduce bleaching, lowered the power of the STED laser, achieving a 3D-resolution of ~160 nm (<20% of the focal volume of a confocal PSF). This way, although we could no longer reliably differentiate the apparent shapes we had previously defined in

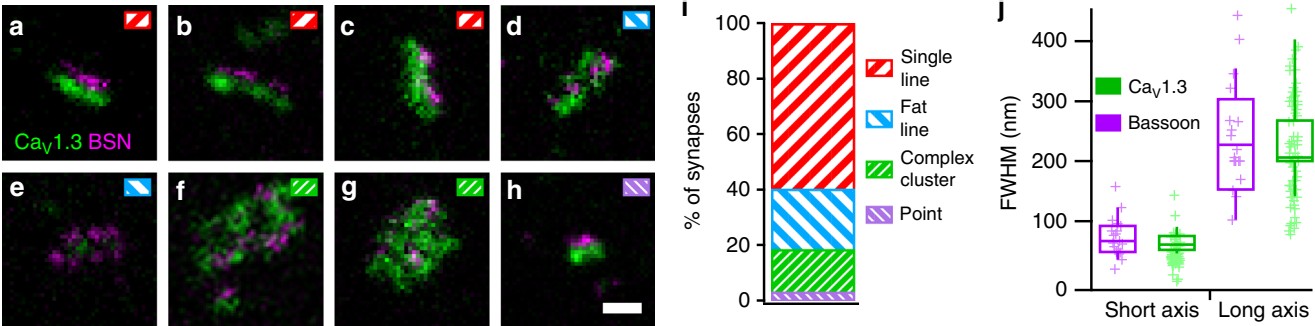

**Fig. 1** 2D nanoscale anatomy of IHC Ca²⁺-channel clusters. **a–h** Representative examples of AZs assumed to run roughly perpendicular to the imaging axis at the base of IHCs, with immunofluorescence for the Ca$_V$1.3 Ca²⁺ channel (green) and bassoon (magenta), a protein of the presynaptic AZ, imaged with 2D-STED microscopy. Most (~60%) of the clusters seemed to form thin stripes, with Ca$_V$1.3 and bassoon lying side by side (**a–c**), while some formed somewhat wider stripes, sometimes double stripes (~20%, **d,e**). About 15% of the AZs displayed a wider and more complex distribution of Ca$_V$1.3 and bassoon (**f,g**) and some clusters displayed small punctate staining for both proteins (~5%, **h**). Inset markers indicate our subjective classification of the displayed AZs according to **i**. Scale bar: 200 nm. **i** Summary of the distribution of different types of AZs as defined by the apparent shape of the individual AZs (**a–h**) measured with 2D-STED microscopy in our sample of $n = 138$ AZs from $N = 4$ mice. **j** Average dimensions of thin stripe-like clusters of Ca$_V$1.3 (green) or bassoon (magenta), as presented in **a–c**, obtained by fitting a 2D Gaussian function to the image and displayed as full-width at half-maximum (FWHM), from $n = 81$ AZs

2D-STED images, we were able to estimate the true length of the long axis of linear clusters by measuring the FWHM of the cluster signal along a line profile in 3D (Fig. 2b–e). The FWHM averaged to $461 \pm 18$ nm (median: 432 nm) and $450 \pm 13$ nm (median: 462 nm) for Ca$_V$1.3 and bassoon clusters, respectively (Fig. 2f; measurements of clusters tagged with both Abberior STAR635p and STAR580 were pooled, since due to the lower STED efficiency the resolution was not notably different). This indicates that the apparent length measured by 2D-STED microscopy underestimated the true length by approximately 45% for both Ca²⁺-channel and bassoon clusters. This was most likely caused by clusters that were tilted relative to the imaging plane and thus appeared shorter in the 2D-STED data. Indeed, performing the 2D-analysis on z-projections of the 3D-data resulted in average apparent lengths of $308 \pm 11$ nm for Ca$_V$1.3 and $286 \pm 10$ nm for bassoon clusters.

**Counting Ca²⁺ channels per AZ by confocal Ca²⁺ imaging.** We devised two different approaches to estimate the number of presynaptic Ca²⁺ channels at individual AZs: selective depletion of synaptic Ca²⁺ current and optical fluctuation analysis, both of which made use of confocal Ca²⁺ imaging.

For the first approach, we used focal microiontophoresis of the Ca²⁺-chelator ethylene glycol-bis(β-aminoethyl ether)-N,N,N',N'-tetraacetic acid (EGTA) to selectively reduce Ca²⁺ influx at a single AZ ($I_{Ca,AZ}$) by locally depleting free extracellular Ca²⁺ while maintaining the Ca²⁺ influx at the other AZs of the IHC. We employed fast confocal spinning-disk microscopy of patch-clamped mouse IHCs, loaded with the low affinity Ca²⁺-indicator Fluo-8FF ($K_D = 10$ µM) and a fluorescently tagged peptide marking ribbon-type AZs[30]. This allowed us to simultaneously image Ca²⁺ signals at several AZs in the confocal section (Fig. 3a, b) and also to switch to adjacent sections by means of piezo-driven rapid refocusing (step size 500 nm). Based on previous results[17], as well as given the low affinity of Fluo-8FF and the strong intracellular Ca²⁺ buffering (10 mM EGTA), we consider the observed Ca²⁺ signals to linearly relate to the voltage-gated Ca²⁺ influx.

We aimed for selective manipulation of a single AZ by (i) targeting a sufficiently isolated AZ, (ii) close apposition of the iontophoresis pipette to the membrane, and (iii) vivid bath perfusion (at least 1 ml min⁻¹). Selective manipulation of an AZ

was verified by monitoring signals from nearby AZs (Fig. 3b) in five confocal sections—the plane with the strongest Fluo-8FF fluorescence reduction of the targeted AZ as well as 500 and 1000 nm above and below it. We applied EGTA in a ramp-like manner after $\Delta F/F_0$ at the AZ had reached a steady state during a 200- or 270-ms long depolarization of the IHC to $-14$ mV (Fig. 3b, c). The reduction of synaptic Ca²⁺ influx resulting from local Ca²⁺ depletion was monitored by the decrease of the Fluo-8FF fluorescence at the targeted synapse ($\Delta(\Delta F/F_0)$, Fig. 3c, green arrow, on average down to 52%) and of the total IHC Ca²⁺ influx ($\Delta I_{Ca}$, Fig. 3c, black arrow). Next, we calculated the influence of the observed changes in neighboring AZs on the whole-cell calcium currents (Supplementary Fig. 2a) and corrected for this when estimating Ca²⁺ influx at the synapse under study. On average, the total signal change in neighboring AZs was $12 \pm 4\%$ that of the targeted AZ. We established a linear relationship between the IHC Ca²⁺ influx and the $\Delta F/F_0$ at the studied synapse (Fig. 3d, Supplementary Fig. 2b, c) and, by multiplying the slope of this linear fit with the maximal $\Delta F/F_0$ prior to EGTA application, obained $I_{Ca,AZ}$, which ranged from 9.9 to 31.1 pA (Supplementary Fig. 2d, mean: $25.0 \pm 1.9$ pA, median: 24.7 pA, $n = 19$ manipulated AZs from $N = 19$ IHCs). We then converted this into a distribution of total Ca²⁺-channel number per AZ assuming a single-channel current of 0.29 pA[15,24,31] ($-7$ mV, 5 mM $[Ca^{2+}]_e$) and an open probability of 0.4[24].

In these experiments, we picked AZs with strong Ca²⁺ signaling to ensure good signal-to-noise ratio, resulting in a high number of synaptic Ca²⁺ channels (mean: $215 \pm 16$, median: 213). We then used the mean slope of those 19 manipulated AZs (14.3 $\pm 1.4$ pA) to estimate $I_{Ca,AZ}$ for a set of 234 AZs representing most of the synapses of a large sample of 25 IHCs (mean synaptic Ca²⁺ influx: $14.6 \pm 0.4$ pA, median: 13.7 pA; coefficient of variation (CV): 0.5). The resulting estimates of total Ca²⁺-channel number revealed major heterogeneity among the AZs: their Ca²⁺-channel complement ranged from 28 to 329 with a mean of 125 and a median of 118 (Fig. 3e). We observed two outliers (>average + 4× standard deviation) featuring 431 and 555 Ca²⁺ channels that we assumed to be multi-ribbon AZs or unresolved neighboring synapse pairs and hence excluded from further analysis.

In order to validate our approach, we repeated the experiment in the presence of the Ca$_V$1.3 channel agonist BayK8644, which maximizes open probability[32]—reaching 0.82 in IHCs[24] (for

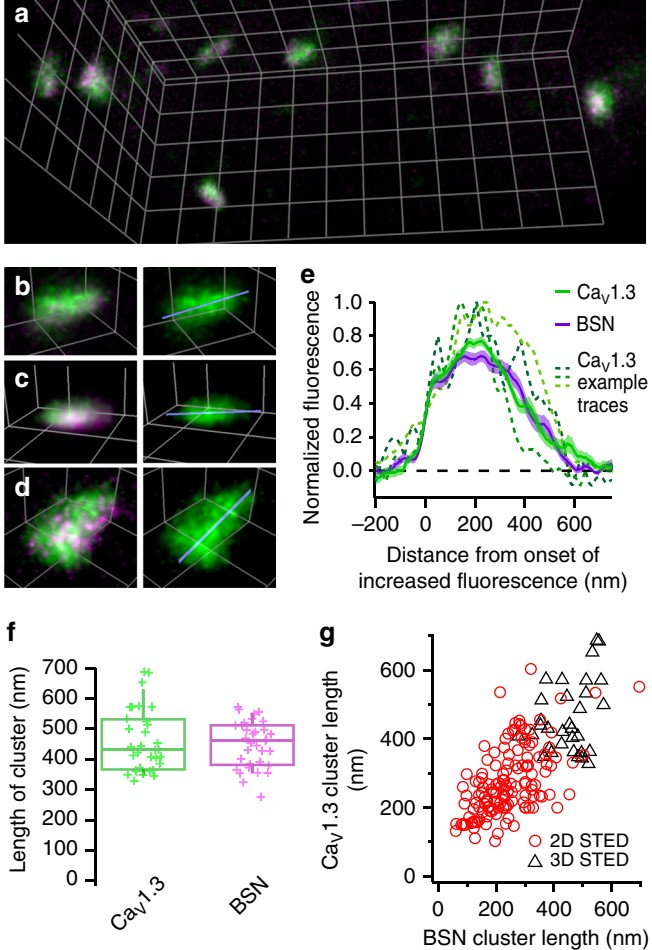

**Fig. 2** 3D nanoscale anatomy of IHC Ca$^{2+}$-channel clusters. **a–d** Volumetric displays of representative examples of Ca$_V$1.3 (green) and bassoon (magenta) immunofluorescence at IHC AZs, acquired with 3D-STED microscopy. In the right panels of **b–d**, Ca$_V$1.3 immunofluorescence is shown in isolation, blue lines indicate the 3D line profile plots used to estimate the length of the clusters by measuring the full-width at half-maximum of the line profile (**e**), here 455 nm (**b**), 389 nm (**c**), and 535 nm (**d**). Grid distance: 500 nm. **e** Average display of line profiles plotted through the 3D data from Ca$_V$1.3 (green) or bassoon (BSN, magenta) clusters along the long axis of Ca$^{2+}$-channel clusters ($n = 33$ AZs from $N = 2$ mice), with representative examples of line profiles from individual clusters of Ca$_V$1.3 shown as dotted green lines. **f** Average length of the clusters of Ca$_V$1.3 channels (green) or bassoon (magenta), as measured from the full width at half maximum of the line profiles summarized in **e**. Crosses indicate individual data points. **g** Comparison of the lengths of associated Ca$_V$1.3- and bassoon-clusters as measured by fitting a 2D Gaussian function (2D-STED) or by measuring the FWHM of a 3D line profile (3D-STED) indicates a strong correlation between both measures (Pearson's correlation coefficient of 0.71 for the combined data set)

lower estimates see ref. [15])—but does not change the single-channel current[32]. We argued that maxing out open probability would reduce potential errors in deriving the number of Ca$^{2+}$ channels from the synaptic Ca$^{2+}$ influx. We observed the expected doubling of $I_{Ca,AZ}$, but the slope of the relationship between $I_{Ca,AZ}$ and $\Delta F/F_0$ was comparable (Supplementary Fig. 2b, c, +BayK8644: 16.67 pA, $n = 12$ AZs, $N = 12$ IHCs, vs. −BayK8644: 16.04 pA, $n = 19$ AZs, $N = 19$ IHCs, not corrected for the EGTA effects on neighboring synapses in either case). Importantly, the estimates of the number of Ca$^{2+}$ channels per

AZ in the presence and absence of BayK8644 did not differ for the strong synapses analyzed (+BayK8644: mean = $258 \pm 39$ vs. −BayK8644: $239 \pm 18$, median: 220 vs. 235, $p = 0.779$, Mann–Whitney $U$ test, statistical power: 0.07).

Secondly, we counted Ca$^{2+}$ channels at single AZs by analyzing non-stationary fluctuations of Ca$^{2+}$-indicator fluorescence (Fluo-4FF, $K_D = 10 \, \mu M$) during the injection of deactivating Ca$^{2+}$ currents (tail currents, Fig. 4a). We depolarized the IHCs to +53 mV, past the (apparent) reversal potential of Ca$^{2+}$ in IHCs (~45 mV), opening Ca$^{2+}$ channels but not permitting Ca$^{2+}$ influx, then elicited Ca$^{2+}$-tail currents by hyperpolarization to −67 mV and recorded the accompanying transient increases in Fluo-4FF fluorescence at the center of the fluorescence hotspot (Fig. 4a, fluorescence plotted on inverted axis) by confocal spot detection[17]. The rationale is that fluctuations within ensembles of trials contain information on single-channel properties, a feature frequently exploited in non-stationary fluctuation analysis of membrane currents[21,22,24,33]. We buffered intracellular Ca$^{2+}$ with 10 mM EGTA and 1 mM Fluo-4FF, such that the Fluo-4FF fluorescence change reported Ca$^{2+}$ influx linearly[17]. We reasoned, therefore, that fluctuations of synaptic Ca$^{2+}$-indicator fluorescence among repetitive trials could likewise provide access to microscopic Ca$^{2+}$-channel properties at single AZs. However, while optical fluctuation analysis has proven helpful for estimating Ca$^{2+}$-channel numbers in small compartments in conjunction with a failure analysis (dendritic spines[34]), it remained to be elucidated whether an analysis of non-stationary fluctuations could work at AZs facing a large cytosolic volume, as is the case in IHCs. The analysis of channel-gating-related fluctuations of Ca$^{2+}$-indicator fluorescence is limited by the relatively low recording bandwidth resulting from the kinetics of Ca$^{2+}$ binding (0.412 $\mu M^{-1} ms^{-1}$) and unbinding (4 ms$^{-1}$)[35] of Fluo-4FF. Moreover, the contribution of the the Ca$^{2+}$ channels to the fluorescence signal varies dependent on their position within the cluster relative to the PSF of the microscope, due to uneven excitation and detection. In order to better match the signal to the limited recording bandwidth, we slowed Ca$^{2+}$-channel deactivation by the agonist BayK8644[21,24] (5 $\mu M$), which increases the channel's open time[32]. We addressed the impact of the inhomogeneous contribution of Ca$^{2+}$ channels by mathematical modeling, which indicated that the mean and variance of the fluorescence signal are both affected in a way causing the error in the estimation of the number of Ca$^{2+}$ channels to be rather limited (Supplementary Fig. 3).

We then isolated the ensemble variance (200 trials) resulting from Ca$^{2+}$-channel gating ('excess variance') by subtracting the detector noise and photon shot noise (Fig. 4b). As expected for gating-related variance, it peaked at intermediate fluorescence intensities (Fig. 4b, c). Finally, we related the variance to the mean for the ensembles of fluorescence transients and also for the corresponding whole-cell Ca$^{2+}$-tail currents (Fig. 4c). We estimated the total number of Ca$^{2+}$ channels ($N_{Ca}$) per AZ (fluorescence) and per IHC (current) by binomial fitting to both data sets (Fig. 4c). $N_{Ca}$ values per AZ ranged between 20 and 294 per AZ (Fig. 4d; mean: 78; median: 63; CV: 0.64). The total number of Ca$^{2+}$ channels per IHC, estimated from the fluctuations of the whole-cell Ca$^{2+}$ current, (mean: 1933; median: 1846; $n = 16$ IHCs) was consistent with previous estimates[22,24].

In summary, both experimental approaches reported somewhat different estimates for the mean number of Ca$^{2+}$ channels per AZ (125 and 78), suggesting either an overestimation of the number of channels during iontophoresis (e.g., due to additional effects on extrasynaptic Ca$^{2+}$ channels, which are difficult—if not impossible—to quantify from our experiments), or an underestimation in the optical fluctuation analysis experiments (e.g., due to activity of synaptic channels outside the microscope's

PSF), or a combination of both. Also, we cannot exclude a small contribution of an age dependence of the AZ $Ca^{2+}$-channel complement, because the two approaches made use of mice of different age (P15–17 for EGTA application and P21–28 for fluctuation analysis), for which we find subtle differences in the $\Delta F/F_{0,max}$ of the $Ca^{2+}$ indicator (Supplementary Fig. 4). Nevertheless, the observation of a large range of $Ca^{2+}$ channels per AZ by both approaches confirms that the previously reported heterogeneity of $Ca^{2+}$ signaling among the AZs of a given IHC[17] results from different numbers of $Ca^{2+}$ channels per AZ. Indeed, the CVs found for $Ca^{2+}$-channel number per AZ by both approaches (0.5 and 0.64) were comparable to the one previously reported for the maximal synaptic $Ca^{2+}$ influx (0.65[17]) and furthermore in good agreement with our data of the integrals of the 2D Gaussian fits to $Ca_V1.3$ immunofluorescence of $Ca^{2+}$-channel clusters (0.67, data from Fig. 1). This AZ heterogeneity might be related to functional diversity of the spiking behavior of postsynaptic spiral ganglion neurons[17,26]. Finally, if we determine the total number of synaptic $Ca^{2+}$ channels by multiplying the average number of $Ca^{2+}$ channels per AZ (as assessed by both methods) with the count of synapses per IHC—typically 12 in the area of the organ of Corti used in our measurements—one arrives at a value between 950 and 1500. Relating this to the total number of $Ca^{2+}$ channels per IHC (~1900, see above) indicates that approximately 20–50% of all channels localize extrasynaptically.

**Modeling $Ca^{2+}$ signals and $Ca^{2+}$ imaging at IHC AZs.** A previous study had estimated presynaptic $[Ca^{2+}]$ to reach ~3 µM at IHC AZs during depolarization[17]. This appears surprisingly low considering the large number of $Ca^{2+}$ channels established here. Most likely, spatial averaging due to the microscope's large PSF caused this low estimate, implying that the true $[Ca^{2+}]$ at the AZ might be far higher. We postulated that STED super-resolution imaging with its greatly reduced PSF size would provide much more accurate measurements of $[Ca^{2+}]$ at the AZ, allowing us to establish the concentrations occurring in vivo. First, we screened several $Ca^{2+}$-indicator dyes for compatibility with STED nanoscopy and found Oregon Green BAPTA-5N (OGB-5N) to be the best-suited $Ca^{2+}$ indicator with low affinity (Supplementary Figs 5, 6; reported $K_D = 32.5$ µM[36]). In order to predict what $[Ca^{2+}]$ could be expected at the IHC AZ and whether superresolution imaging could more accurately measure the local concentration, we performed reaction/diffusion simulations of the IHC AZ using the CalC modeling software[37]. Our model was based on the number and distribution of channels experimentally determined above (Table 1, Supplementary Fig. 7) and used OGB-5N as a $Ca^{2+}$ indicator. Seeking imaging conditions for optimal super-resolution $Ca^{2+}$ imaging, we generated several model implementations, changing the concentrations of $Ca^{2+}$ indicator (from 25 to 1000 µM) and non-fluorescent intracellular $Ca^{2+}$ chelators (from 0.8 mM EGTA and 0.4 mM BAPTA, emulating "physiological" $Ca^{2+}$ buffering in IHCs with both fast and slow binding kinetics[18], to 10 mM EGTA).

The model predicted that free $[Ca^{2+}]$ ($Ca^{2+}$ not bound to $Ca^{2+}$-indicator or non-fluorescent buffers) can reach up to 100–150 µM in close proximity (10–20 nm distance) to the channel mouth (Fig. 5a), but quickly drops as the distance to the channel

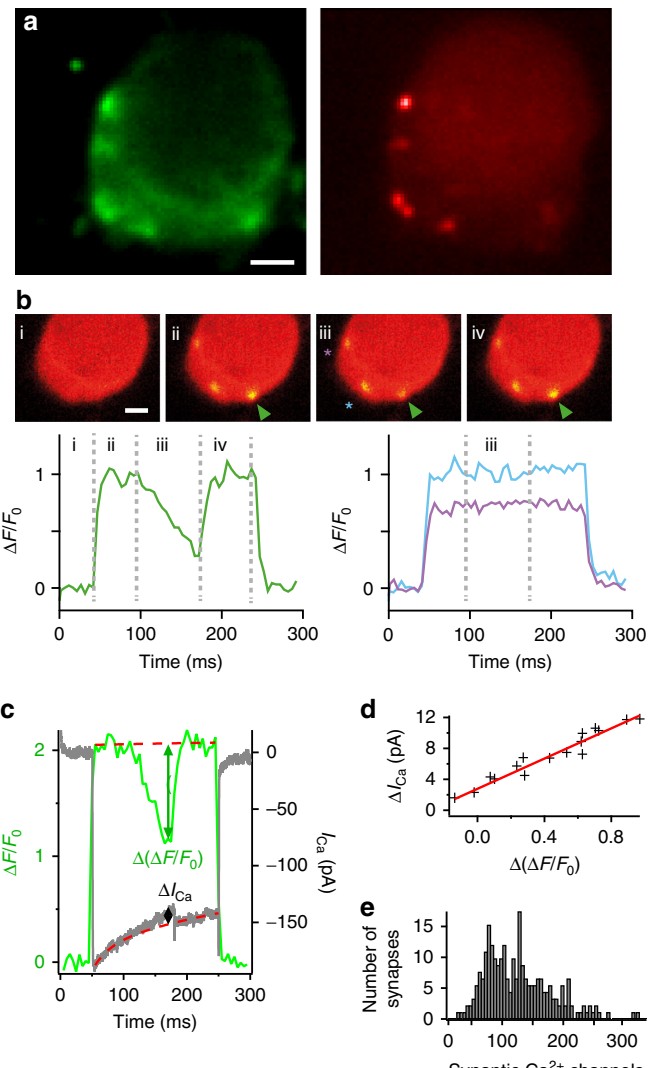

**Fig. 3** Estimating the number of $Ca^{2+}$ channels per AZ by confocal $Ca^{2+}$ imaging with selective suppression of $Ca^{2+}$ influx at individual AZs. **a** Representative image of hotspots of depolarization-evoked $Ca^{2+}$ influx, visualized by increased fluorescence of the $Ca^{2+}$-indicator Fluo-8FF (green, left), localized at AZs that are identified by a TAMRA-conjugated peptide (red, right) binding to the synaptic ribbon. Scale bar: 1 µm. **b** Hotspots of $Ca^{2+}$ influx were additionally monitored in nearby AZs, also in different focal planes (500 and 1000 nm above and below the targeted AZ). Only those recordings were used where the targeted AZ (green arrowhead) showed a clear increase in fluorescence (green trace, bottom left) from baseline (i) upon depolarization (ii), which decreased upon microiontophoresis of EGTA (iii) but returned to full intensity after the end of iontophoresis (iv) and where nearby AZs (blue and purple asterisks) showed no obvious change in fluorescence (blue and purple traces, bottom right) due to iontophoresis. Scale bar: 1 µm. **c** Representative recording showing modulation of the depolarization-evoked increase in fluorescence at a single AZ (green, left axis) and of the evoked whole-cell $Ca^{2+}$ current (gray, right axis) by microiontophoresis of EGTA from a micropipette close to the AZ. Changes in fluorescence increase ($\Delta(\Delta F/F_0)$) and in whole-cell current ($\Delta I_{Ca}$) due to local depletion of extracellular $Ca^{2+}$ by EGTA are estimated as the difference between measured data and a double-exponential function fitted to the data before and after iontophoresis (red dashed line). **d** Plotting $\Delta I_{Ca}$ against $\Delta(\Delta F/F_0)$ allowed conversion of the synaptic fluorescence increase of a single AZ into the corresponding synaptic $Ca^{2+}$ current. Data are from the recording shown in **c**. **e** Histogram of estimated numbers of $Ca^{2+}$ channels per synapse from recordings of depolarization-evoked Fluo-8FF-fluorescence intensity increase due to $Ca^{2+}$ influx. With the average slope of 14.3 pA, the hotspot intensities from a larger data set of $n = 234$ AZs ($N = 25$ IHCs) could be converted into currents. From these, an expected number of $Ca^{2+}$ channels could be calculated using the assumptions of a single-channel current of $i_{Ca} = 0.29$ pA and an open probability of $p_o = 0.4$. On an average, a synapse contained 125 $Ca^{2+}$ channels, with a median of 118 channels per AZ

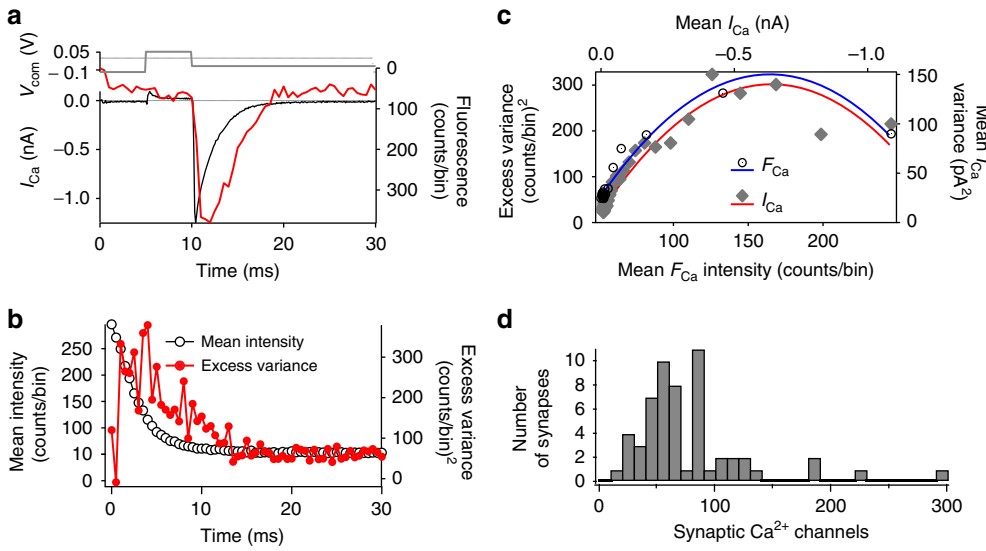

**Fig. 4** Estimating the number of Ca²⁺ channels per AZ by fluorescence fluctuation analysis. **a** Voltage protocol (upper panel) used to evoke Ca²⁺ influx, visualized by the fluorescence signal ($F_{Ca}$, red trace, lower panel, note that increase points down for comparison with current) of local Ca²⁺ influx at a single synapse and simultaneously by recordings of whole-cell Ca²⁺ current ($I_{Ca}$, black trace, lower panel). **b** Trial-to-trial variance (red trace) and mean (black circles) of fluorescence. **c** Plot of baseline-subtracted variance of synaptic fluorescence (open circles) or of whole-cell Ca²⁺ current (filled diamonds) against mean fluorescence or current, respectively, from the exemplary cell shown in **b**. Lines indicate the fits of quadratic functions to the fluorescence data (blue) or the Ca²⁺-current data (red). Extrapolation of these fits back to baseline allows calculation of the total number of channels in the cell (from the Ca²⁺-current data) or at one AZ (from the fluorescence data). **d** Histogram of the number of Ca²⁺ channels, obtained from quadratic fitting of variance against mean as shown in **d**, to data from $n = 57$ AZs from $N = 16$ IHCs

increases, in agreement with previous studies[3,8,18]. Ca²⁺-bound indicator, however, can be found at greater distances from the channel cluster due to lateral diffusion of indicator and free Ca²⁺, such that the Ca²⁺ domains visible in Ca²⁺ imaging are larger than the Ca²⁺-channel cluster itself and do not permit resolution of nanodomains near individual Ca²⁺ channels. Still, when we calculated local [Ca²⁺] from the ratio of bound to unbound Ca²⁺ indicator (simulating the process in Ca²⁺ imaging), we found concentrations of up to ~45 μM (Fig. 5b), which matched the modeled free [Ca²⁺] at the AZ fairly closely.

Evaluation of the model showed that low Ca²⁺-indicator concentrations could provide the most accurate approximations of local [Ca²⁺] (Fig. 5b, c). We therefore settled on the lowest feasible concentration of 25 μM. Similarly, since increasing the non-fluorescent Ca²⁺ buffers did not appreciably decrease the size of the Ca²⁺ domain (Fig. 5d), we decided on the aforementioned "physiological" imaging conditions, which would also more closely describe [Ca²⁺] at IHC AZs in vivo. To predict whether STED imaging could more accurately measure local [Ca²⁺], we convolved the simulated dye distributions with a 3D Gaussian function mimicking the PSF of a 2D-STED microscope (64 × 64 × 542 nm FWHM), and again calculated [Ca²⁺]. As expected, the spatial averaging (especially along the z-axis) caused by the PSF leads to a marked underestimation of [Ca²⁺], with maximum values of ~10 μM (Fig. 5e). Compared to a confocal PSF, however, the measured Ca²⁺ domain was much smaller, displaying a FWHM of 223 × 393 nm (STED) vs. 332 × 454 (confocal), indicating the potential benefits of STED Ca²⁺ imaging.

**Super-resolution imaging of presynaptic Ca²⁺ signals.** Building on the above results, we tested the feasibility of combining patch-clamp with live STED imaging in HEK-293 cells and IHCs, with regard to both the stability for patch-clamping and nanoscale imaging, as well as to the physiological condition of the cell.

During STED imaging, we observed neither strong mechanical drifts nor detrimental effects to the stability of the patch-clamp recording. Both HEK-293 and hair cells tolerated exposure to even the highest possible STED beam intensities for several seconds typically without showing signs of light-induced leak currents.

We then performed live STED Ca²⁺ imaging, using physiological [Ca²⁺]ₑ (1.3 mM) and 0.8 mM EGTA, 0.4 mM BAPTA, and 25 μM OGB-5N in the pipette solution. Aiming for AZs near the base of the cell (perpendicular to the optical axis), we depolarized the cell for 78 ms and imaged OGB-5N fluorescence hotspots in small (3 μm × 1.5 μm) xy-scans (see Fig. 6a–f for representative images). Hotspots appeared roughly spherical at confocal resolution, but STED revealed them to be elongated ellipsoids between 90–300 nm in width (Fig. 6g) and 200–450 nm in length (Fig. 6h), as measured by fitting a 2D Gaussian function (Fig. 6c). We found that both the short and the long axis were significantly smaller with STED Ca²⁺ imaging when compared to confocal Ca²⁺ imaging (Fig. 6a–h, 203 ± 7 nm vs. 303 ± 5 nm for the FWHM along the short axis and 305 ± 11 nm vs. 386 ± 8 nm for the FWHM along the long axis from $n = 55$ and $n = 74$ synapses for STED vs. confocal imaging, respectively, $p = 1.9e-21$, Student's T-test, statistical power: 1, and $p = 3.4e-11$, Mann–Whitney U test, statistical power: 1, for short and long axis, respectively), in good agreement with the results of our modeling (Fig. 5). Typically, about 12–18 mW of STED power (in the back aperture of the objective lens) was sufficient to fully resolve the fluorescence hotspots. Only in a few cases did higher STED beam powers reveal even smaller hotspots. This implies that the measured size of the typical hotspot was, in most cases, limited not by the microscope's lateral resolution, but by the size of the hotspots themselves, which is governed by the diffusion of Ca²⁺ and Ca²⁺ indicator and quickly reaches a steady state (see Supplementary Fig. 8). These functional observations therefore corroborate the stripe-like arrangement of presynaptic Ca_V1.3 immunofluorescence at wildtype AZs (Fig. 1a–e).

**Table 1 Modeling parameters.**

| Parameter | Value | Unit | Ref. | Comments |
|---|---|---|---|---|
| Free Ca$^{2+}$ | | | | |
| Diffusion coefficient | 0.223 | μm$^2$ ms$^{-1}$ | 67 | |
| Resting concentration | 50 | nM | 17 | |
| Fixed buffer | | | | |
| Diffusion coefficient | 0 | μm$^2$ ms$^{-1}$ | | |
| Dissociation constant ($K_D$) | 4.859 | μM | 68 | |
| Association rate ($k_{on}$) | 1.375 | μM$^{-1}$ ms$^{-1}$ | 68 | |
| Concentration | 610 | μM | 68 | |
| EGTA | | | | |
| Diffusion coefficient | 0.14 | μm$^2$ ms$^{-1}$ | 69 | |
| Dissociation constant ($K_D$) | 0.071 | μM | 70 | |
| Association rate ($k_{on}$) | 0.0105 | μM$^{-1}$ ms$^{-1}$ | 70 | |
| Concentration | 800 | μM | | Used in experiments. Simulated range: 800–10,000 μM. |
| BAPTA | | | | |
| Diffusion coefficient | 0.14 | μm$^2$ ms$^{-1}$ | 69 | Assumed to be identical to EGTA |
| Dissociation constant ($K_D$) | 0.17 | μM | 71 | |
| Association rate ($k_{on}$) | 0.45 | μM$^{-1}$ ms$^{-1}$ | 71 | |
| Concentration | 400 | μM | | Used in experiments. Simulated range: 0–400 μM. |
| OGB-5N | | | | |
| Diffusion coefficient | 0.1 | μm$^2$ ms$^{-1}$ | 72 | |
| Dissociation constant ($K_D$) | 195 | μM | | From this study |
| Association rate ($k_{on}$) | 0.25 | μM$^{-1}$ ms$^{-1}$ | 36 | |
| Concentration | 25 | μM | | Used in experiments. Simulated range: 25–1000 μM. |
| ATP | | | | |
| Diffusion coefficient | 0.14 | μm$^2$ ms$^{-1}$ | 73 | |
| Dissociation constant ($K_D$) | 2200 | μM | 73 | |
| Association rate ($k_{on}$) | 0.013 | μM$^{-1}$ ms$^{-1}$ | 73 | |
| Concentration | 68 | μM | | Calculated (3 mM Mg$^{2+}$, 2 mM ATP) |
| Gluconate | | | | |
| Diffusion coefficient | 0.2 | μm$^2$ ms$^{-1}$ | 74,75 | Calculated from ionic mobility relative to Ca |
| Dissociation constant ($K_D$) | 57,000 | μM | 42 | Assumption |
| Association rate ($k_{on}$) | 0.1 | μM$^{-1}$ ms$^{-1}$ | 42 | Used in experiments |
| Concentration | 130,000 | μM | | |
| Ca$^{2+}$ influx | | | | |
| Number of channels | 120 | | | From this study |
| Min. nearest-neighbor distance | 10 | nm | | |
| Cluster width | 67 | nm | | From this study |
| Cluster length | 430 | nm | | From this study |
| Single-channel current | 0.137 | pA | 24,31 | Adjusted for difference in [Ca$^{2+}$]$_e$, assuming 1.3 mM [Ca$^{2+}$]$_e$ |
| Open probability | 0.4 | | 24 | |
| Simulation parameters | | | | |
| Simulation volume size ($x$) | 800 | nm | | Fixed concentration of buffers and Ca$^{2+}$ at boundaries |
| Simulation volume size ($y$) | 800 | nm | | Fixed concentration of buffers and Ca$^{2+}$ at boundaries |
| Simulation volume size ($z$) | 300 | nm | | Fixed concentration of buffers and Ca$^{2+}$ at upper, reflective at lower boundary |
| Distance between nodes | 5 | nm | | |
| Simulation duration | 1 | ms | | Steady-state concentration is reached after ~0.3 ms |
| Simulation time steps | 100 | ns | | |

Parameters used to establish the CalC-based mathematical model of the IHC AZ. The full simulation script is available from http://www.innerearlab.uni-goettingen.de/materials.html

We applied the same imaging protocol to observe IHC AZs in mice lacking functional bassoon ($Bsn^{\Delta Ex4/5}$)[38]; these AZs typically lack the synaptic ribbon and hold fewer Ca$^{2+}$ channels. Previous STED imaging of Ca$_V$1.3 immunofluorescence had suggested that these channels form small, spot-like clusters[22], whereas live confocal Ca$^{2+}$ imaging (with 2 mM intracellular EGTA and 5 mM [Ca$^{2+}$]$_e$) had reported that the width of the presynaptic Ca$^{2+}$ domains at bassoon-mutant AZs was indistinguishable from wildtype AZs (FWHM of ~1 μm in both cases)[22]. Interestingly, we could clearly distinguish the bassoon-mutant AZs, which displayed significantly larger Ca$^{2+}$ domains (Fig. 6i–m, short axis length of 607 ± 50 nm using STED, $p = 8.6\text{e}{-}12$, Mann–Whitney $U$ test, statistical power: 1) than wildtype AZs. STED imaging did not reveal any further details within these large Ca$^{2+}$ domains, suggesting that Ca$^{2+}$ channels are more widely dispersed in the presynaptic membrane when tethering to the AZ by bassoon is

not available. This leads us to suspect that the small spot-like appearance of Ca$_V$1.3 immunofluorescence[22] was not physiological but artificial, e.g., resulting from the aggregation of untethered Ca$^{2+}$ channels during precipitation by the fixative. Based on our above modeling, we assume that the inability of the previous study to visualize the size differences of the presynaptic Ca$^{2+}$ hotspots in both genotypes is explained by the imaging conditions of that study, including a larger confocal PSF, high [Ca$^{2+}$ indicator], and higher [Ca$^{2+}$]$_e$ than [EGTA]. We further examined bassoon-deficient AZs in additional experiments using conditions designed to obtain a stronger signal (by enhancing Ca$^{2+}$ influx (10 mM [Ca$^{2+}$]$_e$), utilizing higher [OGB-5N] (300 μM) and [EGTA] (10 mM); "intensified conditions"; Fig. 6k–l), but this did not affect the appearance of the large Ca$^{2+}$ domains in confocal or STED Ca$^{2+}$ imaging. Interestingly, in wildtype AZs these "intensified conditions" (with the expected presynaptic [Ca$^{2+}$]

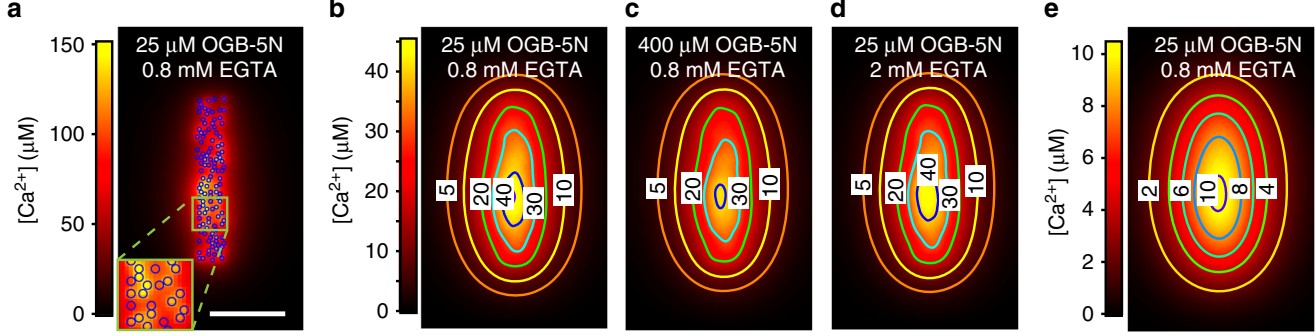

**Fig. 5** Theoretical reaction/diffusion model of $Ca^{2+}$ at the IHC AZ. **a** Theoretical model of $Ca^{2+}$ influx at a 430 nm × 67 nm cluster of 120 $Ca^{2+}$ channels (blue symbols, width of 10 nm) shows that the local increase in free $[Ca^{2+}]$ near the channel mouth, before the $Ca^{2+}$ ions bind to the $Ca^{2+}$-indicator or the non-fluorescent buffers, can reach values as high as 150 μM. Blue symbols indicate the positions of simulated $Ca^{2+}$ channels. Inset shows a magnification of the area marked by the green square. Scale bar: 100 nm. **b** $[Ca^{2+}]$, as calculated from the simulated distribution of OGB-5N at the synapse ("reported $[Ca^{2+}]$"), reaches peak values of 45 μM. The lateral diffusion of $Ca^{2+}$ ions and buffers makes it impossible to acquire the $[Ca^{2+}]$ at the channel mouth and results in an elongated $Ca^{2+}$ domain. Numbers indicate the $[Ca^{2+}]$ at the contour lines. **c** Increasing the simulated $[OGB-5N]$ (here to 400 μM) results in a lower reported peak $[Ca^{2+}]$ of ~40 μM. **d** Increasing the simulated $[EGTA]$ (here to 2 mM) results in a reported peak $[Ca^{2+}]$ of ~44 μM. **e** Same data as in **b**, but additionally convolved with a Gaussian PSF with a FWHM of 64 × 64 × 542 nm, mimicking 2D-STED imaging. The reported $[Ca^{2+}]$ after convolution reaches peak values of up to 10 μM, still considerably lower than the actual concentration near the channels

≪ $[OGB-5N]$) evoked somewhat larger $Ca^{2+}$ hotspots than observed under "physiological" conditions.

In summary, live STED $Ca^{2+}$ imaging is feasible and can be combined with patch-clamp for proper stimulus control and replenishment of $Ca^{2+}$ indicator. STED $Ca^{2+}$ imaging functionally demonstrated that $Ca^{2+}$ channels at AZs of wildtype IHCs are, indeed, organized in stripe-like clusters. The size of the hotspots of $Ca^{2+}$-indicator fluorescence matched well with the dimensions of $Ca^{2+}$ domains in theoretical models based on the morphology of $Ca^{2+}$-channel clusters as revealed by STED imaging of $Ca_V1.3$ immunofluorescence in fixed wildtype IHCs. The simulations show that the larger dimensions of the hotspots, when compared to those of the $Ca^{2+}$-channel clusters in morphological STED imaging, reflect the diffusion of both free and indicator-bound $Ca^{2+}$. Furthermore, $Ca^{2+}$ imaging of bassoon-mutant IHCs revealed large presynaptic $Ca^{2+}$ domains, showing that conclusions based solely on immunolabeled $Ca^{2+}$ channels in fixed tissue do not always represent the physiological organization and should be complemented by functional imaging with high resolution. We conclude that bassoon and/or the synaptic ribbon are required for appropriate clustering of $Ca^{2+}$ channels.

**Estimating presynaptic $[Ca^{2+}]$ from fluorescence lifetimes.** Next, we further adapted our method to not only observe the spatial distribution of presynaptic $Ca^{2+}$, but to quantitatively measure $[Ca^{2+}]$ at the AZ. For this, we established time-correlated single photon counting (TCSPC) of OGB-5N fluorescence with STED resolution (Fig. 7), a method based upon measuring the stark changes in fluorescence decay of the dye upon $Ca^{2+}$ binding[39,40]. OGB-5N is a low-affinity $Ca^{2+}$ indicator with a fluorescence lifetime when bound to $Ca^{2+}$ of $\tau = 3.24$ ns, which is markedly quenched to $\tau = 0.23$ ns in the absence of $Ca^{2+}$ (Fig. 7a, b, Supplementary Fig. 9a–d). The fluorescence decay of both the $Ca^{2+}$-bound and -free state is very well described by a mono-exponential function, allowing us to assume a true two-state system (Supplementary Fig. 9a–e). By double-exponential fitting of the OGB-5N fluorescence decay and analysis of the amplitudes of both components (Supplementary Fig. 9b–e), it is possible to calculate the number of photons per channel (fast or slow decay; Fig. 7c, d) and therefore to determine the ratio of $Ca^{2+}$-free and $Ca^{2+}$-bound dye for estimation of the actual $[Ca^{2+}]$. However,

given the nanoscale size of the $Ca^{2+}$ domains at IHC AZs, we expected high spatial resolution to be essential in obtaining accurate estimates of localized presynaptic $[Ca^{2+}]$.

Accurate, quantitative measurements of $[Ca^{2+}]$ require a precise understanding of the $Ca^{2+}$-indicator dye, its behavior within the actual experimental environment, and its interaction with the microscope in order to perform the necessary calibrations and corrections. Most prominently, the quantum efficiency and extinction coefficient of OGB-5N change upon binding to $Ca^{2+}$, resulting in a different brightness of the $Ca^{2+}$-bound and -free dye (Fig. 7d, e, Supplementary Fig. 10). Furthermore, the shortened lifetime of $Ca^{2+}$-free OGB-5N causes the STED de-excitation to be less efficient, resulting in different effective focal volumes for $Ca^{2+}$-bound and -free dye (Fig. 7e, f, Supplementary Fig. 11). Also, the STED beam itself impacts the measured fluorescence lifetime (Fig. 7b, Supplementary Figs. 11, 12): by its very nature, STED causes an additional fast decay of fluorescence ($\tau = 0.19$ ns), which needs to be considered to avoid overestimating the free (fast) dye contribution during STED measurements (Supplementary Fig. 12). When imaging with STED, we therefore excluded ("blanked") the data recorded in the first 300 ps (i.e., during the STED laser pulse) from the fitting routine, resulting in only minimal errors (see Methods).

Furthermore, in contrast to the in vitro calibration, where the $Ca^{2+}$-bound state of OGB-5N could be either saturated (>99%, 10 mM $[Ca^{2+}]$) or completely de-populated (<0.5%, 10 mM EGTA), this was not the case when measuring inside living IHCs. Even when patching a cell with 60 mM $[Ca^{2+}]$, the contribution from the slow lifetime component never surpassed 60% (Supplementary Fig. 13). This might be due to the dye interacting with molecules inside the cell, which change its properties, as has previously been described elsewhere[41]. To incorporate this into our calculations, we devised two independent approaches (Fig. 7f–h): (i) rescaling the dynamic range of the dye to match the reduced range observed in IHCs ("scaled lifetime ratios") and (ii) expanding our model system by considering a $Ca^{2+}$-insensitive dye fraction (see Methods and Supplementary Methods for a detailed derivation of the calculations).

Finally, the $Ca^{2+}$-binding behavior of OGB-5N varied strongly depending on its immediate experimental environment. Initially, we performed in vitro calibrations of $Ca^{2+}$ binding by OGB-5N in "minimalistic" solutions with well-known $[Ca^{2+}]$ (containing only

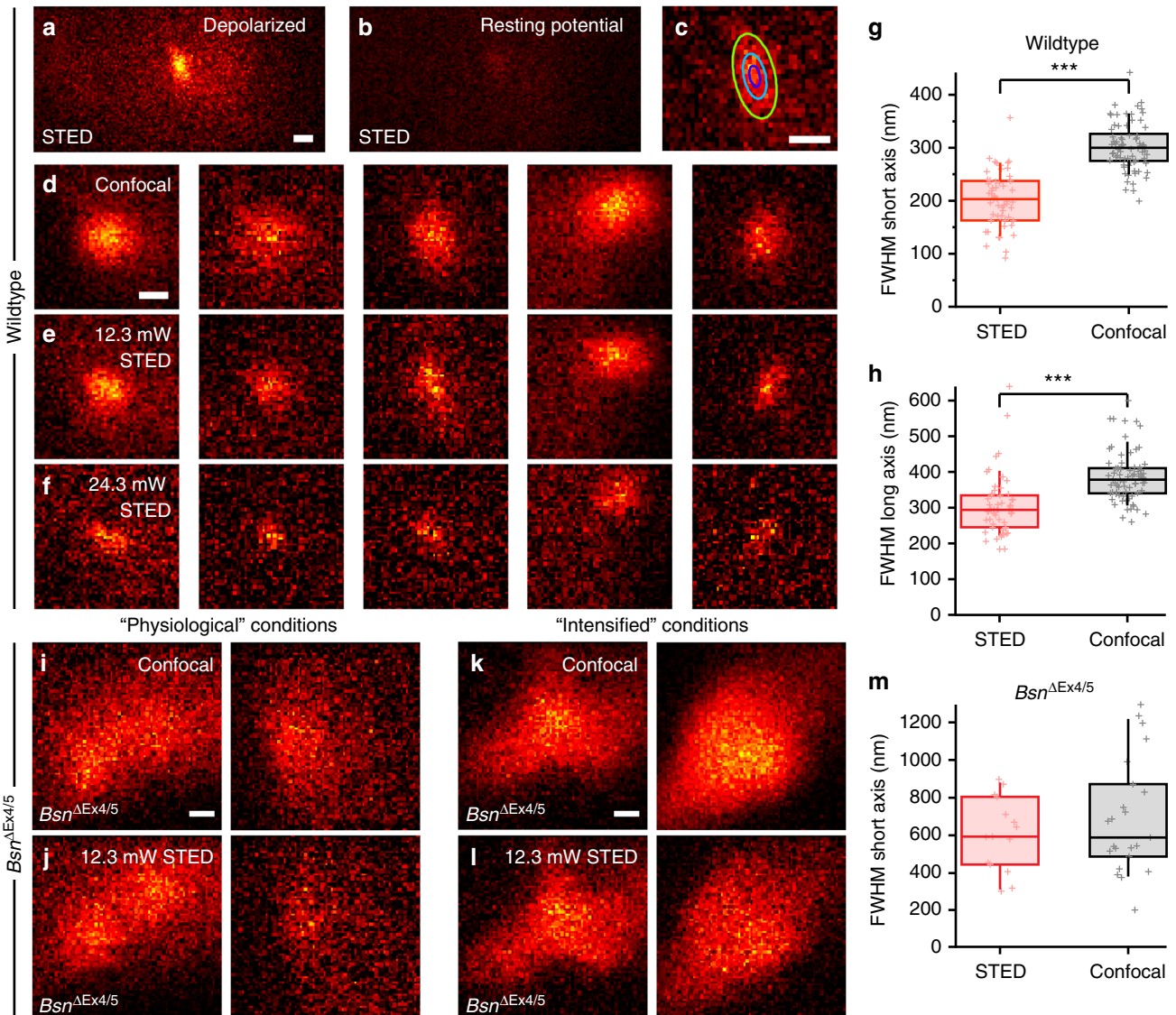

**Fig. 6** Super-resolution imaging of presynaptic Ca²⁺ signals in IHCs. **a–c** Hotspot of OGB-5N fluorescence in a wildtype mouse IHC in "physiological conditions" (same in **d–f**, 25 µM OGB-5N, 0.8 mM EGTA, and 0.4 mM BAPTA inside the pipette, 1.3 mM [Ca²⁺]ₑ), evoked by a 78-ms depolarization of the cell to −14 mV (**a**), as well as the corresponding fluorescence signal at the holding potential of −84 mV (**b**), both imaged using STED. **c** A 2D Gaussian fit to the background-subtracted hotspot image. Scale bars: 200 nm. **d–f** Background-subtracted images of depolarization-evoked hotspots of OGB-5N fluorescence, indicating synaptic Ca²⁺ influx, each imaged in confocal mode (**d**), with 12.3 mW STED laser intensity (**e**), and with 24.3 mW STED laser intensity (**f**). Increasing STED power leads to smaller and more elongated hotspots. Scale bar: 200 nm. **g, h** Significantly increased resolution of Ca²⁺ imaging by STED microscopy: the dimensions of the Ca²⁺ domains were measured by fitting a 2D Gaussian function to the background-subtracted images. Both short (**g**) and long axis (**h**) of the Gaussian were significantly smaller when measured with STED microscopy (red, n = 55 synapses, N = 43 IHCs, STED power 12.3–35 mW) compared to confocal microscopy (black, n = 74 synapses, N = 55 IHCs) with p = 1.7e−21 for **g**, Student's T-test, statistical power: 1, and p = 3.3e−11 for **h**, Mann–Whitney U test, statistical power: 1. **i, j** Background-subtracted images of two representative depolarization-evoked hotspots of fluorescence from Bsn^ΔEx4/5 mouse IHCs, imaged in "physiological conditions" (see **a–c**) in confocal mode (**i**) and with 12.3 mW STED power (**j**). The hotspots are much larger than those from wildtype animals (**a–h**) and do not decrease in size when using STED microscopy. Scale bar: 200 nm. **k, l** Same as in **i, j**, but imaged in "intensified conditions" (300 µM OGB-5N and 10 mM EGTA inside the pipette, 10 mM [Ca²⁺]ₑ). **m** The size of hotspots of Bsn^ΔEx4/5 mice (as measured by fitting of a 2D Gaussian function) is unchanged when using STED (red, n = 15 synapses, N = 13 IHCs, STED power 12.3–24.3 mW) compared to confocal recordings (black, n = 23 synapses, N = 21 IHCs), with p = 0.605, Mann–Whitney U test, statistical power: 0.176

Ca²⁺, Ca²⁺ buffer, and OGB-5N) and found a $K_D$ of ~40 µM (Supplementary Fig. 14a), close to the published values of 30–40 µM[36]. When repeating the same calibrations in solutions designed to closely match the intracellular solutions used in patch-clamp (with additional Ca²⁺ and only citrate as Ca²⁺ buffer to set the free [Ca²⁺]), we found a much higher effective $K_D$ ($K_{eff}$) of ~195 µM (Supplementary Fig. 14b). Such behavior has been observed before[42], and might be explained by differences in ionic

strength of the solution and by the binding and buffering capabilities of additional components of the intracellular solution[41,43]. As these second conditions most closely matched the actual experimental conditions, we used this value ($K_{eff}$) for all further calculations (Fig. 7h,i).

We applied the TCSPC method to measure [Ca²⁺] at IHC AZs at confocal and STED resolution by measuring the fluorescence decay before and during depolarization of the patch-clamped

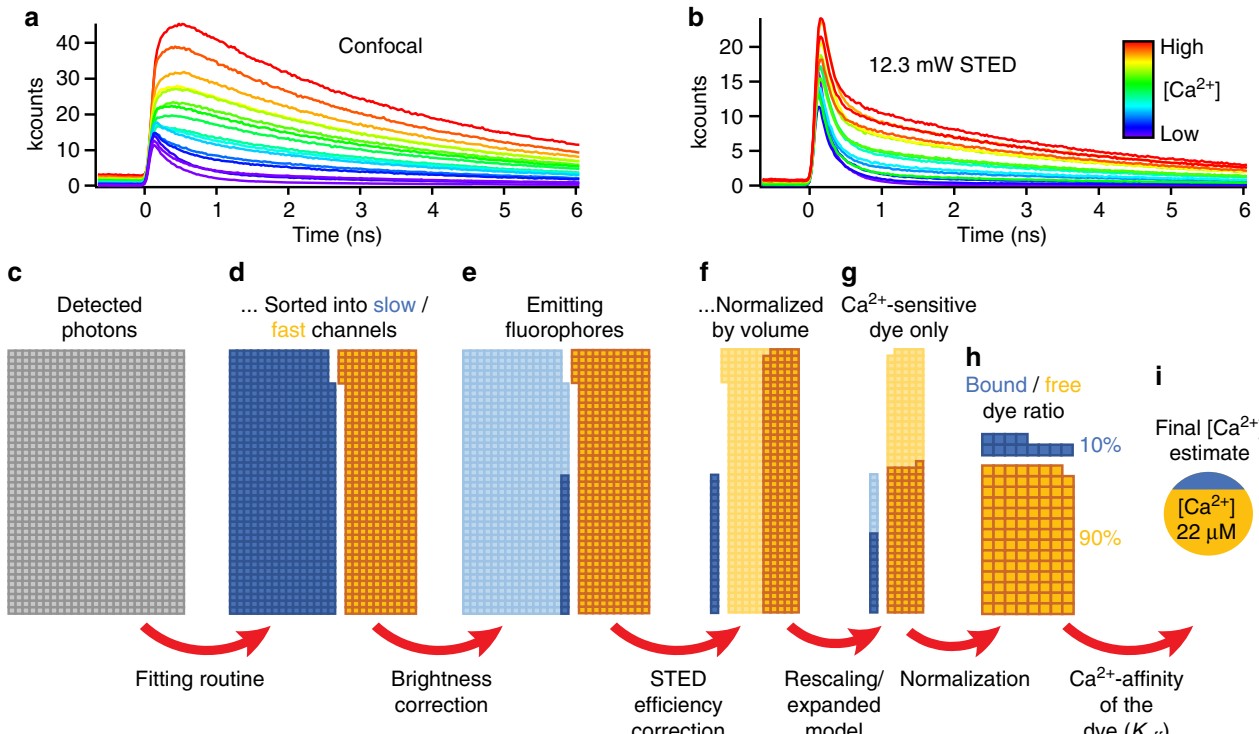

**Fig. 7** Calculation of [Ca²⁺] from measured photons by application of correction factors. **a** Time-correlated single photon counting (TCSPC) reveals how the fluorescence lifetime traces of OGB-5N are highly sensitive to different [Ca²⁺], with sensitivity over several orders of magnitude (measured in solution, 10 nM < [Ca²⁺] < 90 mM, traces averaged from 10 s recordings). Both amplitude and average lifetime of the fluorescence signal increase with rising [Ca²⁺]. **b** Even though STED imaging (here, with 12.3 mW of STED power) introduces an additional (fast) component to the fluorescence decay, the recorded fluorescence lifetime signal is still very sensitive to different Ca²⁺ concentrations. **c–i** Detailed knowledge of the dye characteristics is needed to calculate accurate [Ca²⁺] estimates from the recorded TCSPC signal. After measuring the fluorescence decay of OGB-5N with TCSPC, the incoming photons (**c**, each box representing one photon) can be sorted into two channels, one corresponding to the fast (yellow) and one to the slow lifetime (blue), by fitting the fluorescence decay with a bi-exponential function (**d**, Supplementary Fig. 9). Downscaling the number of slow photons by the brightness correction factor $b$ takes the diminished brightness (smaller quantum efficiency and extinction coefficient) of free vs. Ca²⁺-bound OGB-5N into account (**e**, Supplementary Fig 10), converting the number of fluorescence photons (per channel) into an approximation of the number of emitting fluorophores (molecules of OGB-5N). The STED efficiency correction factor $s$ compensates the larger effective focal volume of the Ca²⁺-free dye (due to decreased STED efficiency), which (uncorrected) results in a larger number of sampled molecules than for Ca²⁺-bound dye (**f**, Supplementary Fig. 11). Finally, in IHCs (as opposed to in vitro), some of the dye appears not to be sensitive to Ca²⁺ anymore (**g**, Supplementary Fig. 13). When taking this into account (**g**, either by rescaling the ratios of bound over free dye or by introducing a Ca²⁺-insensitive dye fraction to the model), a corrected ratio of bound over free dye can be calculated (**h**) which allows calculating the [Ca²⁺] from the previously measured effective dissociation constant $K_{eff}$ (**i**, Supplementary Fig. 14)

IHCs in an $8 \times 8$ $xy$-matrix of points (spot detection[17,44], each separated by 50 nm) covering an AZ (Fig. 8a–f). Taking into account the response function of the instrument, we sorted the total photon counts according to whether they originated from either Ca²⁺-free OGB-5N (fast lifetime) or Ca²⁺-bound OGB-5N (slow lifetime). After applying the aforementioned corrections, the ratio of fast and slow lifetime components allowed us to approximate the [Ca²⁺] at the AZ during depolarization (Fig. 8e, f, red traces) and at rest (black traces) both with confocal and STED resolution. The calculated [Ca²⁺] distributions typically matched the observed shape of the AZ Ca²⁺ domains, with peak [Ca²⁺] values agreeing with or slightly above our mathematical simulations. Both calculation methods were in agreement with each other, finding peak [Ca²⁺] in the center of the depolarization-evoked presynaptic Ca²⁺ domain to be slightly higher when imaging using STED ("scaled lifetime ratios": Fig. 8g; STED: 14.8 ± 1.9 μM [CV: 0.5] vs. confocal: 11.4 ± 1.6 μM [CV: 0.6], "expanded model": Fig. 8h; STED: 19.7 ± 2.6 μM [CV: 0.56] vs. confocal: 15.1 ± 2.6 μM [CV: 0.75]). Both calculation methods demonstrated a clear, yet modest advantage of 2D-STED Ca²⁺ imaging (over confocal) for reporting [Ca²⁺] close to the Ca²⁺-channel cluster. Clearly, however, the low resolution along the

optical ($z$) axis of our 2D-STED microscope still caused significant spatial averaging, conflating the high [Ca²⁺] within the nanoscale space around the Ca²⁺ channels with the lower [Ca²⁺] further away from the AZ.

To approach this issue, we modified our 2D-STED experiments to further reduce the volume from which [Ca²⁺] was sampled. For this, we shifted the focus of the objective below the AZ while measuring [Ca²⁺], so that the majority of the volume illuminated by the microscope's PSF was outside the cell and thus devoid of dye (Fig. 9a). As the focus was moved outside the cell, the recorded fluorescence dropped (e.g., Figure 9b), yet the measured Ca²⁺ concentration indeed increased with increasing distance beneath the membrane (Fig. 9c, d), reaching average values as high as 25 ± 7 μM and peak values between 45 and 50 μM (for STED recordings) at a distance of 800 nm from the brightest point of the hotspot (and thus most likely the membrane). This agrees with our modeling that predicted similar concentrations (Fig. 9d, dashed lines). These recordings displayed high variance (especially when recording with STED), most likely because (due to drift) it was difficult to place and keep the recording in the exact center of the fluorescent hotspot, where [Ca²⁺] is at its highest.

## Discussion

In the present study, we used high- and super-resolution light microscopy to quantify the number and spatial organization of synaptic $Ca^{2+}$ channels and the resulting $Ca^{2+}$ domains at a presynaptic AZ that is experimentally well accessible. We found

that most of the IHC AZs form linear clusters that run in parallel to the presynaptic density, marked by the presynaptic scaffold bassoon (Fig. 1). In wildtype IHCs, the dimensions of the $Ca^{2+}$ domains observed during voltage-gated $Ca^{2+}$ influx (Fig. 6g, h) were compatible with those of the immunolabeled $Ca^{2+}$-channel clusters, when considering diffusional spread of $Ca^{2+}$ and $Ca^{2+}$ indicator predicted by a theoretical model (Fig. 5e). Observations of intramembrane particles in the presynaptic membrane in freeze-fracture electron micrographs of frog hair cells[21] had previously suggested similar arrangements of presynaptic $Ca^{2+}$ channels. However, the molecular identity of these particles remained unclear.

Our study corroborates the notion that bassoon and/or the synaptic ribbon organize the spatial arrangement of $Ca^{2+}$ channels. While previously published immunohistochemical data suggested a punctate clustering of $Ca^{2+}$ channels in the absence of bassoon[22], our $Ca^{2+}$-imaging data indicate a widespread distribution of $Ca^{2+}$ channels in the presynaptic membrane (Fig. 6i–m). This discrepancy might be explained by an artificial clustering of $Ca^{2+}$ channels during immunohistochemistry, which in the absence of bassoon appear to be less efficiently localized to the AZ. In the presence of bassoon, on the other hand, the physiological organization of $Ca^{2+}$-channel clusters appears to be well represented by immunohistochemistry: the elongated depolarization-evoked presynaptic $Ca^{2+}$ domains seen in STED $Ca^{2+}$ imaging are compatible with the stripe-like $Ca_V1.3$ immunofluorescence. In some instances, we observed more spatially extended synaptic $Ca^{2+}$ signals in IHCs. These might represent the more complex clusters of $Ca^{2+}$ channels (Fig. 1f, g). Whether these, in turn, represent especially large AZs or cases where multiple ribbons are situated at one AZ (as is sometimes observed in electron micrographs[45]) will need to be clarified in further experiments.

It will now be interesting for future studies to investigate the topography of the individual channels within the clusters, e.g., by SDS-replica labeling[9], and ideally their spatial relationship to synaptic vesicles tethered at the AZ. This will provide important input into biophysical modeling of how $Ca^{2+}$ influx couples to exocytosis of synaptic vesicles. Biophysical experiments and modeling have so far indicated that, at the IHC AZ, exocytosis of a given readily releasable vesicle is under "$Ca^{2+}$-nanodomain-like" control of few $Ca^{2+}$ channels with a mean effective coupling distance between mouth of channel and $Ca^{2+}$ sensor of

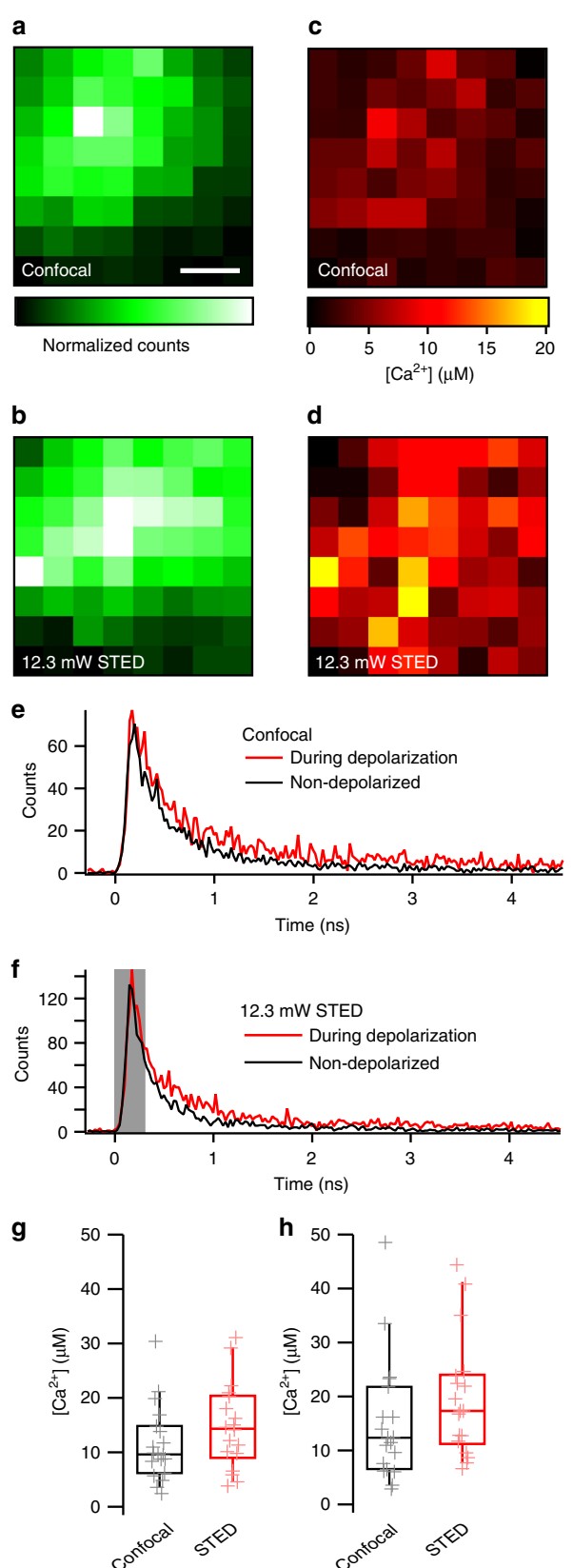

**Fig. 8** Measuring synaptic $Ca^{2+}$ concentration by fluorescence lifetime imaging. **a**–**d** Representative data from two $8 \times 8$ (50 nm steps in $x$ and $y$) pixel raster scans of a hotspot of $Ca^{2+}$-influx-evoked fluorescence at the AZ, depicting the rise in fluorescence during depolarization (left, shown for confocal imaging (**a**) and 12.3 mW STED imaging (**b**)) and the corresponding $Ca^{2+}$ concentrations calculated from the lifetime data (right, **c** and **d** for confocal and STED, respectively). Scale bar: 100 nm. **e**, **f** Fluorescence lifetime traces from single pixels in the same recordings shown in **a**–**d**, acquired with confocal (**e**) and STED imaging (**f**). Red traces show the fluorescence data during depolarization of the IHC, black traces at rest. For analysis of the STED imaging traces, we employed a blanking range (gray box). Data points inside this range were excluded from the fitting procedure (used to establish the ratio of $Ca^{2+}$-bound and -free dye) to avoid influence of the artificial STED-evoked short lifetime component. **g** Average maximum $Ca^{2+}$ concentrations (black symbols) calculated as "scaled lifetime ratios" from the fluorescence lifetime data of recordings acquired using both confocal imaging (left, $n = 19$ AZs, $N = 19$ IHCs) and STED imaging (right, $n = 18$ AZs, $N = 18$ IHCs) show an increase in measured $Ca^{2+}$ concentration using STED imaging. **h** Same as in **g**, calculated considering a $Ca^{2+}$-independent background signal according to Equation 11

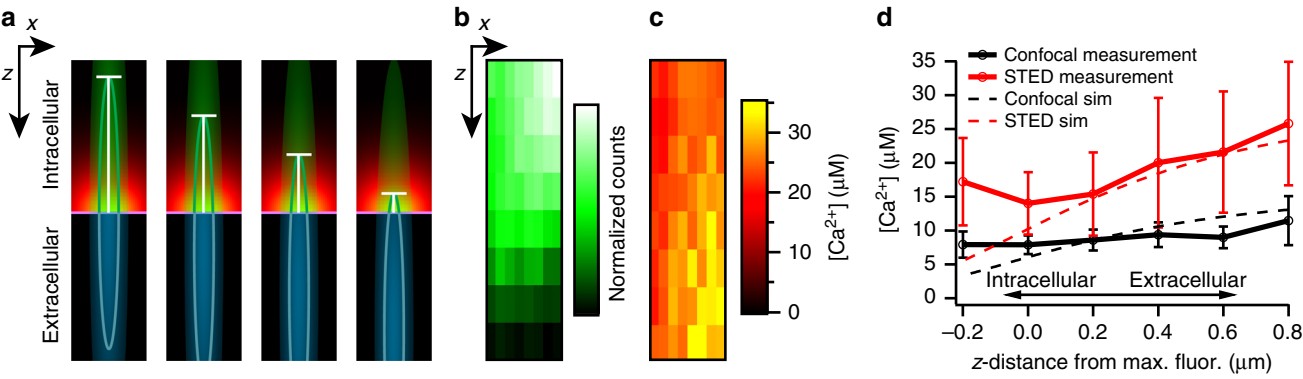

**Fig. 9** Decreasing the focal volume of Ca²⁺ imaging by moving the PSF in z. **a** Schematic drawing illustrating the decreased volume of intracellular dye that is sampled when moving the focus of the microscope down below the membrane in an xz-scan. Green lines indicate the FWHM of the PSF. **b** Representative data from an 8 × 8 pixel raster scan in x (50 nm steps) and z (200 nm steps) of a hotspot of Ca²⁺-influx-evoked OGB-5N fluorescence at the AZ, depicting the rise in fluorescence during depolarization. **c** The corresponding Ca²⁺ concentrations calculated from the lifetime data of the recording shown in **b**. Notice the increase in [Ca²⁺] toward the bottom of the plot. **d** Average maximum Ca²⁺ concentrations ("scaled lifetime ratios") calculated from the fluorescence lifetime data of recordings acquired using both confocal imaging (black circles, n = 10 AZs, N = 10 IHCs) and STED imaging (red circles, n = 13 AZs, N = 13 IHCs), overlaid on top of the modeling data (from the model shown in Fig. 5) convolved with a 243 nm × 243 nm × 542 nm Gaussian "PSF" ("confocal", black dotted line) or a 64 nm × 64 nm × 542 nm Gaussian "PSF" ("STED", red dotted line). Experimental data have been aligned in z relative to the maximal fluorescence intensity as estimated from a 1D Gaussian fit to the average fluorescence per z-line during stimulation and binned in 200-nm z-steps

17 nm[8,18,24,46], which we postulate to be common to IHC AZs regardless of their size. Using two independent methods to assess the number of Ca²⁺ channels at a single synapse, we found on average approximately 80–120 Ca²⁺ channels per AZ, whereby the counts varied dramatically between AZs (20–330). The wide distribution of lengths we measured for the long axis of Ca$_V$1.3 clusters by STED microscopy (Fig. 2f, g) suggests that much of the variance in presynaptic Ca²⁺ influx is realized through varying the length of Ca$_V$1.3 clusters. In contrast, the widths of the clusters appear to be more similar among AZs. This is compatible with the hypothesis that the topography of Ca²⁺ channels and membrane-proximal vesicles is conserved at AZs with different numbers of Ca²⁺ channels. Such presynaptic heterogeneity is a candidate mechanism for the diversity in firing properties of postsynaptic spiral ganglion neurons[17,47,48,26].

IHC ribbon-type AZs achieve the high average density of 4000 Ca²⁺ channels per μm² (~120 channels on ~0.03 μm²)—considerably higher than typically found in central nervous system (CNS) neurons where it ranges from 500 to 1500[9,12,13]—likely through a protein network organized by bassoon[22,49] and including the synaptic ribbon[22,50], RIM[51,52], and RIM-binding protein[53,54]. Given this very high density it seems likely that the Ca²⁺ influx at IHC AZs serves additional roles besides triggering synaptic release, e.g., increasing [Ca²⁺] to facilitate vesicle resupply[55]. The distance between Ca²⁺ channels and synaptic vesicles appears to be comparable between IHC AZs and some CNS AZs[56–58], which in some cases employ a placement of synaptic vesicles around the perimeter of Ca²⁺-channel clusters[9]. Thus, it seems possible that similar mechanisms are used in these presynaptic preparations, even though IHCs make use of a different set of proteins for the release of synaptic vesicles[59]. Our Ca²⁺-imaging data illustrates how disruption of bassoon disintegrates this sophisticated supramolecular machinery at the AZ in a manner reminiscent of what was seen at drosophila neuromuscular junctions upon mutation of the scaffold bruchpilot[60]. In addition, our study validated the previously assumed presence of extrasynaptic channels in IHCs[8,20,24], which form a considerable fraction of 20–50%. In CNS AZs, a subdivision into smaller clusters has been described, with a 1:1 relationship between the number of Ca²⁺-channel clusters and the number of vesicular

release sites[13]. Future analysis, e.g., employing MINFLUX microscopy[61] or electron microscopy of immunolabeled SDS-replica[9], will be required to test such a scenario for IHCs. Interestingly, despite the different complement of Ca²⁺ channels per AZ, the number of individual channels per release site appears comparable: nine channels/site reported in parallel fiber synapses[13] vs. 80–120 channels for 10–15 release sites[62] at IHC AZs.

Our study also shows, through theoretical modeling and direct [Ca²⁺] measurements, that insufficient spatial resolution causes artificially low estimates of AZ [Ca²⁺], because "spatial averaging" conflates the true [Ca²⁺] levels directly at the AZ with lower-[Ca²⁺] regions further from the Ca²⁺-influx sites. By establishing fluorescence lifetime Ca²⁺ imaging at confocal and 2D-STED resolution, we indeed saw higher (average) peak AZ [Ca²⁺] using 2D-STED (15–20 μM) compared to confocal microscopy (10–15 μM). We saw a further increase when additionally restricting the focal volume within the cell (by focusing slightly below the AZ membrane) up to 25 μM (average) with peak values up to 50 μM. Further refinement of the technique by application of 3D-STED microscopy can be expected to lead to more precise measurements of [Ca²⁺] in the immediate proximity of the Ca²⁺-channel cluster. We note, however, that functional 3D-STED imaging in living tissue faces challenges, such as the tissue-induced aberrations distorting both STED donuts in a different fashion, higher required laser powers, as well as the seemingly trivial problem of locating Ca²⁺-indicator hotspots with the reduced PSF size. Furthermore, we believe the comparatively modest increase in [Ca²⁺] measured with STED shows that our STED-efficiency correction is causing us to further underestimate [Ca²⁺]: the lower STED efficiency for the Ca²⁺-unbound dye means we are sampling the Ca²⁺-free dye from a much larger volume than the Ca²⁺-bound dye. So even if the focal volume of the bound dye is contained within the immediate AZ, the focal volume of the free dye might stretch considerably farther, encompassing more areas with lower [Ca²⁺] (and thus more free dye). Our STED-efficiency correction, however, is based on measurements within a homogeneous distribution of bound/unbound dye. This effect could be accounted for by either significantly increasing the spatial resolution or by considering the physical dimensions of the AZ when correcting.

  

In summary, using novel approaches we present an in-depth quantification of presynaptic $Ca^{2+}$ signaling using the hair cell AZ as a model system. We propose that IHCs vary the strength of their AZs primarily by scaling the length of the ribbons, presynaptic densities, and $Ca^{2+}$-channel clusters.

## Methods

**Animals**. C57B6/N mice and mice carrying a deletion of exons 4 and 5 of the bassoon gene ($Bsn^{\Delta Ex4/5}$)[38] were used at the age of postnatal day 26 (P26) to P33 (for optical fluctuation analysis) or P15 to P18 (for all other experiments). Both male and female mice were used. All experiments complied with German national animal care guidelines and the guidelines issued by the University Medical Center Göttingen.

**Immunohistochemistry**. Freshly dissected apical cochlear turns were fixed in methanol for 20 min at −20 °C. Thereafter, the tissue was washed three times for 10 min in PBS and incubated for 1 h in goat serum dilution buffer (GSDB) (16% normal goat serum, 450 mM NaCl, 0.3% Triton X-100, and 20 mM phosphate buffer, pH 7.4) in a wet chamber at room temperature. Primary antibodies were diluted in GSDB and applied overnight at 4 °C in a wet chamber. After washing three times for 10 min (wash buffer: 450 mM NaCl, 20 mM phosphate buffer, and 0.3% Triton X-100), the tissue was incubated with secondary antibodies in GSDB in a wet light-protected chamber for 1 h at room temperature. Then, the preparations were washed three times for 10 min in wash buffer and one time for 10 min in 5 mM phosphate buffer, placed onto the glass microscope slides with a drop of fluorescence mounting medium (Mowiol), and covered with thin glass coverslips. The following antibodies were used: mouse anti-Sap7f407 to bassoon (1:600, Abcam ab82958), rabbit anti-$Ca_V$1.3 (1:75, Alomone Labs ACC-005), STAR 580-tagged goat-anti-rabbit or goat-anti-mouse (1:200, Abberior 2-0002-005-1 or 2-0012-005-8), and STAR 635P-tagged goat-anti-mouse or goat-anti-rabbit (1:200, Abberior 2-0002-007-5 or 2-0012-007-2). 2D- and 3D-STED immunofluorescence images were acquired on an Abberior Instruments Expert Line 775 nm 2-color STED microscope, with excitation lasers at 561 nm and 633 nm and a STED laser at 775 nm, 1.2 W, using a 1.4 NA 100× oil immersion objective. Images were acquired with pixel sizes of 20 × 20 nm (2D-STED) or 40 × 40 × 40 nm (3D-STED). Volumetric display of 3D stacks was performed using the software Imaris (Bitplane, Zurich, Switzerland). Images were analyzed using Igor Pro 6 software (Wavemetrics, Lake Oswego, OR, USA).

**Patch-clamp recordings**. IHCs from apical coils of freshly dissected organs of Corti were patch-clamped as described previously[19]. The pipette solution contained (in mM): for recordings with EGTA-mediated suppression of $Ca^{2+}$ influx at individual synapses ("EGTA"): (123 Cs-glutamate, 1 $MgCl_2$, 1 $CaCl_2$, 10 EGTA, 13 tetraethylammonium (TEA)-Cl, 20 HEPES, 2 Mg-ATP, 0.3 Na-GTP, 0.8 Fluo-8FF (AAT Bioquest), and the TAMRA-conjugated CtBP2/RIBEYE-binding dimer peptide (20 μM, Biosynthan, Berlin, Germany) (pH 7.3); for fluorescence fluctuation analysis ("FA"): 92 Cs-glutamate, 13 TEA-Cl, 20 CsOH-Hepes, 1 $MgCl_2$, 2 Mg-ATP, 0.3 Na-GTP, 10 EGTA, 10 Phosphocreatine-Na, 8 CsCl, and 1 Fluo-4FF (penta-$K^+$ salt; Invitrogen) (pH 7.2); for STED $Ca^{2+}$ imaging ("STED"): 130 Cs-gluconate, 10 TEA-Cl, 10 4-Aminopyridine (4-AP), 10 CsOH-HEPES, 1 $MgCl_2$, 2 Mg-ATP, 0.3 Na-GTP, as well as either 0.025 OGB-5N, 0.8 EGTA, and 0.4 BAPTA (for "physiological" buffering conditions) or 0.3 OGB-5N and 10 EGTA (for "intensified" conditions) (pH 7.2). The extracellular solution contained: for "EGTA": 102.2 NaCl, 2.8 KCl, 1 $MgCl_2$, 5 $CaCl_2$, 35 TEA-Cl, 10 HEPES; 2 g l$^{-1}$ glucose (pH 7.2); for "FA": 95 NaCl, 35 TEA-Cl, 2.8 KCl, 10 $CaCl_2$, 0.005 BayK8644, 1 $MgCl_2$, 1 CsCl, 10 NaOH-HEPES, and 10 D-glucose (pH 7.3); for "STED": 35 TEA-Cl, 2.8 KCl, 1 $MgCl_2$, 5 4-AP, 1 CsCl, 10 NaOH-HEPES, 10 D-glucose, as well as either 107.7 NaCl and 1.3 $CaCl_2$ ("physiological") or 99 NaCl and 10 $CaCl_2$ ("intensified") (pH 7.2). EPC-9 and -10 amplifiers controlled by Patchmaster or Pulse software (HEKA Elektronik, Lambrecht, Germany) were used for measurements. All voltages were corrected for liquid junction potentials (calculated). Currents were low-pass filtered at 5 kHz and sampled at 20 or 40 kHz, except for fluctuation analysis measurements, where currents were low-pass filtered at 8.5 kHz and sampled at 100 kHz. $Ca^{2+}$ currents were leak-corrected using a p/n protocol (except for fluorescence lifetime recordings). Cells were patched at a holding potential of −84 to −87 mV.

**Confocal $Ca^{2+}$ imaging for isolation of synaptic $Ca^{2+}$ current**. Experiments were performed with a custom-built spinning disk confocal microscope. A Zeiss Axio Examiner microscope (Carl Zeiss Microscopy GmbH, Göttingen, Germany) was equipped with a spinning disk scanner (CSU22, Yokogawa Electric Corporation, Tokyo, Japan), a scientific CMOS camera (Neo, Andor Technology, Belfast, UK), and a Zeiss 63× water immersion objective (1.0 NA) mounted on a fast piezo-electric focus drive (MIPOS 100 PL, Piezosystem Jena, Germany). The fluorescent $Ca^{2+}$ indicator and ribbon-labeling peptide were excited with a 491-nm diode-pump solid-state laser (Calypso, Cobolt AB, Solna, Sweden), and a 561 nm diode-pumped solid-state laser (Jive, Cobolt AB), respectively. Images were acquired at 5 ms/frame and processed with the software Andor Solis (Andor Technology). After

the formation of the ruptured-patch configuration, the $Ca^{2+}$ indicator was loaded into the cell for at least 4 min to reach a steady-state concentration. Data were analyzed using Igor Pro 6. Nine pre-depolarization frames were averaged to obtain the background fluorescence ($F_0$); for the calculation of $\Delta F/F_0$, the central pixel of a background-subtracted depolarization-evoked $Ca^{2+}$ hotspot and its eight neighbors were averaged and divided by the corresponding values from the $F_0$ frame.

Neighboring AZs found within ±1 μm focal distance to the targeted AZ were monitored for changes in fluorescence during application of EGTA (Supplementary Fig. 2a). We found that some neighboring AZs were slightly affected by the application of EGTA and quantified this by fitting a line to the $\Delta F/F_0$ data for the time of the recording during which the iontophoresis occurred and noting the final $\Delta F/F_0$ value at the end of the fit. The final value of the targeted AZ was then divided by the sum of all values (targeted as well as neighboring AZs) to obtain the fractional contribution of the targeted AZ ($r_{target}$) to the total change in $Ca^{2+}$ current ($\Delta I_{Ca}$), and the $Ca^{2+}$ current at the target synapse was obtained according to

$$\Delta I'_{Ca} = \Delta I_{Ca} \cdot r_{target}. \tag{1}$$

**Optical fluctuation analysis of synaptic $Ca^{2+}$ influx**. $Ca^{2+}$ imaging used a Fluoview 300 confocal scanner mounted on an upright microscope (BX50WI, Olympus, Tokyo, Japan) equipped with a 60× water immersion objective (0.9 NA) and a fiber-based (Picoquant) detection by a single-photon counting avalanche photo-diode (Perkin Elmer) read-out by custom hardware and software. The fluorescent $Ca^{2+}$ indicator and ribbon-labeling peptide were excited with a 50 mW, 488 nm solid-state laser (Cyan, Newport-Spectraphysics, Santa Clara, CA, USA) and a 1.5 mW, 543 nm He−Ne laser, respectively. Fluorescent hotspots were identified during 200 ms depolarizations to −7 mV in $xy$-scans at ≈10 Hz (using 0.5% of maximum laser intensity [488 nm]) and further characterized using spot detection ("point scan" mode of the confocal scanner, centered on the center of the fluorescent hotspot, using 0.05% of maximum laser intensity [488 nm]) with detection by the single-photon counting avalanche photo-diode at 2 kHz. To elicit $Ca^{2+}$-tail currents, cells were depolarized to +53 mV for 5 ms (opening $Ca^{2+}$ channels, but preventing $Ca^{2+}$ influx due to lack of driving force) and then repolarized to −67 mV. After recording fluorescence and whole-cell currents during ensembles of $K = 200$ tail currents, trial-to-trial variance was calculated as

$$\sigma^2(t) = \frac{1}{2(K-1)} \sum_{i=1}^{K-1} \left[ x_i(t) - x_{i+1}(t) \right]^2. \tag{2}$$

Fluorescence data was subsampled to 667 Hz to avoid the impact of correlation among neighboring data points[63]. After subtraction of baseline variance (shot noise and detector noise), variance of $Ca^{2+}$-indicator fluorescence was plotted against mean fluorescence and fitted with a parabolic function

$$\mathrm{var} = f_s \cdot S_{mean} - \frac{S_{mean}^2}{N}, \tag{3}$$

with $f_s$ being the contribution by a single open $Ca^{2+}$ channel (fluorescence increase or single-channel current), $S_{mean}$ being the average signal (mean fluorescence or current) of all trials at a single time point, and $N$ being the number of $Ca^{2+}$ channels at the synapse (fluorescence data) or in the entire cell (whole cell currents). Curve fitting used weighting according to the error-covariance of the variance (estimated generalized least squares)[63]. An equivalent analysis was performed on the whole-cell current to estimate the total number of $Ca^{2+}$ channels per IHC, as described in ref. [22].

**Modeling**. Diffusion and binding of $Ca^{2+}$ and buffers at a simulated AZ were modeled with CalC software[37] version 6.86 using the parameters given in Table 1. Data were analyzed using Igor Pro 6. Simulation of the effect of the microscope's PSF was done by convolving both the distribution of $Ca^{2+}$-bound and -free OGB-5N with Gaussian functions with FWHMs of 64 × 64 × 542 nm (STED) or 243 × 243 × 542 nm (confocal), corresponding to the PSFs of the microscope used in the experiments (Supplementary Fig. 16). The $Ca^{2+}$ concentration derived from the ratio of $Ca^{2+}$-bound and -free OGB-5N was calculated as

$$[Ca^{2+}] = K_D \cdot \frac{[OGB-5N]_{bound}}{[OGB-5N]_{free}}. \tag{4}$$

**STED $Ca^{2+}$ imaging**. Super-resolution $Ca^{2+}$ imaging was performed using a custom-built STED microscope[64]; the fluorophores were excited with a pulsed 488 nm diode laser (PicoTA, Toptica Photonics, Graefelfing, Germany) and de-excited using a pulsed 595 nm STED beam with a donut-shaped beam profile (Ti:Sapphire laser, Spectra-Physics, Darmstadt, Germany, which was frequency shifted using an optical parametric oscillator, APE, Berlin, Germany). Laser power in the back aperture of the objective lens ranged typically from 1.5 to 15 μW for the excitation and between 10 and 35 mW for the de-excitation, with STED pulse durations between 200 and 300 ps, depending on the power used. When acquiring super-

  

resolution images, the excitation laser intensity was typically increased to maintain comparable signal levels between confocal and STED.

In order to find and image depolarization-evoked synaptic $Ca^{2+}$ domains, we first needed to locate hotspots of $Ca^{2+}$ influx, then bring them into the correct focal plane, and finally image them before, during, and after depolarization. We found hotspots of $Ca^{2+}$ influx by focusing at the basal pole of the patched cell and depolarizing the cell briefly to −14 mV for 78 ms while acquiring $xy$-frames of 150 × 150 pixels (four frames, depolarization during the 2nd frame, 3 μm × 3 μm, 78 ms per frame) without activation of the STED laser. After identification of a depolarization-evoked $Ca^{2+}$ hotspot, an $xz$-scan (±1.5 μm in $z$, 100 nm step size) was performed through the center of the hotspot to determine the correct focal plane. IHCs were then depolarized 4× for 78 ms each (78 ms inter-depolarization interval) while acquiring frames of 150 × 75 pixels (16 frames, with the depolarization occurring during the 5th, 8th, 11th, and 14th frame, 3 μm × 1.5 μm, 78 ms per frame). Fluorescence was acquired with a Single Photon Avalanche Diode (SPAD detector, PDM series, Micro Photon Devices, Bolzano, Italy) using the TTL counting output of the detector. For the analysis of the spatial extent of the $Ca^{2+}$ domains in $xy$-scans, the detector signal was electronically time-gated by diverting the first 450 ps (equaling twice the fluorescence lifetime of unbound OGB-5N) of the signal into a separate detection channel, thereby reducing the contribution of the unbound OGB-5N signal by 80–90%, while only sacrificing 15–20% of the $Ca^{2+}$-bound OGB-5N signal. Images were acquired with Inspector software (Max-Planck-Innovation, Munich, Germany) and data were analyzed with Igor Pro 6 as follows: the frames acquired during depolarization were averaged and the average background signal was subtracted (calculated from 11 non-depolarization frames, omitting the first one to exclude possible timing, shutter, or vibrational disturbances). The size of the $Ca^{2+}$ domain was measured by fitting a 2D Gaussian function using a genetic fit algorithm[65] to the subset of the frame containing the $Ca^{2+}$ hotspot.

**Measurement of [$Ca^{2+}$] using fluorescence lifetime recordings.** For the analysis of $Ca^{2+}$ concentration at the synapse by fluorescence lifetime, we established TCSPC. Fluorescence was acquired using the NIM timing output of the SPAD detector with 35 ps timing resolution. The timing of the fluorescence photons was correlated with the excitation pulse at a TCSPC Module (SPC150N, Becker Hickl, Berlin, Germany) and sorted into a histogram of 25 ps time bins. The overall timing resolution of the setup was 110 ps FWHM, determined by measuring the instrument response function (IRF, see section "Fitting procedure and IRF" below and Supplementary Fig. 9a). In $xy$-scans (Fig. 8), $Ca^{2+}$ hotspots were scanned with an 8 × 8 pixel matrix in $xy$ in 50 nm steps, recording the fluorescence twice for each pixel: first for one 15 ms frame during a 19-ms depolarization, and then for two frames at rest (15 ms each), 330 ms after the initial depolarization. In $xz$-scans (Fig. 9b–d), hotspots were scanned with an 8 × 8 pixel matrix in $xz$ with 50 nm $x$- and 200 nm $z$-steps, again recording the fluorescence twice for each pixel: first for one 8 ms frame during a 12-ms depolarization, and then for two frames at rest (8 ms each), 176 ms after the initial depolarization.

To analyze the fluorescence decay, the IRF was iteratively reconvolved with a bi-exponential function and optimized with a fitting routine implemented in Matlab (Mathworks) (Supplementary Fig. 9). The fluorescence lifetimes of free and $Ca^{2+}$-bound OGB-5N ($\tau_{free}$ = 0.23 ns and $\tau_{bound}$ = 3.24 ns) were determined in saturated ([$Ca^{2+}$]=90 mM) and desaturated ([$Ca^{2+}$]=10 nM, using EGTA) $Ca^{2+}$ conditions (see section "Determination of the lifetime of $Ca^{2+}$-free and -bound OGB-5N" below) and kept fixed in the analysis of the $Ca^{2+}$ hotspots. When analyzing lifetime measurements taken with STED, the fitting routine ignored the first 300 ps of the fluorescence decay (during which the STED pulse was active) to avoid artifacts caused by the STED-evoked quenching of the signal. To ensure that this "blanking" of the first 300 ps did not significantly alter the results of the routine, we fitted fluorescence decay traces recorded in confocal mode using both the full data range and the blanked data (Supplementary Fig. 12). Both methods produced highly similar results: the differences in the number of photons assigned to the fast and slow lifetime channel were as small as 0.4%–2.8% for [$Ca^{2+}$] between 10 nM and 40 μM.

The analysis procedure of STED fluorescence lifetime data is summarized graphically in Fig. 7c–i. First, the ratio of free to $Ca^{2+}$-bound dye was obtained by first calculating the number of photons $F_i$ that were assigned to the fast and slow channels, respectively ($F_i = \alpha_i \times \tau_i$, with $\alpha_i$ the amplitudes from the fitting routine; for STED data, the amplitude extrapolated to the (blanked) onset of the fit was used). Then the photon count of the $Ca^{2+}$-bound dye was downscaled by a brightness factor of $b$ = 28.02 (see section "Estimation of $F_{max}$, $F_{min}$, and the brightness increase factor" below and Supplementary Fig. 10) in order to account for the increased fluorescence when the dye is bound to $Ca^{2+}$, thereby shifting the perspective from emitted photons per lifetime-channel to the actual ratio of fluorophores per channel. For STED imaging, an additional correction factor is needed to adjust for the lower STED efficiency (i.e., it requires a higher saturation intensity $I_{sat}$) of the quenched, $Ca^{2+}$-free dye with the short fluorescence lifetime. Less STED efficiency corresponds to a larger recorded focal volume, which would otherwise overestimate the $Ca^{2+}$-free component. Therefore, the STED efficiency correction factor depends on the STED beam power and was determined by measuring the fluorescence depletion of the STED beam at different powers for OGB-5N dye solutions that were saturated with $Ca^{2+}$ and $Ca^{2+}$-free (see section

"STED efficiency correction" below and Supplementary Fig. 11). The photon count of the $Ca^{2+}$-free dye was typically downscaled by $s$ = 0.491 or $s$ = 0.424 (depending on STED beam power). Further corrections were necessary for recordings taken in live IHCs. As opposed to in vitro calibrations, where the percentage of $Ca^{2+}$-bound/-free dye was measured to be <1% in a $Ca^{2+}$-depleted/-saturated environment, in live IHCs the long-lifetime component of the signal (i.e., the $Ca^{2+}$-bound dye) neither dropped below 2% for confocal and 4% for STED measurements (even in $Ca^{2+}$-depleted cells that were patched with 10 mM EGTA in the pipette) nor did it rise above 60% (even in $Ca^{2+}$-saturated cells patched with 60 mM $Ca^{2+}$ in the pipette; Supplementary Fig. 13). We devised two separate approaches to correct for this: one by rescaling the results to accommodate the diminished dynamic range of the $Ca^{2+}$ dye in cells, and one by adding a $Ca^{2+}$-unresponsive dye fraction to our model system and deriving modified equations for calculating the $Ca^{2+}$-bound/-free ratio of $Ca^{2+}$-dependent dye (see Supplementary Methods).

According to the first method, we rescaled the fraction of the photon counts assigned to the $Ca^{2+}$-bound and -free states of OGB-5N, respectively, by first calculating the corrected fraction $r_B$ of $Ca^{2+}$-bound and $r_U$ of $Ca^{2+}$-free OGB-5N as

$$r_B = \frac{F_B/b}{F_B/b + F_U \cdot s} \tag{5}$$

and

$$r_U = \frac{F_U \cdot s}{F_B/b + F_U \cdot s}, \tag{6}$$

with $F_B$ and $F_U$ the unmodified photon count assigned to the $Ca^{2+}$-bound and -free (unbound) dye, respectively, $b$ the brightness factor of 28.02 (describing the higher extinction coefficient and quantum yield of the $Ca^{2+}$-bound OGB-5N), and $s$ the STED correction factor of 1 (for confocal) or 0.424–0.491 (for STED, depending on STED beam intensity, describing the difference in effective focal volume for $Ca^{2+}$-bound and -free dye). We then further scaled these ratios according to the minimal and maximal fraction of $Ca^{2+}$-bound OGB-5N (Supplementary Fig. 13) as

$$r'_B = \frac{r_B - r_{min}}{R} \tag{7}$$

and

$$r'_U = \frac{r_U - (1 - r_{max})}{R}, \tag{8}$$

with $r_{min}(=r_{min,bound})$ and $r_{max}(=r_{max,bound})$ the minimum and maximum fractions of photons assigned to the $Ca^{2+}$-bound dye in $Ca^{2+}$-free and $Ca^{2+}$-saturated conditions, respectively, ($r_{min}$ = 0.02 or $r_{min}$ = 0.04 for confocal and STED, respectively, $r_{max}$ = 0.6), and $R = r_{max} - r_{min}$ the dynamic range of the dye.

The ratio of $Ca^{2+}$-bound to free OGB-5N was then calculated as

$$\frac{[OGB-5N_{bound}]}{[OGB-5N_{free}]} = \frac{r'_B}{r'_U}. \tag{9}$$

$Ca^{2+}$ concentrations for each pixel were calculated using

$$[Ca^{2+}] = K_{eff} \cdot \frac{[OGB-5N_{bound}]}{[OGB-5N_{free}]}, \tag{10}$$

with $K_{eff}$ = 195 μM being the effective dissociation constant ($K_D$) we had measured for OGB-5N in intracellular solution (Supplementary Fig. 14b).

Alternatively, if a $Ca^{2+}$-insensitive dye fraction is assumed to be present in IHCs, [$Ca^{2+}$] can be calculated according to a modified equation, which is less reliant on the more difficult to determine fast dye component (see below for a detailed description):

$$[Ca^{2+}] = K_{eff} \cdot \frac{F_{B,stim} - F_{B,rest}}{F_{U,rest} \cdot b \cdot s \cdot R - (F_{B,stim} - F_{B,rest})}. \tag{11}$$

Here, $F_{B,(stim/rest)}$ and $F_{U,(stim/rest)}$ are the observed photon counts (during depolarization [stim] or at rest), which were assigned by the fitting routine to the slow ($Ca^{2+}$-bound) and fast ($Ca^{2+}$-free) states, respectively. Again, $K_{eff}$ is the effective $K_D$ of OGB-5N in intracellular solution, $b$ is the brightness factor, $s$ the STED correction factor, and $R = r_{max} - r_{min}$ the dynamic range of the dye. Assuming a uniform [OGB-5N] and [$Ca^{2+}$] inside the cell in resting condition, $F_{U,rest}$ was taken as the average of the 8 × 8 pixel frame for $xy$-raster scans to minimize noise. For $xz$-scans, where $F_{U,rest}$ decreases with $z$, no averaging was performed and individual pixel values were used. Comparison of this method with the first method of scaled lifetime ratios showed good agreement between the results (Supplementary Fig. 15).

**Determination of the lifetime of Ca²⁺-free and -bound OGB-5N.** In order to determine the lifetime of OGB-5N in the Ca²⁺-free and Ca²⁺-bound states, we prepared diluted solutions containing OGB-5N, Ca²⁺ ions, and Ca²⁺ buffer (non-fluorescent Ca²⁺ chelators) with various free [Ca²⁺]. We also verified the lifetime values in solutions that matched the used intracellular solutions for patch-clamping as close as possible.

We recorded each solution at various excitation powers for durations in excess of 9000 ms, with pixel dwell times of 25 ps, for 1024 pixels in total, therefore spanning a time of 25.6 ns, although due to triggering requirements of the single-photon detection module the actual time during which signal was recorded amounted to 11.5 ns. The data was read into the Matlab fit routine, which fitted a single-exponential curve convolved with the IRF to the recorded data (Supplementary Fig. 9).

The fluorescence signal was fitted between bins 450 and 850, corresponding to 11.2–21.2 ns, hence spanning a 10 ns interval. The onset of fluorescence (corresponding also to the IRF) was at around 12.2 ns and the recorded signal was fitted from 1 ns before onset of fluorescence until 9 ns after. The threshold for considering a pixel was set to 18 counts, the "shift" parameter, denoting the shift between the IRF and the recorded fluorescence signal, was typically left floating, but was consistently determined to be ~20 ps ± 15 ps (so the duration of one bin).

The lifetime values for Ca²⁺-free and Ca²⁺-bound OGB-5N (Supplementary Fig. 9) were identical for both the diluted and the intracellular-like solutions, and resulted in values of

$$\tau_{\text{bound}} = 3.24 \text{ ns} \pm 0.03 \text{ ns},$$

$$\tau_{\text{free}} = 0.23 \text{ ns} \pm 0.03 \text{ ns}.$$

These lifetime values were used for the entirety of the bi-exponential fit procedures, where they were held constant, leaving only the two amplitude parameters floating freely.

**Fitting procedure and IRF.** When fitting a fluorescence decay curve recorded with STED, the fitting process needed to be modified to minimize the impact of STED de-excitation. The STED pulses were between 200 and 400 ps long (depending on the power, in typical recording conditions 200–300 ps) and started about 100 ps before the excitation pulse arrived. During the 300 ps in which the STED beam de-excites the fluorophores, a temporary, albeit steep, drop in apparent fluorescence lifetime is visible, which ceases after the end of the STED pulse. In order to avoid misinterpreting this additional short lifetime component as unbound OGB-5N, we omitted ("blanked") the first 300 ps from the fitting procedure and only included the subsequent bins in the fit (Supplementary Fig. 12), i.e., we fitted from 12.75 to 21.25 ns. Despite omitting a significant portion (~50%) of fluorescence from the unbound dye, the "blanked" fitting method proved to be very robust: when imaging calibration solutions in vitro, the blanked fit results from the STED recordings matched the values obtained from the confocal recordings to a very high degree, at least for unbound dye ratios of <40% (Supplementary Fig. 15b) and the omission resulted in only minor errors of 0.4–2.8% (Supplementary Fig. 12e). Additionally, we tested the blanking robustness by fitting experimental data from IHCs recorded without STED using both the full and the blanked data range, resulting in an average [Ca²⁺] error of 2–3% (Supplementary Fig. 12c, d).

One other parameter that needed to be handled differently when recording with STED was the shift between IRF and fluorescence data, which was (for confocal data sets) determined by the fit automatically. However, as the initial onset of fluorescence was omitted from our fitting range, the algorithm had difficulty determining the shift accurately. Therefore, the shift parameter was first automatically determined in the standard range (from bins 450–850), then locked down as a fixed parameter for the actual fit of the STED data with the aforementioned truncated boundaries.

**STED efficiency correction.** The Ca²⁺-indicator OGB-5N exists in two distinct states: Ca²⁺-bound or Ca²⁺-free, which display different STED efficiencies (or: saturation intensities). This is mainly due to the fact that the lifetime of the unbound state is extremely short, of the same order of magnitude as the STED pulse itself. Therefore, the STED beam has little time to effectively de-excite the dye before it spontaneously fluoresces. Different STED efficiencies correspond to different effective focal areas achieved using the same STED beam power. More STED-efficient dyes are switched off better and show a much smaller focal area and thus higher spatial resolution. Since the resolution along the optical axis is unchanged by the 2D-STED donut, this corresponds to a smaller "effective focal volume", meaning that in the same recording, we will be sampling a higher volume of Ca²⁺-free dye than of Ca²⁺-bound dye. For quantitative measurements, we need to compare the amount of Ca²⁺-bound to Ca²⁺-free dye, so we need to correct for any differences in sampling volume.

Assuming a homogenous dye distribution (on the scale of the focal area), we can say that the number of fluorophores in solution is proportional to the (sampled) volume. Assuming linear responses from each dye component, the number of recorded photons will depend on the number of fluorophores in sampled volume and in consequence on the volume and focal area.

To correct for different STED efficiencies, we measured OGB-5N dye solutions, either Ca²⁺-bound or Ca²⁺-free, for different STED powers, including zero STED power, i.e., a pure confocal recording. For each STED power, we counted the number of photons emitted from the dye (keeping the excitation power and dye concentration constant) and plotted the ratio of deactivated dye against the STED power. Note the much steeper descent of the Ca²⁺-bound dye as opposed to relatively gentle decline of the Ca²⁺-free dye (Supplementary Fig. 11e). In order to correct for different sample volumes for different STED powers, we calculated the ratio of the remaining activated fluorophores. With this factor, dubbed the "STED efficiency correction" $s$, we reduced the number of recorded photons in the fast (unbound) channel by the factor of effective focal volumes.

To validate this correction factor, we measured [Ca²⁺] using the lifetime method at different STED powers. Without applying the STED efficiency correction factor, we got discordant results. When correcting with the factor, however, the resulting ratio of bound dye matches the values for all the used STED powers, as well as for the confocal powers (i.e., zero STED power). This held true for the range of bound dye (<40%), as encountered in the IHC experiments. When confronted with higher fractions of bound dye, additional considerations during evaluation need to be taken to correctly discern the diminishing amount of photons in the fast (unbound) channel amidst the increasing number of photons emitted from the bound dye.

**Estimation of $F_{\text{max}}$, $F_{\text{min}}$, and the brightness increase factor.** The brightness factor describes how many more photons are emitted from the bound dye, as opposed to the free dye, for the same amount of incoming light. The different brightness depends both on the extinction coefficient, i.e., the number of photons that are absorbed from a certain amount of incoming light, as well as on the quantum yield, referring to the fraction of excited dye molecules that go on to emit a fluorescence photon instead of decaying non-radiatively. The extinction coefficient can be observed in fluorescence decay curves as different peak heights, whereas the quantum yield is proportional to the inverse lifetime of the dye and can be seen as longer lifetimes of the decay. The brightness factor can be calculated either by measuring the amount of photons directly, or by measuring the relative quantum yield and the relative extinction factors separately. For all the following measurements, we recorded the fluorescence decay of OGB-5N in solutions containing identical amounts of dye, which were either saturated with Ca²⁺ (>8 mM) or devoid of Ca²⁺ (~10 nM).

The difference in quantum yield between both states is comparatively easy to measure and can be calculated directly from the ratio of fluorescence lifetimes in the bound and free states (see Supplementary Fig. 9):

$$\frac{\tau_{\text{bound}}}{\tau_{\text{free}}} = \frac{3.24 \text{ ns}}{0.23 \text{ ns}} = 14.1. \quad (12)$$

The ratio of extinction factors proved to be more challenging to measure (as has been observed previously[36,40]), probably due in part to the difficulty of obtaining identical concentrations of Ca²⁺-bound and Ca²⁺-free OGB-5N, as well as slight fluctuations in excitation laser power. Using our fitting algorithm, we extracted a peak height ratio (i.e., amplitude ratio) of bound to unbound dye of

$$\frac{\alpha_{\text{bound}}}{\alpha_{\text{free}}} = 1.99 \pm 0.12. \quad (13)$$

Combined with the above lifetime ratio, this results in a brightness ratio of

$$b = \frac{\sigma_{\text{B}}}{\sigma_{\text{U}}} = \frac{\alpha_{\text{B}}\tau_{\text{B}}}{\alpha_{\text{U}}\tau_{\text{U}}} = 28.03. \quad (14)$$

We additionally measured the brightness ratio directly, by recording fluorescence decay curves for Ca²⁺-saturated and -"free" solutions at different excitation powers and directly comparing the recorded photons. This was repeated over and over, so as to be able to exclude strongly deviating measurements and resulted in a brightness factor of (Supplementary Fig. 10)

$$b = \frac{\sigma_{\text{B}}}{\sigma_{\text{U}}} = 28 \pm 3. \quad (15)$$

Both methods produced comparable results, albeit with a notable error. It is possible that the brightness factor in an ex vivo environment would deviate from these values, due to different dye behavior. Due to the difficulty of the involved experiments, this could not be measured directly, however.

**In vitro determination of $K_{\text{eff}}$, the effective $K_D$ of OGB-5N.** The measured dissociation constant $K_D$ of a dye may be influenced by other Ca²⁺-binding factors, such as those present in the intracellular solution used in our patch-clamp experiments. Therefore, for our purposes, we refer to our measured $K_D$ as $K_{\text{eff}}$ to indicate that this does not represent the behavior of the pure dye. In order to calculate the effective dissociation constant $K_{\text{eff}}$ of the dye OGB-5N in an

environment as close to the cellular surrounding as possible, we prepared solutions with different values of [Ca$^{2+}$], ranging from nearly Ca$^{2+}$-free (10 nM) to Ca$^{2+}$-saturated (80 mM), with the highest number of different solutions near to the expected $K_D$ values between 20 and 200 μM. The solutions matched the intracellular solutions used for patch-clamping exactly, except for the amount of added Ca$^{2+}$ and buffer. For each solution, we recorded various fluorescence decay curves, both for various excitation powers as well as for different STED powers. Free [Ca$^{2+}$] was calculated using the software Patcher's Power Tools in Igor pro.

We determined the $K_{eff}$ by plotting the photon counts $F_{slow}$ in the slow channel (as determined by the fitting routine) against the calculated free [Ca$^{2+}$] in the solution (Supplementary Fig. 14b). This curve was then fitted with

$$F_{slow} = F_{slow,max} \cdot \frac{[Ca^{2+}]}{[Ca^{2+}] + K_{eff}}, \tag{16}$$

With $F_{slow,max}$ the maximum number of counts recorded in the slow channel. The resulting values of $K_{eff} = 194.3 \pm 23.7$ μM or $K_{eff} = 195.7 \pm 24.8$ μM for confocal and STED data, respectively, were about six-fold higher than the $K_D$ we measured in diluted solution (Supplementary Fig. 13a) and as published in literature[36].

**Statistical analysis and presentation**. Significance testing was performed using two-tailed tests, either Student's *T*-test or the Mann–Whitney *U* test, as indicated. Statistical power was calculated post hoc using the software G*power[66] with an $\alpha$ error probability of 0.05. When calculating the power for Mann–Whitney *U* tests, normal distributions were assumed and the asymptotic relative efficiency method was used. No statistical method was used to estimate sample size and no randomization or blinding was performed. Where box plots are presented, boxes indicate 25–75 percentile, whiskers 10–90, and the medians are indicated by the horizontal lines. Individual data points are marked by crosses. Averages are presented as ± standard error of the mean (SEM).

**Data availability**. The data supporting the findings of this study are available from the corresponding authors upon request. The CALC scripts of the Ca$^{2+}$ reaction/diffusion modeling at IHC AZs can be obtained from www.innerearlab.uni-goettingen.de/materials.html. The Igor Pro and Matlab code is available from the corresponding authors upon request.

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

## Acknowledgements

This work was supported by grants of the German Research Foundation: Collaborative Research Center 889 (to project B7 to K.I.W. and project A2 to T.M.), DFG Research Center and Cluster of Excellence, Center for Nanoscale Microscopy and Molecular Physiology of the Brain (FZT-103) to K.I.W., S.W.H., and T.M., and the Leibniz program (to T.M.) as well as by the German Federal Ministry of Education and Research (Bernstein Focus for Neurotechnology 01GQ0810 to T.M. and Alexander Egner and Bernstein Center for Computational Neuroscience 01GQ1005A to T.M.). T.L.O. was supported by a fellowship of the DAAD. We thank Drs. William Roberts and Murat Alp for sharing information on Ca2+-indicator properties. We thank Drs. David DiGregorio, Valentin Nägerl, Alexander Egner, Benjamin Harke, Mark Rutherford, and Tina Pangršič for participation in the early phase of experiments, Dr. Guiseppe Vicidomini for providing the code for the lifetime fitting routine, Drs. Gerald Donnert and Christian Wurm of Abberior Instruments for providing access to a STED microscope for morphological analysis, and C. Senger-Freitag and S. Gerke for expert technical assistance. We thank Eckart D. Gundelfinger for providing bassoon mutant mice. We are very grateful to Erwin Neher, Silvio Rizzoli, and Christian Vogl for critical discussion of the data and the manuscript.

## Author contributions

J.N., N.T.U., T.-L.O., T.F., P.J., S.W.H., K.I.W. and T.M. designed the experiments. J.N. performed immunohistochemistry, and J.N., N.T.U., T.-L.O., T.F. and P.J. performed electrophysiology and Ca2+ imaging. STED and lifetime Ca2+ imaging was performed in the Department of NanoBiophotonics at the Max Planck Institute for Biophysical Chemistry by N.T.U. and J.N. J.N. set up the AZ model. J.N., N.T.U., T.-L.O., T.F., P.J., K.I.W. and T.M. analyzed the data. J.N., N.T.U., T.L.O. and T.M. wrote the manuscript.

## Additional information

**Competing interests:** S.W.H. owns shares of the company Abberior Instruments GmbH that built one of the STED microscopes used in this study. The remaining authors declare no competing financial interests.

