## [Peer Review File · Nature Communications]

Editorial Note: The figure on page 32 in this Peer Review File is reproduced with permission from Elsevier.

Reviewers' comments:

Reviewer #1 (Remarks to the Author):

This study examines the properties of calcium channel clusters at ribbon synapses in hair cells. The authors use a combination of confocal, STED, electrophysiology and modelling to estimate the size of clusters, the number of calcium channels they contain and the cytoplasmic $[Ca^{2+}]$ that drives transmitter release. The novel measurements, particularly using STED lifetime imaging to monitor presynaptic calcium, provide a nice experimental verification of previous, less direct assays of the $[Ca^{2+}]$ within the nanodomain. So while there are no really new biological insights, apart from the result in the bassoon knockout, the strength of this study is in testing existing models with new experimental approaches. However, there are a number of technical issues that need to be addressed and improvements to the presentation that are required before it is suitable for publication.

Scientific points

- 1) What was the criterion for deciding the groups in figure 1j?
- 2) Was deconvolution used to estimate the true size of the clusters using 3D STED?
- 3) The major assumption underlying the first estimate of the number of calcium channels present is that local iontophoresis of EGTA only affects the Ca channels located in a cluster. The authors control for effects on other clusters by imaging their fluorescence, which is nice, but the impact on the neighbors needs to be quantified for the 19 IHC measurements. As the authors point out later in the manuscript this approach could overestimate the number of channels in a cluster if calcium channels are present in the extrasynaptic membrane or in unresolved IHCs. Again the potential error that this introduces should be quantified, particularly as they come up with an estimated density of extrasynaptic channels. On a related point, on what basis were the two outliers excluded?
- 4) Please quantify the degree of correlation in Figure 2d with a statistical test.
- 5) Would deconvolution of the 3D PSF provide a more accurate estimate of the cluster sizes? Also it would be good to quantify the CV of the cluster lengths/areas and relate these to the CV of the estimated channel number.
- 6) While I understand that the STED approach is prone to underestimate the unbound dye contribution for OGB-5N, due to the fast life time, I don't think this explains the better SNR for the 'near-physiological' solution than the intensified solution. Rather, I think it much more likely that the high $[OGB-5N]$ in the latter solution rapidly buffered Ca^{2+} close to the site of influx, due to its fast on-rate, reducing the size of the change in $[Ca^{2+}]$.
- 7) How dependent was the estimate of unbound dye on the blanking window used? The decay due to STED and the lifetime of the unbound dye look almost identical. Can they really be distinguished?
- 8) I am unclear on the justification for applying a Ca-insensitive contribution to the fast and slow channels?
- 9) It seems like 3D STED would be much better suited than 2D STED for measuring $[Ca^{2+}]$, why wasn't this approach used as for the anatomy?
- 10) Fixed buffers do not seem to have been included in the simulations. These can have strong effects on the predicted $[Ca^{2+}]$ in the nanodomain, particularly if they are low affinity. On a related note, how dependent was the model based prediction of $[Ca^{2+}]$ on equilibrium assumptions?
- 11) What accounts for the discrepancy in the estimate of the K_d for OGB-5N and previous studies?

Stylistic points

- 12) The title does not convey much about the findings.
- 13) The delta on the α_2 subunit symbol is not in formatted properly (not a Greek symbol).

14) The introduction rather skirts around the key findings of previous work, rather than stating them clearly, which I found frustrating. I think the authors need to be a more candid about what is known about the properties of Ca channel clusters and the current models at both ribbon and central synapses. Also it sets up the necessity of measuring both the properties of Ca channel clusters and vesicles (which has not been possible), but this is not really achieved as the release sites are only assayed with bassoon labelling as opposed to identifying the location of individual vesicles. In particular the papers of Holderith et al 2012, Nakamura et al., 2015 and Keller et al., 2015 warrant more discussion given their conclusions. In addition, a more focused discussion on the parallels between findings at non-ribbon and ribbon synapses (e.g. clustering of Ca channels with highly variable numbers, nanodomain coupling (10-20 nm), predicted $[Ca^{2+}]$ in the tens of micromolar and vesicles arranged at the perimeter of the channel cluster) in the context of the results presented in the paper would significantly strengthen the paper.

Reviewer #2 (Remarks to the Author):

Neef and colleagues studied Ca dynamics at active zones (AZs) of inner hair cells (IHCs) of the mouse cochlea. They used 2D or 3D STED microscopy to visualise immunohistochemically labelled voltage-gated Ca channels (Cav1.3) and bassoon. They found different morphological classes as well as strongly varying sizes of AZs. Using confocal spot-detection measurements of depolarisation-evoked Ca signals at putative AZs they estimated the number of Ca channels per AZ. Comparing spot-detection measurements with STED-based Ca imaging, in which Ca was quantified by time-correlated photon-counting lifetime imaging (FLIM), they analysed the Ca concentration within the STED point-spread function (SPF) at the AZ. Finally they incorporated the measured data into a model simulation of AZ Ca dynamics and estimated the Ca concentration at the AZ if the optical resolution in the z axis could have been better than 400-600 nm.

The authors used an impressive array of optical techniques to address the important and timely question of the nature of the Ca signal that drives release at AZs. However, the study falls short in addressing this question in a clear, unambiguous, convincing and quantitative way. Instead the manuscript makes the impression of a collection of different findings and techniques that were put together under the umbrella of "AZ nanophysiology", without providing clear advances to the field and, in several instances, without solving the issue of unexpected and/or conflicting findings.

1) In the first part of the study the authors used 2D STED imaging to describe different morphological classes of AZs (Fig. 1a-j) and the average lengths and widths of AZs (Fig. 1k). They then switch to 3D STED (Fig. 2) and note that the length estimates of AZs increases if the 3D curvature is considered (Fig. 2d). Correctly, they assign this discrepancy to the obvious problem of taking 2D images of a curved structures oriented at different angles in space. Surprisingly, however, the authors do not go back to their 2D-based classification of AZs and reanalyse the putative classes of AZs in 3D images. This leaves the reader with a 2D-based, "new classification" of AZs although, for example, "point-shaped AZs" could well represent "single-line AZs" imaged from the side, "single lines" may be "complex clusters" imaged from the side and so on. The boomerang-shaped AZ shown in Fig.2a'" represents a class not even identified in 2D images. Taken together, the first part of STED-based immunohistochemistry did not provide convincing results other than estimates of the lengths of AZs, information well known from EM data.

2) The authors went on by addressing the questions of how many Ca channels are present per AZ. They used two different approaches: local suppression of Ca influx at single AZs and optical fluctuation analysis. Using these two methods they arrive at significantly divergent numbers of 144 and 74 channels per AZ, respectively. They conclude that the numbers probably represent over- or underestimates of the true values but did not try to dig deeper by either choosing a third method for counting channels or by analysing potential biases in the two methods. (For example, the significant age differences of the mice used in the two approaches [approx. P16 and P30,

respectively] may explain part of the observed differences.)

3) Finally the authors developed STED-based Ca imaging in order to describe depolarization-induced Ca signal at AZs at a better-than-confocal resolution. This section starts rather ambiguous: The authors state that they used two-different solutions ("intensified" vs. "physiological") for recording Ca signals (p. 12). One page later, however, they state to have focussed on the "physiological" solution. One page of the text could have been spared by streamlining the study. More importantly, the authors fell short in fully establishing STED-based Ca imaging: the pitfalls of this technique should have been addressed more thoroughly by discussing a) the exact dimensions of the PSFs for apo- and Ca-bound dye moieties, b) showing calibrations that validate the diverse correction factors deduced by the authors, c) discussing how diffusion of Ca, either free or buffered by endogenous and exogenous buffers, affects the PSF and, finally, d) whether increasing the lateral resolution via STED can be expected to yield more accurate estimates of Ca at the AZ than confocal imaging; given the size of AZs and the similar z resolution of both techniques, the difference can be expected and indeed were found, to be rather small (p. 17 and Fig. 6e,f). In summary, the section on STED-based Ca imaging remained descriptive without going sufficiently deep into photophysical problems of the technique and did not yield new experimental insight into AZ Ca dynamics.

4) Similarly, the establishment of STED-based FLIM remained rather superficial.

a) Lifetimes (LTs) were described to be monoexponential although the data and the corresponding fits shown that they are not (Fig. S5b). Here the authors did not adhere to the standards in the field where data are shown as log-lin plots with residuals and χ^2 values indicating the goodness of fits.

b) Blanking the part of the lifetime curve distorted by the STED pulse may simplify the analysis of LT curves, however, the authors then need to state how they normalized the fractions of fast and slow LT components: to the extrapolated peak of the LT curve within the blanked period or to the first data point after the blanking period?

c) The authors failed to explain why there is no fast, STED-induced LT component in Fig. S4c, red dotted data curve.

d) Similarly, they failed to convincingly explain the finding that the K_d of OGB5N in pipette solutions is 5 times higher than in commercial buffer solutions. P. 16 leaves many open questions on how the calibration was done.

e) Labelling time axes with "a.u." in S4b is inappropriate for understanding the increases in fluorescence induced by the STED-Laser for some of the dyes.

5) The data on Bassoon-mutant AZs are insufficient to draw strong conclusions because the number of observations is quite small (cf. Fig. 5: $n=9$ for BSN-mutants, $n>50$ for wild-type mice). More importantly, the data are neither connected to the topic of the manuscript nor are they sufficiently worked out to represent a "bassoon story" on their own.

Additional comments:

6) Throughout the manuscript exact information about the PSFs of the different imaging techniques (and how PSFs were measured) is missing.

7) Why would a holding potential of +53 mV not permit Ca influx into IHCs (p. 9, line 212)?

8) For statistical tests, p and power values should be provided.

9) The schemes illustrated in Fig. 7 are based on published EM work, not on data from the present study.

Reviewer #3 (Remarks to the Author):

In this report, Neef and colleagues characterize the features of Ca^{2+} signaling at the active zones

(AZs) of cochlear inner hair cells (IHCs). They make extremely careful functional measurements using a truly awesome array of optical approaches including new super-resolution techniques (e.g., STED-based fluorescence lifetime measurements) to quantify, beyond all previous attempts, details pertaining to Ca²⁺ channel organization, number, and Ca²⁺ domain signaling. Because of the widespread interest in presynaptic mechanisms regulating neurotransmission, this report will provide a wealth of knowledge to a broad readership and will be a common reference for advanced optical approaches to study the function of Ca²⁺ channels and their organization at presynaptic elements.

I have no major concerns with the paper and only have a couple of points that the authors should consider.

1) Their use of selective suppression of Ca²⁺ influx at an individual AZ to estimate Ca²⁺ channel number is novel approach and seems well-controlled regarding single AZ selectivity (i.e., lack of effect on nearby AZs). However, the EGTA-induced reduction of the whole-cell current seems large given that it should be occurring at a single AZ (at least from the example in 3c), especially considering that there are >10 AZs per IHC. They mention that this could be due to selection bias (picking AZs with strong Ca²⁺ signaling) but is it realistic to believe that a few AZs make contain a majority of Ca²⁺ channels in an IHC especially if 10-30% are extrasynaptic (maybe they are blocking a significant portion of this population as well)? It might be helpful if they reported the average effective block of EGTA on the whole-cell tail current in this experiment (the current lost by EGTA application) rather than the current size of a single AZ alone. Also, they don't report the percent block of the Ca²⁺ response at the targeted AZ (again, their example makes it look sub-maximal). Could they use "intensified" conditions (i.e., higher external [Ca²⁺]) to pull signal out from more weakly responding AZs? How might a selection bias effect the calibration procedure against the remainder of the AZs examined?

2) It is surprising that the authors did not estimate Ca²⁺ channel numbers in their Bassoon mutant. They rely on the mutant for comparison of structural details of Ca²⁺ channels organization but little in the way of functional details (apart from the width of a Ca²⁺ hotspot in 5e). This is not only pertinent (they previously published reduced Ca²⁺ channel number following loss of Bassoon [see Frank et al. 2010]) but also would help validate their techniques (e.g., the expectation is that average number of Ca²⁺ channels per active zone will decrease and there will be an accompanying increase in the extrasynaptic population).

Reviewer #4 (Remarks to the Author):

This review is mostly interested in the large number of delicate optical techniques that the authors use to achieve this study. In this manuscript, two dimensional and three dimensional super-resolution STED imaging and fluorescence lifetime measurements were performed to study the morphology and the calcium concentration in the vicinity of the active zones of synapses of sensory hair cells. Super-resolution STED imaging appeal careful characterizations that the authors rigorously manage and expose in a quite clear manner.

I have a few comments and questions:

- The manuscript first present morphological images using a STED system lacking super-resolution along the longitudinal axis before using a other system achieving isotropic super-resolution. The purpose of this first study is unclear to us. If the goal of the first part is to conclude that biological results are irrelevant with 2D STED, I would suggest placing these inaccurate data in supplementary materials. Moreover, the introduction stresses on the results about Ca imaging but do not motivate the need for a morphological study.

- l. 142: Why do some antibodies give higher resolution than others? Is it about the antibody (as stated) or the dye (as given in parenthesis)?

- l. 184: The Ca imaging is described in detail but a short description of the electrophysiology/stimulation is missing before mentioning a "depolarization-evoked fluorescence".

- l. 194: the pA/a.u. unit is irrelevant and useless. Furthermore $\Delta F/F$ is unitless and not an arbitrary unit. Same remark for labels of scale bars in Fig. 3b,c.

- l.291: The argument in the sentence is unclear to me: "By searching at the basal pole of the IHCs we favored synapses that seemed to lay en face to the image plane"

- The stripe-like arrangement of channels at synapses was already observed with STORM imaging for instance and is not something so surprising. A citation to Dani et al. *Neuron Neurotechniques* 68 , 843–856, (2010) may here make sense at least from the instrumental point of view. At lines 294-297, it is thus valuable to observe this structure under live-conditions but the conclusion the authors draw is weak. Moreover, the reference to a possible "artifact in immunochemistry" should be made explicit if the authors wish to keep this conclusion. Finally, a single example of STED image of OGB is given in Fig. 5a' in which the elongated structure is not so clear as in Fig. 1 and 2: it would be appreciated by the reader to have other illustrations in a supplementary figures.

- l. 299 - 301. The authors deduce the intrinsic size of structures from the STED-power independence of the imaged structures. We feel that this is not a foregone conclusion since other imaging artifacts, especially regarding the dye photo-dynamic, could result in the same power-independent behaviour.

- l. 306: "re-excitation by the STED beam" is jargon and should be rephrased and explained.

- l. 325: The authors dismiss earlier results in Ref 23. This needs more explanation especially since the senior author is the same in the present manuscript as in Ref 23.

-l. 349: What is a "highly monoexponential decay"?

-l. 514: It is not clear why do the pipet solutions contain Calcium in addition to the Calcium Chelator. Wouldn't it be more efficient omitting the Ca?

- In Fig. 5b-c, the presented cluster sizes seem to significantly differ from the results presented in Fig. 1 and Fig. 2. We understand that morphological images in Fig. 1 and 2 were carried with a better dye but the discrepancy in the results should at least be pointed out and discussed in the manuscript.

- Fluorescence lifetime measurements with STED were performed after 300ps to wait for depletion by the STED beam while the fluorescence lifetime of the free OGB is only 230ps. This means that data fittings are performed on a very small fraction of fluorophores. This fraction is not clearly given and in my opinion, the quick component of the double exponential fitting in Fig. S5e does not work. Is this remark somehow related to the "additional correction factor" mentioned at line 627? These latter sentences were unclear to me.

- Fig. S7, the inefficient STED effect on free OGB is solely attributed to the fluorescence life-time. What about the stimulated emission cross-section?

- In the discussion, the limit of the performance of STED calcium imaging in the light of optical effects such as resolution and the influence of the binding states of the OGB (which is nicely discussed). However, the effect of diffusion of ions (extremely fast at the scale of a STED PSF or

even a confocal PSF) should be discussed. A detailed and quantitative discussion and comparison of diffusion to imaging speed is missing.

From the optical techniques point of view, I recommend the manuscript for publication after addressing these remarks.

Reviewers' comments:

Reviewer #1 (Remarks to the Author):

This study examines the properties of calcium channel clusters at ribbon synapses in hair cells. The authors use a combination of confocal, STED, electrophysiology and modelling to estimate the size of clusters, the number of calcium channels they contain and the cytoplasmic [Ca²⁺] that drives transmitter release. The novel measurements, particularly using STED lifetime imaging to monitor presynaptic calcium, provide a nice experimental verification of previous, less direct assays of the [Ca²⁺] within the nanodomain. So while there are no really new biological insights, apart from the result in the bassoon knockout, the strength of this study is in testing existing models with new experimental approaches. However, there are a number of technical issues that need to be addressed and improvements to the presentation that are required before it is suitable for publication.

We thank the reviewer for the appreciation of our work and the valuable comments that helped us to further improve our MS. Below, please find our point-by-point responses.

Scientific points

1) What was the criterion for deciding the groups in figure 1j?

This was a subjective decision; we based it solely on our impression of the data. While not further defined by parameters, we find the categories to work quite robustly also for several observers and projects on Ca²⁺-channel clusters (unpublished) at least within our lab. This is now clearly stated in the MS.

“We introduced a subjective classification of the shapes of the Ca²⁺-channel- and bassoon-clusters that we found to arrive at comparable results when performed by different observers.”

2) Was deconvolution used to estimate the true size of the clusters using 3D STED?

No, this was based on raw data. We would like to avoid deconvolution since using it to estimate the true sizes underlying the acquired image is very unreliable – we would be able to reach arbitrarily small sizes, depending on the settings chosen for deconvolution and the number of iterations. Unfortunately, it is not possible to say which settings would return the “true” size of the measured object. Therefore, we chose to base our measurements on raw data and thus return an upper estimate of the Ca²⁺-channel cluster sizes. This is now clearly stated in the MS.

“Quantification was done on the raw images and hence reported upper estimates but avoided the dependence on assumptions inherent to deconvolution.”

3) The major assumption underlying the first estimate of the number of calcium channels present is that local iontophoresis of EGTA only affects the Ca channels located in a cluster. The authors control for effects on other clusters by imaging their fluorescence, which is nice, but the impact on the

neighbors needs to be quantified for the 19 IHC measurements. As the authors point out later in the manuscript this approach could overestimate the number of channels in a cluster if calcium channels are present in the extrasynaptic membrane or in unresolved IHCs. Again the potential error that this introduces should be quantified, particularly as they come up with an estimated density of extrasynaptic channels. On a related point, on what basis were the two outliers excluded?

Done. We quantified the decrease in F_{Ca} of the neighboring synapses and found that the summed (total) signal change upon iontophoretic application of EGTA in neighboring AZs was on average $12 \pm 4\%$ of that of the targeted AZ and state this in the text. We corrected this for the data set acquired in the absence of BayK8644. This correction caused a smaller estimate for the total number of Ca^{2+} -channels at the IHC synapse which is now closer to our second estimate obtained from optical fluctuation analysis.

“We calculated the influence of the observed changes in neighboring AZs on the whole cell calcium currents (Figure S1a, Supplementary Methods) and corrected for this when estimating Ca^{2+} -influx in the synapse under study. On average, the total signal change in neighboring AZs was $12 \pm 4\%$ of that of the targeted AZ.”

Unfortunately, we are unable to properly quantify the effect of additional blocking of extrasynaptic Ca^{2+} -channels on the change in measured Ca^{2+} -current, since we can't measure the spread of the iontophoretically applied EGTA. Therefore, we also can't estimate the membrane surface that is depleted of extracellular Ca^{2+} . Additionally, we do not know whether the extrasynaptic channel density is the same all over the hair cell or if it is different near AZs. However, if we assume the lateral spread of EGTA to be maximally $2 \mu m$, we would estimate that a membrane surface of up to $12.5 \mu m^2$ would be deprived of Ca^{2+} . Assuming that extrasynaptic Calcium channels can only be found in the basolateral membrane of IHCs ($\sim 550 \mu m^2$, see ref. 26, Brandt et al., 2005, for details) and assuming that 30% of the ~ 1900 channels are extrasynaptic, we would arrive at a density of ~ 1 extrasynaptic channel per μm^2 . This would indicate that, using the method of iontophoretic application of EGTA, we overestimate the number of synaptic Calcium channels by $\sim 10\%$. However, it also appears quite possible that the number of extrasynaptic Calcium channels is higher near the AZs. We now discuss this issue more explicitly in the MS:

“In summary, both experimental approaches reported somewhat different estimates for the mean number of Ca^{2+} -channels per AZ (125 and 78), suggesting either an overestimation of the number of channels after application of EGTA (e.g. due to additional effect on extrasynaptic Ca^{2+} -channels that is difficult if not impossible to quantify from our experiments), or an underestimation in the optical fluctuation analysis experiments (e.g. due to activity of synaptic channels outside the microscope's PSF), or a combination of both. Nevertheless, the observation of a large range of Ca^{2+} -channels per AZ by both approaches confirms that the previously reported heterogeneity of Ca^{2+} -signaling among the AZs of a given IHC¹⁵ results from different numbers of Ca^{2+} -channels per AZ. Indeed, the coefficients of variation found for Ca^{2+} -channel number per AZ by both approaches (0.5 and 0.64) were comparable to that previously reported for the maximal synaptic Ca^{2+} -influx (0.65, ref. 15) and to that which we found for the integrals of the 2D Gaussian fits to $Ca_v1.3$ immunofluorescence of Ca^{2+} -channel clusters (0.67, data from Fig. 1). This AZ heterogeneity might be related to functional diversity of the spiking behavior of postsynaptic spiral ganglion neurons^{15,44}. Finally, we can determine the total number of synaptic Ca^{2+} -channels by multiplying the average number of Ca^{2+} -channels per AZ with the count of synapses per IHC (typically 12 in the area of the organ of Corti

used in our measurements), arriving at a value between 950–1500. Relating this to the total number of Ca²⁺-channels per IHC (~1900, see above) indicates that approximately 20%–50% of all channels localize extrasynaptically.”

The outliers were excluded based on them being greater than the average+4 x SD (average before exclusion of 129 + 4 x SD of 64 = 385). This is now stated in the MS, page 9/10:

“We observed two outliers (greater than average + 4x standard deviation) featuring 431 and 555 Ca²⁺-channels that we assume to be multi-ribbon AZs or unresolved neighboring synapse pairs and that we excluded from the dataset.”

4) Please quantify the degree of correlation in Figure 2d with a statistical test.

The correlation coefficient for the entire data presented in Figure 2d is r=0.71, for the 2D data: r=0.58, for 3D data: 0.34. We have added the following sentence to page 6/7 of the MS:

“When relating the higher resolution STED-estimate of the length for Ca²⁺-channel- and bassoon-clusters with the lower resolution STED-estimate (Abberior STAR580-labelled secondary antibody) of the length of their corresponding bassoon- and Ca²⁺-channel-clusters, respectively, we found a high degree of correlation (Figure 2d; Pearson’s correlation coefficient of 0.71 for the entire data set (including 3D-STED, see below), 0.58 for the 2D-data). Together with the identical length estimate, this indicates that Ca²⁺-channel-clusters and presynaptic density (marked by bassoon) scale with each other.”

5) Would deconvolution of the 3D PSF provide a more accurate estimate of the cluster sizes? Also it would be good to quantify the CV of the cluster lengths/areas and relate these to the CV of the estimated channel number.

As explained above (response to comment #2) we would like to refrain from using deconvolution to estimate the cluster sizes, since there is no possibility to figure out when the decrease of the size caused by the consecutive iterations of the deconvolution algorithm has reached the true size of the measured object and when the iterations have sharpened the image so much that the object actually appears smaller than it is in reality. We have contacted experts in the field (working in the microscopy lab of Stefan Hell, Göttingen, and at Abberior Instruments) who have also advised us to avoid using deconvolution on images used to quantify sizes.

We have added information about the CV of the cluster sizes into the MS, page 12 as follows:

“Indeed, the coefficients of variation found for Ca²⁺-channel number per AZ by both approaches (0.5 and 0.64) were comparable to that previously reported for the maximal synaptic Ca²⁺-influx (0.65, ref. 15) and to that which we found for the integrals of the 2D Gaussian fits to Ca_v1.3 immunofluorescence of Ca²⁺-channel clusters (0.67, data from Fig. 1).”

6) While I understand that the STED approach is prone to underestimate the unbound dye contribution for OGB-5N, due to the fast life time, I don’t think this explains the better SNR for the ‘near-physiological’ solution than the intensified solution. Rather, I think it much more likely that the high

[OGB-5N] in the latter solution rapidly buffered Ca^{2+} close to the site of influx, due to its fast on-rate, reducing the size of the change in $[\text{Ca}^{2+}]$.

We are not entirely sure whether we follow the reviewer's argument. If we get it right the reviewer expects the hotspot of OGB-5N fluorescence to be weaker/smaller under the intensified conditions, because of rapid buffering of incoming Ca^{2+} by the Ca^{2+} -indicator. More Ca^{2+} -bound OGB-5N, however, should in turn lead to higher fluorescence, considering the significantly higher brightness of Ca^{2+} -bound OGB-5N. Instead, we suspect that the higher [OGB-5N] led to more background signal from the unbound dye and also facilitated the spread the Ca^{2+} -signal, a notion which we support by CalC modeling. Therefore, we consider the STED recordings in conditions where [OGB-5N] is more similar to the expected $[\text{Ca}^{2+}]$ most interesting and, when revising the MS, put most emphasis on these recordings. We now introduce the experiments with "intensified conditions" only after the discussion of the bassoon data, on page 15/16, where the higher [OGB-5N] aids the comparison::

„We further addressed this question in additional experiments using conditions that enhanced Ca^{2+} -influx (10 mM $[\text{Ca}^{2+}]_e$) and also involved higher OGB-5N (300 μM) and EGTA (10 mM; "intensified conditions"; Figure 5f). We still observed the above described wide-spread Ca^{2+} -signals in bassoon-deficient IHCs in confocal and STED Ca^{2+} -imaging. Interestingly, we found larger Ca^{2+} -hotspots for wildtype synapses in the "intensified conditions", where [OGB-5N] was much greater than the expected presynaptic $[\text{Ca}^{2+}]$, than described above for more physiological conditions, where [OGB-5N] was more similar to $[\text{Ca}^{2+}]$ at the synapse.“

7) How dependent was the estimate of unbound dye on the blanking window used? The decay due to STED and the lifetime of the unbound dye look almost identical. Can they really be distinguished?

This an important concern and we have undertaken substantial efforts to further test the validity of our analysis. We conclude that despite the fact that the time course of stimulated fluorophore depletion is not much faster than the fast lifetime of OGB-5N, this component can still be estimated with good reliability. For one, the effect of the STED beam ends abruptly with the cessation of STED pulse, whereas the fluorescence decay of the short-lifetime component continues even past the fluorescence lifetime (albeit with diminished amplitude). Additionally, even though the excitation of the fluorophores is fast, it is not instantaneous. This convolution of the fluorescence decay with the instrument response function leads to a "slowing effect": when convolving an exponential of 230 ps lifetime with the instrument response function of our microscope, we found that still about 63% of the peak fluorescence were left outside the blanking window. Furthermore, due to the much higher brightness of the Ca^{2+} -bound OGB-5N (30-fold), an increase in Ca^{2+} -binding is much more noticeable in the long-lifetime component, which is much easier to fit and almost unaffected by blanking the first 300 ps of the recorded signal.

Most importantly, we were able to directly test the impact of blanking on the estimation of the fast component of fluorescence decay by applying it to confocal lifetime measurements, where the analysis of the fast component is not complicated by the STED-associated fluorescence decline (Fig. S6). We examined both in situ calibration data, as well as experimental data from IHC recordings, considering measurements with both varying brightness and $[\text{Ca}^{2+}]$ levels. In essence, the exponential fits using the "blanked" data range were highly similar to the fits using the full data range. No systematic bias was detectable, and the deviations between the "blanked" and the "full-range" fits

were minor: expressed in photons, only 0.4–2.8% of the photons were assigned to a different channel (i.e. slow instead of fast, or vice versa); for experimental data, the average (absolute) $\Delta[\text{Ca}^{2+}]$ was between 0.4–1 μM , or between 2–3% of the maximum $[\text{Ca}^{2+}]$ measured in that specific experiment (Figure S6). This systematic analysis is detailed in Supplemental Figures S5 and S6, as well as in the supplemental methods. Furthermore, we now state on page 18:

“Finally, the STED beam itself impacts the measured fluorescence lifetime (Figure S5, S6): by its very nature, STED causes an additional fast decay of fluorescence ($\tau = 0.19$ ns, Figure 6b, S6a, S8a), which needs to be considered to avoid overestimating the free (fast) dye contribution during STED measurements. When imaging with STED, we therefore excluded the data recorded in the first 300 ps (during the STED laser pulse) from the fitting routine. To ensure that this “blinking” of the first 300 ps did not significantly alter the results of the fitting routine, we fitted fluorescence decay traces recorded in confocal mode using both the full data range and the blanked data (Figure S6). Indeed, both methods produced highly similar results: the differences in the number of photons assigned to the fast and slow lifetime channel were as small as 0.4%–2.8% for $[\text{Ca}^{2+}]$ between 10 nM – 40 μM .”

8) I am unclear on the justification for applying a Ca-insensitive contribution to the fast and slow channels?

This is based on our experience with measuring Ca^{2+} -signals in IHCs. In severe contrast to our in vitro measurements, we found that in IHCs, even under strong buffering conditions (up to 10 mM EGTA), there was always a small fraction of dye exhibiting a slow fluorescence lifetime. Additionally, there remained a considerable fraction of fast lifetime even when $[\text{Ca}^{2+}]$ in the IHC was in the mM range: e.g. when patching IHCs with extreme amounts of Ca^{2+} in the intracellular solution (60 mM) or in IHCs that had lost cellular integrity and thus been flooded by extracellular Ca^{2+} (concentrations of 1.3 or 10 mM). We are sure that this was not due to poor imaging conditions, as we could recover the high dynamic range by measuring the identical (intracellular) dye solution not in the cell, but either directly inside the glass pipette or as it was blown out into the extracellular solution. Even though we have, thus far, not been able to deduce the origin of this behavior, the effect is clearly present and needs to be taken under consideration. Hence we expanded our model system to encompass not only “regular” OGB-5N, but also a fraction of dye that (for whatever reasons) was insensitive to $[\text{Ca}^{2+}]$, and either was permanently in the long-lifetime (“bound”) or short-lifetime (“free”) state. From this we derived to separate approaches to take this unresponsive dye fraction into account, after first performing calibration experiments in IHCs to determine the amount of Ca^{2+} -unresponsive dye (Fig.S11). Both approaches, the first of which directly re-scaled the dynamic range of the dye, the second of which only needed the unresponsive dye fraction to properly determine the minimum fluorescence in the resting state, yielded highly similar $[\text{Ca}^{2+}]$ results, strongly suggesting that the underlying assumptions are, indeed, correct. In the revised MS we have now offered more discussion of this observation:

“In contrast to the *in vitro* calibration, where the Ca^{2+} -bound state of OGB-5N could be either saturated (>99%, 10 mM $[\text{Ca}^{2+}]$) or completely de-populated (<0.5%, 10 mM EGTA), this was not the case when measuring inside living IHCs. Even when patching a cell with 60 mM $[\text{Ca}^{2+}]$, the contribution from the slow lifetime component never surpassed 60% (Figure S11). This might be due

to the dye interacting with molecules inside the cell, which change its properties, as has previously been described elsewhere⁴⁹.”

Regardless of the precise reasons, we argue that reliable calculation of synaptic $[Ca^{2+}]$ requires taking this “in vivo” calibration into account.

9) It seems like 3D STED would be much better suited than 2D STED for measuring $[Ca^{2+}]$, why wasn't this approach used as for the anatomy?

In response to the reviewer's comment we have experimentally explored the feasibility of 3D-STED Ca^{2+} -imaging at hair cell active zones in living tissue employing two different STED microscopes implementing 3D-STED i) by aligning STED beams shaped by 2 phase plates and ii) by use of a spatial light modulator. In accordance with our prior theoretical considerations, these efforts turned out to be neither straight forward nor simple to implement, primarily for two reasons. First, the tissue-induced aberrations affect the shape and the displacement of the xy-(vortex-) and z-(tophat)-donuts differently, making it very hard if not impossible to generate a “well-defined” 3D STED volume (we attribute this primarily to the imaging through the collagen-rich basilar membrane and the aqueous solution). Second, it became much more challenging to image the Ca^{2+} -domains at the higher resolution, as the domains were not immobile and the greatly reduced focal volume both required much higher precision in positioning the beams (the domains were easily lost when switching from confocal to 3D-STED), while simultaneously the number of pixels needed to be increased, resulting in considerably longer acquisition times. We note that this does not invalidate 3D-STED Ca^{2+} -imaging in general. For example, having a second super-resolution channel for localizing the active zone independently of Ca^{2+} -imaging and the related depolarizations will be of major help. Moreover, aberrations and sample drift will be much less severe in isolated cells on a cover-slip than in a tissue preparation as used here and hence 3D-STED will likely be considerably simpler under these conditions. This is beyond the scope of our current study, however, and calls for a new study in its own right. We now comment on the difficulties inherent to 3D-STED Ca^{2+} -imaging in the tissue, which we think is very valuable information to have when considering the approach. We note that the morphological “top-down” (from apex to base, avoiding imaging through the basilar membrane) 3D-STED imaging of IHC synapses in fixed organs of Corti embedded in Mowiol between two coverslips (which additionally flattens the tissue) has much less of this problem. The manuscript now reads:

“Further refinement of the technique by application of 3D-STED microscopy can be expected to lead to more precise measurements of $[Ca^{2+}]$ in the immediate proximity of the Ca^{2+} -channel cluster. We note, however, that functional 3D-STED imaging in tissue faces challenges, such as the tissue-induced aberrations distorting both STED donuts in a different fashion, as well as the trivial problem of locating Ca^{2+} -indicator hotspots with the reduced PSF size.”

In the light of the above problems with running 3D-STED Ca^{2+} -imaging in the tissue we have explored alternative ways to get closer to the $[Ca^{2+}]$ in the very proximity of the Ca^{2+} -channel cluster. Reasoning that the extracellular space should not contribute Ca^{2+} -indicator fluorescence, we focused the microscope slightly beneath the active zone of synapses that stratify primarily in the xy-plane (en face), aiming to reduce the measurement volume inside the cell (and thereby the volumetric averaging). In line with this reasoning and predictions of additional mathematical simulations we

recorded higher $[Ca^{2+}]$ than previously measured when we focused precisely on the maximum of the Ca^{2+} -indicator fluorescence hotspot. In conclusion, we corroborated our hypothesis that 2D-STED lifetime Ca^{2+} -imaging underestimates the $[Ca^{2+}]$ at the active zone due to the limited z-resolution and provide one practical way out without requiring 3D-STED imaging. We report on these results in the results part as follows:

“Finally, in order to further approach this question we modified our experiments to further reduce the volume from which $[Ca^{2+}]$ was sampled. For this, we shifted the focus of the objective below the AZ while measuring $[Ca^{2+}]$, so that the majority of the volume illuminated by the microscope’s PSF was outside the cell and thus devoid of dye (Fig. 6j). As the focus was moved outside the cell, the recorded fluorescence dropped (e.g. Fig. 6k), yet the measured Ca^{2+} -concentration indeed increased with increasing distance beneath the membrane (Fig. 6l,m), reaching average values as high as $25 \pm 7 \mu M$ (for STED recordings) at a distance of 800 nm from the brightest point of the hotspot (and thus most likely the membrane). This further supports our modeling data, which predicted similar concentrations (Fig 6m). These recordings displayed high variance (especially when recording with STED, reaching peak values between 45–50 μM), most likely because (due to drift) it was difficult to place and keep the recording in the exact center of the fluorescent hotspot, where $[Ca^{2+}]$ is at its highest.”

10) Fixed buffers do not seem to have been included in the simulations. These can have strong effects on the predicted $[Ca^{2+}]$ in the nanodomain, particularly if they are low affinity. On a related note, how dependent was the model based prediction of $[Ca^{2+}]$ on equilibrium assumptions?

After having initially found that fixed buffers did not have a significant effect on the modeling results (size of “hotspot”, $[Ca^{2+}]$) we had excluded those buffers from our model to speed up the calculations. We have now re-run the simulations with inclusion of fixed buffers and have found that indeed the effects are only minor - $[Ca^{2+}]_{max}$ as calculated from the distribution of Ca^{2+} -bound and -free dye, was decreased from 47 μM to 45 μM . We have now improved on the modeling results by using the fixed buffer data in the MS.

The model based prediction of free $[Ca^{2+}]$ at the channel mouth was very stable independent of buffering or the length of the simulation. The prediction of free $[Ca^{2+}]$ as reported from the ratio of Ca^{2+} -bound to -unbound dye quickly reached a steady state (see also Figure 2 of this letter below). The exact $[Ca^{2+}]$ reported from the model was dependent on the K_D used in the simulation (or, more precisely, the on- and off-rates). When assuming a K_D of 40 μM for OGB-5N the model predicted $[Ca^{2+}]$ that was about half as high as when assuming the K_D of 195 μM (21 vs 45 μM).

11) What accounts for the discrepancy in the estimate of the K_D for OGB-5N and previous studies?

The precise composition of the solution impacts the dissociation behavior of the dye in a major way. This became apparent when we tried to characterize OGB-5N as precisely as possible in an environment as close to the actual experimental conditions as possible. Although these findings were surprising, we were able to reliably reproduce them, in repeated experiments using different dye batches and different calculation methods. The apparent K_D reported here was measured in the intracellular solution which we have also used in our experiments. When recording the K_D in water

(with added Ca^{2+} , buffer, and dye) we found values which are much more comparable to previously published data. We have now included this data in Fig S7a. As to what exactly causes the shift in K_D in the intracellular solution, we unfortunately cannot tell with certainty. We now provide some possible interpretations, most importantly differences in the ionic strength which is known to affect the properties also of other Ca^{2+} -indicators:

“Next, we performed in vitro calibrations of Ca^{2+} -binding by OGB-5N in “minimalistic” solutions with well-known $[Ca^{2+}]$ (containing only Ca^{2+} , Ca^{2+} -buffer, and OGB-5N), under which conditions we found a K_D of $\sim 40 \mu M$ (Figure S7a), close to the published values of $30\text{--}40 \mu M$ ⁴⁵. We then repeated the exact same calibrations in solutions designed to closely match the intracellular solutions used in patch-clamp (with the addition of Ca^{2+} and citrate buffer to set the free $[Ca^{2+}]$). Under these conditions we found the dye to behave differently, with a much higher effective K_D (K_{eff}) of $\sim 195 \mu M$ (Figure S7b). Such behavior has been observed before⁴⁸, and might be explained by differences in ionic strength of the solution and by the binding- and buffering capabilities of additional components of the intracellular solution^{49,50}. As these second conditions most closely matched the actual experimental conditions, we used this value (K_{eff}) for all further calculations.”

Stylistic points

12) The title does not convey much about the findings.

We do see the point of the reviewer and have changed the title to “Quantitative optical nanophysiology of Ca^{2+} -signaling at inner hair cell active zones” to be more specific.

13) The delta on the alpha2 delta subunit symbol is not in formatted properly (not a Greek symbol).

We do not mention the alpha2delta subunit in our MS.

14) The introduction rather skirts around the key findings of previous work, rather than stating them clearly, which I found frustrating. I think the authors need to be a more candid about what is known about the properties of Ca channel clusters and the current models at both ribbon and central synapses. Also it sets up the necessity of measuring both the properties of Ca channel clusters and vesicles (which has not been possible), but this is not really achieved as the release sites are only assayed with bassoon labelling as opposed to identifying the location of individual vesicles. In particular the papers of Holderith et al 2012, Nakamura et al., 2015 and Keller et al., 2015 warrant more discussion given their conclusions. In addition, a more focused discussion on the parallels between findings at non-ribbon and ribbon synapses (e.g. clustering of Ca channels with highly variable numbers, nanodomain coupling (10-20 nm), predicted $[Ca^{2+}]$ in the tens of micromolar and vesicles arranged at the perimeter of the channel cluster) in the context of the results presented in the paper would significantly strengthen the paper.

In response to the concern of the reviewer we have now also added the Keller 2015 ref and further enhanced the comparative aspects in the introduction and discussion.

Reviewer #2 (Remarks to the Author):

Neef and colleagues studied Ca dynamics at active zones (AZs) of inner hair cells (IHCs) of the mouse cochlea. They used 2D or 3D STED microscopy to visualise immunohistochemically labelled voltage-gated Ca channels (Cav1.3) and bassoon. They found different morphological classes as well as strongly varying sizes of AZs. Using confocal spot-detection measurements of depolarisation-evoked Ca signals at putative AZs they estimated the number of Ca channels per AZ. Comparing spot-detection measurements with STED-based Ca imaging, in which Ca was quantified by time-correlated photon-counting lifetime imaging (FLIM), they analysed the Ca concentration within the STED point-spread function (SPF) at the AZ. Finally they incorporated the measured data into a model simulation of AZ Ca dynamics and estimated the Ca concentration at the AZ if the optical resolution in the z axis could have been better than 400-600 nm.

The authors used an impressive array of optical techniques to address the important and timely question of the nature of the Ca signal that drives release at AZs. However, the study falls short in addressing this question in a clear, unambiguous, convincing and quantitative way. Instead the manuscript makes the impression of a collection of different findings and techniques that were put together under the umbrella of “AZ nanophysiology”, without providing clear advances to the field and, in several instances, without solving the issue of unexpected and/or conflicting findings.

We thank the reviewer for the appreciation of our work and the valuable comments that helped us to further improve our MS. Below, please find our point-by-point responses.

General: We respectfully disagree with the reviewer’s notion of the study representing a collection of various imaging results short of advances in understanding of synaptic structure and function. Next to the major technical breakthroughs (e.g. functional STED; non-stationary optical fluctuation analysis) we contribute to the understanding of presynaptic function by i) counting of Ca^{2+} at the active zone, ii) immunohistochemical and functional estimation of Ca^{2+} -channel cluster shape, iii) super-resolution estimation of the $[Ca^{2+}]$ at the active zone. When revising our MS we undertook major efforts to further narrow down the measurement volume at the active zone to better approximate the $[Ca^{2+}]$ near the Ca^{2+} -channels. Moreover, we further strengthened our report on STED fluorescence lifetime measurements providing a comprehensive treatment of the method and important considerations for analysis. We very much hope that our revisions further strengthening our study will please the reviewer.

1) In the first part of the study the authors used 2D STED imaging to describe different morphological classes of AZs (Fig. 1a-j) and the average lengths and widths of AZs (Fig. 1k). They then switch to 3D STED (Fig. 2) and note that the length estimates of AZs increases if the 3D curvature is considered (Fig. 2d). Correctly, they assign this discrepancy to the obvious problem of taking 2D images of a curved structures oriented at different angles in space. Surprisingly, however, the authors do not go back to their 2D-based classification of AZs and reanalyse the putative classes of AZs in 3D images. This leaves the reader with a 2D-based, “new classification” of AZs although, for example, “point-shaped AZs” could well represent “single-line AZs” imaged from the side, “single lines” may be “complex clusters” imaged from the side and so on. The boomerang-shaped AZ shown in Fig.2a” represents a class not even identified in 2D images. Taken together, the

first part of STED-based immunohistochemistry did not provide convincing results other than estimates of the lengths of AZs, information well known from EM data.

A classification of synapses based on 3D STED data does not support such categorization in the same classes as used for the 2D STED data primarily due to the lower resolution in the x and y axis (~160 nm vs ~40 nm): e.g. it would not be possible to differentiate between “single lines” and “fat lines”. We do agree with the reviewer that the “point-shaped” clusters found in 2D STED images are likely to be single lines oriented along the z-axis, which is supported by their low incidence and the fact that we did not find any point-shaped clusters in the 3D STED data. We have clarified this by adding the following to the MS:

“At about 15% of the synapses, larger and more complex arrangements of Ca²⁺-channel and bassoon immunofluorescence were observed (Figure 1f,g) and less than 5% of the synapses exhibited small, spot-like Ca²⁺-channel and bassoon immunofluorescence (Figure 1h). The latter might represent spot-like clusters or stripe-like clusters that were aligned perpendicular to the imaging plane of the microscope.”

We find it unlikely, however, given the high incidence of linear clusters compared to only ~15% more complex clusters, that the linear clusters are simply complex clusters imaged from the side. This presumption is further supported by the finding that in EM data, presynaptic densities also form linear shapes (e.g. ref. 9, Wong et al., 2014, schematized in Figure 7). The aim of our study was an examination of Ca²⁺-channels in IHCs and the STED images from immunohistochemical stainings of Ca²⁺-channel clusters helped us to localize the channels on a nanometer scale. Even though the reviewer correctly states that the lengths of AZs have been examined in EM data before, these measurements did not allow a direct conclusion on the localization of Ca²⁺-channels, which are not visible in EM data, unless SDS replica-labeling (e.g. ref. 10, Nakamura et al., 2015) are used, which, on the other hand, do not provide information on channel location relative to the presynaptic density. Therefore, we are confident that our data does provide valuable new information.

2) The authors went on by addressing the questions of how many Ca channels are present per AZ. They used two different approaches: local suppression of Ca influx at single AZs and optical fluctuation analysis. Using these two methods they arrive at significantly divergent numbers of 144 and 74 channels per AZ, respectively. They conclude that the numbers probably represent over- or underestimates of the true values but did not try to dig deeper by either choosing a third method for counting channels or by analysing potential biases in the two methods. (For example, the significant age differences of the mice used in the two approaches [approx. P16 and P30, respectively] may explain part of the observed differences.)

Unfortunately, we do not have a third method for assessing the number of synaptic Ca²⁺-channels at the AZ at our disposal. We have undertaken a collaborative effort to perform SDS replica labeling of Ca²⁺-channels with Dr. Shigemoto, but this has not (yet) come to fruition, mostly because of the inhomogeneous tissue composition of the cochlea providing only very small fragments upon fracture. Regarding the age difference, we note that published data exclude a significant change in whole-cell Ca²⁺-current between p16 and p21 (ref. 9, Wong et al., 2014). Because of the maturational confinement of Ca²⁺-channels before p14 we do not expect a reversal of this process with a potential decrease in synaptic localization of Ca²⁺-channels to contribute to the lower Ca²⁺-channel counts at

the IHC AZs of the older mice. In fact, in unpublished data of a molecular physiology study we did not find significant differences between the maximal amplitude of the presynaptic Ca^{2+} -signals in different wildtype mice of different ages (see Figure 1 of this response letter), supporting this view.

Figure 1:

Synaptic Ca^{2+} -signals imaged by spinning-disk confocal microscopy as described in the MS. We plot the maximal Fluo-8FF fluorescence increase (background normalized) in response to ramp-depolarization of IHC active zones from wildtype mice of different age. We did not find significant differences.

In response to the reviewer's comment, we have further scrutinized our analysis of the EGTA experiments and also considered the extrasynaptic Ca^{2+} -channel population more quantitatively. Quantifying the effects of EGTA iontophoresis on the neighboring synapses revealed a minor but measurable 'off-target' effect that we now corrected for.

We state in the main MS, page 9:

"We calculated the influence of the observed changes in neighboring AZs on the whole cell calcium currents (Figure S1a, Supplementary Methods) and corrected for this when estimating Ca^{2+} -influx in the synapse under study. On average, the total signal change in neighboring AZs was $12 \pm 4\%$ of that of the targeted AZ."

This reduced the average count of synaptic Ca^{2+} -channels from 144 to 125 for this method, which is still greater than the 78 estimated by the fluctuation analysis. Therefore, in addition to presenting this corrected approach we added further discussion of the potential shortcomings of the two methods to the supplementary materials:

"Potential overestimation of the number of Ca^{2+} -channels measured by application of EGTA

A considerable fraction of Ca^{2+} -channels (estimated here to be up to 50 %) in IHCs is extrasynaptic. Spread of EGTA following iontophoretic application at the AZ might also suppress Ca^{2+} -currents through nearby extrasynaptic channels. If we assume that in IHCs extrasynaptic Ca^{2+} -channels are only found in the $\sim 550 \mu\text{m}^2$ of basolateral membrane (see ref. 5 for details), we arrive at a density of ~ 1 extrasynaptic channel per μm^2 . Further assuming that EGTA does not spread further than $2 \mu\text{m}$ from the site of application, we would conclude that no more than ~ 12 extrasynaptic channels are suppressed by EGTA near the analyzed AZs, resulting in an overestimation of on average $\sim 10\%$, which

would more strongly affect the smaller AZs. However, it appears quite possible that the density of extrasynaptic channels is higher near the AZs, which would result in an even higher overestimation of the number of synaptic Ca²⁺-channels.”

as well as in

“Modeling uneven contribution of Ca²⁺-channels

In addition to potential artifacts arising from bandwidth limitations imposed by the reaction kinetics of the Ca²⁺-indicator, we expected an effect of an uneven contribution of the individual Ca²⁺-channels, due to their differential localization within the PSF of our imaging system, which we addressed by modeling (Fig. S2d). To this end, we simulated Ca²⁺-currents for each channel and each sweep in the same way as described above (however, without considering any reaction between free Ca²⁺-ions and the Ca²⁺-indicator (Fluo-4FF) as a simplification) and then scaled it by the relative position of the Ca²⁺-channel with regard to the center of the PSF. Again, we then summed the simulated traces for all Ca²⁺-channels in each sweep and calculated the ensemble mean and variance for all 50 sweeps (Fig. S2e,f).

However, the activity of Ca²⁺-channels situated completely outside the PSF would still be lost to the imaging process if the Ca²⁺ entering through these channels would not diffuse into the PSF. Considering the larger size of IHC Ca²⁺-channel clusters (100 – 600 nm, compared to a PSF size of ~280 nm), it appears likely that this method underestimates the number of Ca²⁺-channels present at the AZ.”

3) Finally the authors developed STED-based Ca imaging in order to describe depolarization-induced Ca signal at AZs at a better-than-confocal resolution. This section starts rather ambiguous: The authors state that they used two-different solutions (“intensified” vs. “physiological”) for recording Ca signals (p. 12). One page later, however, they state to have focussed on the “physiological” solution. One page of the text could have been spared by streamlining the study. More importantly, the authors fell short in fully establishing STED-based Ca imaging: the pitfalls of this technique should have been addressed more thoroughly by discussing a) the exact dimensions of the PSFs for apo- and Ca-bound dye moieties, b) showing calibrations that validate the diverse correction factors deduced by the authors, c) discussing how diffusion of Ca, either free or buffered by endogenous and exogenous buffers, affects the PSF and, finally, d) whether increasing the lateral resolution via STED can be expected to yield more accurate estimates of Ca at the AZ than confocal imaging; given the size of AZs and the similar z resolution of both techniques, the difference can be expected and indeed were found, to be rather small (p. 17 and Fig. 6e,f). In summary, the section on STED-based Ca imaging remained descriptive without going sufficiently deep into photophysical problems of the technique and did not yield new experimental insight into AZ Ca dynamics.

Good points that we have addressed one by one. We have now “streamlined” the presentation by focusing the MS on the more favorable “physiological” imaging conditions. The saved space has been devoted to further establish STED Ca²⁺-imaging and provide more insight into the photophysical issues. There and in the suppl. we have addressed the 4 concerns brought up by the reviewer. We have then introduced the intensified conditions where it supports our discussion of the discrepancy of the present and previous Ca²⁺-imaging result on the AZs of Bassoon-deficient IHCs.

a) the exact dimensions of the PSFs for apo- and Ca-bound dye moieties:

We cannot directly measure the exact dimensions of the PSFs of Ca²⁺-bound and -free dye, since we would need a sub-resolution-sized sample of dye for this (e.g. a dye-filled 10 nm bead or droplet), which is hard if not impossible to generate. Moreover, encasing the dye within a bead or attaching it to a surface might well significantly change its properties. The confocal PSFs of the Ca²⁺-bound and -free dye should be practically identical, as the spectral properties of the dye do not change upon Ca²⁺-binding. What does change, however, is the fluorescence lifetime, which in turn should only affect the effective STED PSF by reducing the STED efficiency of the unbound OGB-5N. As a very rough estimate, we approximated the STED PSF size from the decrease in fluorescence intensity upon STED illumination, resulting in upper estimates of the spatial resolution of 80 nm and 150 nm for Ca²⁺-bound and -free dye, respectively.

b) showing calibrations that validate the diverse correction factors deduced by the authors

The calibrations had already been provided in the supplement. We have, however, considerably expanded the discussion of the considerations necessary for STED Ca²⁺-imaging by introducing the corrections and calibrations in a more systematic and intelligible way and by showing more of the underlying data. The supplement now encompasses detailed information regarding the brightness and STED efficiency correction parameters, the effective dissociation constant in intracellular solution, the lifetime estimation of the Ca²⁺-bound and -free dye states, the “blanking” method for fitting STED data, the reduced dynamic range of the dye within IHCs, as well as all the equations, their origins and how they were derived. We are confident that this meets the reviewer’s expectations.

c) discussing how diffusion of Ca, either free or buffered by endogenous and exogenous buffers, affects the PSF

We assume that the reviewer primarily refers to how our imaging-based approximation of the size of the presynaptic Ca²⁺-signals also depends on the diffusional spread of Ca²⁺ and Ca²⁺-indicator. This is a very valid point and we addressed this by experiments and modeling. First of all, both fast line-scans of depolarization-evoked presynaptic Ca²⁺-hotspots in IHCs as well as our simulations state that both the spatial extent of the fluorescence change of the Ca²⁺-indicator as well as the Ca²⁺-concentration reach a steady state after less than one millisecond in our recording conditions. Experiments further support the notion that increased Ca²⁺-influx and high Ca²⁺-indicator concentration (“intensified” conditions) increases the size of the presynaptic Ca²⁺-signals. This is in agreement with CalC modeling where we previously showed that the spatial extent of Ca²⁺-indicator fluorescence generally exceeds that of the Ca²⁺-channel cluster. We now found that increasing Ca²⁺-influx and Ca²⁺-indicator concentration goes along with larger size of the Ca²⁺-indicator fluorescence spots. We have now further enhanced this discussion in the MS on page 14/15:

“The dimensions of the observed hotspots matched well with reaction/diffusion simulations of presynaptic Ca^{2+} -signals we performed using the CalC modeling software⁴⁶, based on the number and distribution of channels as measured before (430 x 67 nm cluster of 120 Ca^{2+} -channels, Figure 6g–i, S14, Table 1), which resulted in a presynaptic Ca^{2+} -domain with a FWHM of 190 x 343 nm (as estimated by fitting of a 2D Gaussian function to the distribution of Ca^{2+} -bound OGB5-N), considerably larger than the size of the Ca^{2+} -channel cluster due to lateral diffusion of dye and Ca^{2+} -ions (Figure S13).”

And on page 15/16:

“We further addressed this question in additional experiments using conditions that enhanced Ca^{2+} -influx (10 mM $[\text{Ca}^{2+}]_e$) and also involved higher OGB-5N (300 μM) and EGTA (10 mM; “intensified conditions”; Figure 5f). We still observed the above described wide-spread Ca^{2+} -signals in bassoon-deficient IHCs in confocal and STED Ca^{2+} -imaging. Interestingly, we found larger Ca^{2+} -hotspots for wildtype synapses in the “intensified conditions”, where $[\text{OGB-5N}]$ was much greater than the expected presynaptic $[\text{Ca}^{2+}]$, than described above for more physiological conditions, where $[\text{OGB-5N}]$ was more similar to $[\text{Ca}^{2+}]$ at the synapse.”

Additionally, we have added the Supplementary Figure S13 and the following paragraph to the supplement:

“Diffusional spread of Ca^{2+} -bound OGB-5N from the AZ

There are two relevant time scales when considering what influence diffusion of the Ca^{2+} -dye or –ions might have on our measurements. First, the diffusion of ions and dye away from the AZ over the course of an image, i.e. over (tens of) milliseconds or until a steady state sets in, and second, the possible movement of the dye during one excitation/detection-cycle (as determined by the temporal separation of the individual laser pulses of ~25ns), which could conceivably cause a “blurring of the PSF” as excited dye molecules from inside the (effective) focal volume of the STED nanoscope diffuse outside before spontaneously fluorescing.

We performed line-scan recordings of the depolarization-evoked fluorescence increase at IHC AZs in order to measure the speed of the lateral increase of fluorescence away from the AZ. Even though we measured at scan rates of 1.5 kHz, we still could only detect a smaller-sized presynaptic Ca^{2+} -domain during the first line of the scan (< 1ms, see Fig. S13a,b). This finding is in line with modeling data indicating that the lateral spread of Ca^{2+} -bound OGB-5N reaches a steady state after as little as 400 μs (Fig. S13c). Since this steady state is reached early on during the depolarization, we can assume the Ca^{2+} -domain to be stable during our image acquisition, such that lateral diffusion of the dye does not impair the imaging conditions.

Additionally, diffusional movement of individual excited dye molecules out of the effective focal volume of the STED nanoscope is not expected to have any noticeable impact on the spatial resolution, given the limited time frame. Within the 25 ns elapsing between two consecutive laser pulses, a single dye molecule is expected to move on average as little as 2.3 nm (calculated from $l = \sqrt{t \cdot 2D}$, with l the average diffusion length, t the time available for diffusion [here 25 ns], and D the diffusion coefficient [here 0.1 $\mu\text{m}^2/\text{ms}$]). Given an effective focal area (during STED measurements) between 50 nm–100 nm, this effect should be entirely negligible.”

d) whether increasing the lateral resolution via STED can be expected to yield more accurate estimates of Ca at the AZ than confocal imaging

We share the reviewers view that the increase in lateral resolution was not sufficient to measure the “true” Ca^{2+} -concentration at the AZ under most conditions (i.e. measuring at AZs where the Ca^{2+} -channel cluster is not running in parallel to the z-axis).

In response to the reviewer’s comment we have experimentally explored the feasibility of 3D-STED Ca^{2+} -imaging at hair cell active zones in living tissue employing two different STED microscopes implementing 3D-STED i) by aligning STED beams shaped by 2 phase plates and ii) by use of a spatial light modulator. In accordance with our prior theoretical considerations, these efforts turned out to be neither straight forward nor simple to implement, primarily for two reasons. First, the tissue-induced aberrations affect the shape and the displacement of the xy-(vortex-) and z-(tophat)-donuts differently, making it very hard if not impossible to generate a “well-defined” 3D STED volume in the functional experiments (we attribute this primarily to the imaging through the collagen-rich basilar membrane and the aqueous solution). Second, it became much more challenging to image the Ca^{2+} -domains at the higher resolution, as the domains were not immobile and the greatly reduced focal volume both required much higher precision in positioning the beams (the domains were easily lost when switching from confocal to 3D-STED), while simultaneously the number of pixels needed to be increased, resulting in considerably longer acquisition times. We note that this does not invalidate 3D-STED Ca^{2+} -imaging in general. For example, having a second super-resolution channel for localizing the active zone independently of Ca^{2+} -imaging and the related depolarizations will be of major help. Moreover, aberrations and sample drift will be much less severe in isolated cells on a cover-slip than in a tissue preparation as used here and hence 3D-STED will likely be considerably simpler under these conditions. This is beyond the scope of our current study, however, and calls for a new study in its own right. We now comment on the difficulties inherent to 3D-STED Ca^{2+} -imaging in the tissue, which we think is very valuable information to have when considering the approach. We note that the morphological “top-down” (from apex to base, avoiding imaging through the basilar membrane) 3D-STED imaging of IHC synapses in fixed organs of Corti embedded in Mowiol between two coverslips (which additionally flattens the tissue) has much less of this problem. The manuscript now reads:

“Further refinement of the technique by application of 3D-STED microscopy can be expected to lead to more precise measurements of $[\text{Ca}^{2+}]$ in the immediate proximity of the Ca^{2+} -channel cluster. We note, however, that functional 3D-STED imaging in tissue faces challenges, such as the tissue-induced aberrations distorting both STED donuts in a different fashion, as well as the trivial problem of locating Ca^{2+} -indicator hotspots with the reduced PSF size.”

In the light of the above problems with running 3D-STED Ca^{2+} -imaging in the tissue we have explored alternative ways to get closer to the $[\text{Ca}^{2+}]$ in the very proximity of the Ca^{2+} -channel cluster. Reasoning that the extracellular space should not contribute Ca^{2+} -indicator fluorescence, we focused the microscope slightly beneath the active zone of synapses that stratify primarily in the xy-plane (en face), aiming to reduce the measurement volume inside the cell (and thereby the volumetric averaging). In line with this reasoning and predictions of additional mathematical simulations we recorded higher $[\text{Ca}^{2+}]$ than previously measured when we focused precisely on the maximum of the

Ca²⁺-indicator fluorescence hotspot. In conclusion, we corroborated our hypothesis that 2D-STED lifetime Ca²⁺-imaging underestimates the [Ca²⁺] at the active zone due to the limited z-resolution and provide one practical way out without requiring 3D-STED imaging. We report on these results in the results part as follows:

“Finally, in order to further approach this question we modified our experiments to further reduce the volume from which [Ca²⁺] was sampled. For this, we shifted the focus of the objective below the AZ while measuring [Ca²⁺], so that the majority of the volume illuminated by the microscope’s PSF was outside the cell and thus devoid of dye (Fig. 6j). As the focus was moved outside the cell, the recorded fluorescence dropped (e.g. Fig. 6k), yet the measured Ca²⁺-concentration indeed increased with increasing distance beneath the membrane (Fig. 6l,m), reaching average values as high as 25 ± 7 μM (for STED recordings) at a distance of 800 nm from the brightest point of the hotspot (and thus most likely the membrane). This further supports our modeling data, which predicted similar concentrations (Fig 6m). These recordings displayed high variance (especially when recording with STED, reaching peak values between 45–50 μM), most likely because (due to drift) it was difficult to place and keep the recording in the exact center of the fluorescent hotspot, where [Ca²⁺] is at its highest.”

4) Similarly, the establishment of STED-based FLIM remained rather superficial.

a) Lifetimes (LTs) were described to be monoexponential although the data and the corresponding fits shown that they are not (Fig. S5b). Here the authors did not adhere to the standards in the field where data are shown as log-lin plots with residuals and Chi² values indicating the goodness of fits.

We have revised Supplementary Figures S5 and S6 to display log-lin plots and residuals.

In response to the reviewers comment we have added our considerations regarding double-exponential fitting for the low Ca²⁺ condition to the supplements (the high Ca²⁺ condition is well-described by a monoexponential decay). Even though the fluorescence decay of the free OGB-5N cannot be described cleanly with a monoexponential decay, fitting with more than one exponential decay proved ineffective – the fitted lifetime of a second decay component fluctuated wildly from measurement to measurement (and hence was impossible to lock down), and the amplitude of this second component was typically an order of magnitude smaller than the (original) fast component. Yet despite the imperfections concerning the fit of the short-lifetime component, due to the stark differences in lifetime between the long and short dye component, paired with the very accurate fitting of the long lifetime component, the fitting algorithm proved to very robustly be able to discriminate between both dye moieties. We tested the robustness of the fits using tri-exponential fitting, using different short lifetime values (which were increased or decreased up to twofold), as well as in combination with blanking. In all cases, the resulting deviations from our used parameters were fairly small, typically less than 3–5%. So, despite the imperfections of the fits, the method was highly robust in discerning the Ca²⁺-bound from the -free dye state, which could be shown among others by the consistent in vitro calibration measurements. Please note that double-exponential fitting (with the two time constants: 3.24 ns and 0.23 ns fixed) was used for all other conditions and regularly reported fast and slow components.

b) Blanking the part of the lifetime curve distorted by the STED pulse may simplify the analysis of LT curves, however, the authors then need to state how they normalized the fractions of fast and slow LT components: to the extrapolated peak of the LT curve within the blanked period or to the first data point after the blanking period?

Done, this has now been stated in the methods section. Even though the blanked period was excluded from the actual fitting process, the amplitudes were back-extrapolated to the onset of fluorescence (i.e. in the blanked data period). These amplitudes (α'_B and α'_{U}) were then used as in the confocal case to calculate the integral of the fitted curve, $I = I_B + I_U = (\alpha'_B \cdot \tau_B + \alpha'_{U} \cdot \tau_U)$, which was then normalized using the total number F of detected photons: $F_s = \frac{F \cdot I_s}{I_s + I_f}$ and $F_f = \frac{F \cdot I_f}{I_s + I_f}$.

We have updated the methods part (as well as the supplements) to clarify the issue as follows:

“The ratio of free to Ca^{2+} -bound dye was obtained by first calculating the number of photons F_i that were assigned to the fast and slow channels, respectively ($F_i = \alpha_i \cdot \tau_i$, with α_i the amplitudes from the fitting routine; for STED data, the amplitude extrapolated to the (blanked) onset of the fit was used).”

We further validated the robustness of the blanking approach by using it on confocal data, which is not influenced by a STED-evoked fluorescence lifetime component, and comparing the results of blanked and unblanked fits. We tested this both on in situ and experimental data from IHCs, for varying brightness and $[\text{Ca}^{2+}]$ levels, and found the results to be very close (Fig. S6).

c) The authors failed to explain why there is no fast, STED-induced LT component in Fig. S4c, red dotted data curve.

There is no STED-induced lifetime shortening here, because the recording was taken in “reverse-pulse” mode, meaning that the STED beam arrived before the excitation pulse. This “mode” has nothing to do with STED per se, but was just used as a test whether the STED beam itself caused damage to certain dyes (which the two panels in Fig. S4c illustrate). We have clarified this in the figure legend:

“(c) Lifetime measurement of Fluo-4FF and Fluo-5N with and without STED in low and high $[\text{Ca}^{2+}]$. Due to the reverse pulse used here, no change in lifetime should be visible when using STED light. Both indicators showed a change towards longer fluorescent lifetime with STED light in low $[\text{Ca}^{2+}]$ (dotted and straight blue line) while there was no change in high $[\text{Ca}^{2+}]$ (red). The increase in lifetime of the Ca^{2+} -free dye might reflect photo-damage of the Ca^{2+} -chelator moiety of the dye which normally quenches the fluorescence. OGB-1, OGB-5N, Fluo-4 and Fluo-5F, showed no indication for photo-damage with lifetime imaging or with STED in reverse pulse mode (b) and therefore are suitable for STED Ca^{2+} -imaging. Our study thereafter focused on using OGB-5N because its low affinity met our requirements for imaging AZ Ca^{2+} -signals.”

d) Similarly, they failed to convincingly explain the finding that the K_D of OGB5N in pipette solutions is 5 times higher than in commercial buffer solutions. P. 16 leaves many open questions on how the calibration was done.

The apparent k_D reported here was measured in the intracellular solution which we have also used in our experiments. When recording the k_D in water we found values which are much more comparable to previously published data (Suppl, Fig. S7a). Obviously, the precise composition of the solution impacts the dissociation behavior of the dye in a major way. Such discrepancy has also been reported by other studies: For example, ref. 49 (Uto et al., 1991) found that ionic strength, pH and co-existing proteins strongly affect the apparent k_D of the Ca^{2+} -indicator Fura-2 in cells. Although the extent of change found for the k_D of OGB-5N was surprising, we were able to reliably reproduce this finding, in repeated experiments using different dye batches and different calculation methods. We assume that OGB-5N is strongly affected and conclude that calibration in solutions mimicking the recording conditions as closely as possible is of paramount importance and have expanded the MS as follows:

“Next, we performed in vitro calibrations of Ca^{2+} -binding by OGB-5N in “minimalistic” solutions with well-known $[Ca^{2+}]$ (containing only Ca^{2+} , Ca^{2+} -buffer, and OGB-5N), under which conditions we found a K_D of $\sim 40 \mu M$ (Figure S7a), close to the published values of $30\text{--}40 \mu M$ ⁴⁵. We then repeated the exact same calibrations in solutions designed to closely match the intracellular solutions used in patch-clamp (with the addition of Ca^{2+} and citrate buffer to set the free $[Ca^{2+}]$). Under these conditions we found the dye to behave differently, with a much higher effective K_D (K_{eff}) of $\sim 195 \mu M$ (Figure S7b). Such behavior has been observed before⁴⁸, and might be explained by differences in ionic strength of the solution and by the binding- and buffering capabilities of additional components of the intracellular solution^{49,50}. As these second conditions most closely matched the actual experimental conditions, we used this value (K_{eff}) for all further calculations.”

e) Labelling time axes with “a.u.” in S4b is inappropriate for understanding the increases in fluorescence induced by the STED-Laser for some of the dyes.

We agree that even though the panels clearly show that there is no time dependence other than the “on/off” behavior, the rough range of the depicted time window is critical for understanding the data. The electro-optical modulator switched the STED beam on and off every 40 ms. We have amended the graph to clarify this.

5) The data on Bassoon-mutant AZs are insufficient to draw strong conclusions because the number of observations is quite small (cf. Fig. 5: $n=9$ for BSN-mutants, $n>50$ for wildtype mice). More importantly, the data are neither connected to the topic of the manuscript nor are they sufficiently worked out to represent a “bassoon story” on their own.

In response to the reviewer comment we have increased the number of bassoon-mutant recordings. Our main point remains: there is no obvious difference between confocal and STED images, since the underlying Ca^{2+} -signal is larger than the resolution of the confocal microscope (which was already apparent from a smaller sample). On their own, these recordings do not represent a full “bassoon story”. We do however believe that these data are not only interesting in their own right, but especially because they underline the conclusion that the shape of Ca^{2+} -channel clusters as seen in

immunohistochemical images does not necessarily represent the native state in the living cell. In our example, we could show that, since the depolarization-evoked hotspots of synaptic Ca^{2+} -influx appear elongated in STED Ca^{2+} -imaging, a stripe-like clustering of Ca^{2+} -channels at the AZ as seen in immunohistochemical data, appears to be the likely arrangement in wildtype IHCs, while the point-like clustering observed in Bassoon mutant IHCs appears to be an artifact of fixation which could only be determined by Ca^{2+} -imaging.

Additional comments:

6) Throughout the manuscript exact information about the PSFs of the different imaging techniques (and how PSFs were measured) is missing.

Done, we added confocal and STED PSFs measured on 40 nm fluorescent microspheres, both along the optical axis and in the optical plane, to the supplement (Suppl. Fig. 12).

7) Why would a holding potential of +53 mV not permit Ca influx into IHCs (p. 9, line 212)?

Because in IHCs we encounter an apparent reversal potential for the Ca^{2+} -currents of $\sim +45$ mV. We have clarified this with the following addition:

“We depolarized the IHCs to +53 mV, past the (apparent) reversal potential of Ca^{2+} in IHCs (~ 45 mV), opening Ca^{2+} -channels but not permitting Ca^{2+} -influx, (...)”

8) For statistical tests, p and power values should be provided.

P values and the applied tests have been added where missing.

9) The schemes illustrated in Fig. 7 are based on published EM work, not on data from the present study.

These schemes are indeed based on a previously published scheme which is based on EM data, as is also explained in the figure legend. Reuse of the figure (modified or not) is permissible as per journal (EMBO J) policy. Still, we have added the following to the figure legend to clearly state the source:

“Schematic representation of four exemplary IHC AZs with similar layout but different length and number of the Ca^{2+} -channel clusters (modified from ref. 9).”

Furthermore, we have slightly updated the number of Ca^{2+} -channels visible in the scheme to more directly reflect our data.

Reviewer #3 (Remarks to the Author):

In this report, Neef and colleagues characterize the features of Ca²⁺ signaling at the active zones (AZs) of cochlear inner hair cells (IHCs). They make extremely careful functional measurements using a truly awesome array of optical approaches including new super-resolution techniques (e.g., STED-based fluorescence lifetime measurements) to quantify, beyond all previous attempts, details pertaining to Ca²⁺ channel organization, number, and Ca²⁺ domain signaling. Because of the widespread interest in presynaptic mechanisms regulating neurotransmission, this report will provide a wealth of knowledge to a broad readership and will be a common reference for advanced optical approaches to study the function of Ca²⁺ channels and their organization at presynaptic elements.

I have no major concerns with the paper and only have a couple of points that the authors should consider.

We thank the reviewer for the appreciation of our work and the valuable comments that helped us to further improve our MS. Below, please find our point-by-point responses.

In this report, Neef and colleagues characterize the features of Ca²⁺ signaling at the active zones (AZs) of cochlear inner hair cells (IHCs). They make extremely careful functional measurements using a truly awesome array of optical approaches including new super-resolution techniques (e.g., STED-based fluorescence lifetime measurements) to quantify, beyond all previous attempts, details pertaining to Ca²⁺ channel organization, number, and Ca²⁺ domain signaling. Because of the widespread interest in presynaptic mechanisms regulating neurotransmission, this report will provide a wealth of knowledge to a broad readership and will be a common reference for advanced optical approaches to study the function of Ca²⁺ channels and their organization at presynaptic elements.

I have no major concerns with the paper and only have a couple of points that the authors should consider.

1) Their use of selective suppression of Ca²⁺ influx at an individual AZ to estimate Ca²⁺ channel number is novel approach and seems well-controlled regarding single AZ selectivity (i.e., lack of effect on nearby AZs). However, the EGTA-induced reduction of the whole-cell current seems large given that it should be occurring at a single AZ (at least from the example in 3c), especially considering that there are >10 AZs per IHC. They mention that this could be due to selection bias (picking AZs with strong Ca²⁺ signaling) but is it realistic to believe that a few AZs make contain a majority of Ca²⁺ channels in an IHC especially if 10-30% are extrasynaptic (maybe they are blocking a significant portion of this population as well)? It might be helpful if they reported the average effective block of EGTA on the whole-cell tail current in this experiment (the current lost by EGTA application) rather than the current size of a single AZ alone. Also, they don't report the percent block of the Ca²⁺ response at the targeted AZ (again, their example makes it look sub-maximal). Could they use "intensified" conditions (i.e., higher external [Ca²⁺]) to pull signal out from more weakly responding AZs? How might a selection bias effect the calibration procedure against the remainder of the AZs examined?

The current trace in figure 3c had been cut in order to more clearly display the decrease in whole-cell Ca^{2+} -current. We realize that the diagonal bars that were intended to signify this are not sufficiently clear. We have updated Figure 3c and hope that it is now more clearly visible that the decrease in whole-cell current is not as large as it might have appeared from the old version of the figure (i.e. in this example: up to 18 pA reduction from a whole-cell Ca^{2+} -current of 180 (initial) to 140 pA (late), so a reduction of 10-13 %). We believe that this, in combination with the mentioned selection bias (those AZs from which the relationship between $\Delta F/F_0$ and ΔI_{Ca} was calculated had an average maximal $\Delta F/F_0$ of 1.92 ± 0.78 while the total pool of all measured synapses had an average maximal $\Delta F/F_0$ of 1.05 ± 0.52) seems compatible with a total number of ~ 12 AZs per IHC and 20-50% extrasynaptic channels.

Additionally, in response to the reviewer's comment, we have further scrutinized our analysis of the EGTA experiments and also considered the extrasynaptic Ca^{2+} -channel population more quantitatively. Quantifying the effects of EGTA iontophoresis on the neighboring synapses revealed a minor but measurable 'off-target' effect that we now corrected for.

We state in the main MS, page 9:

"We calculated the influence of the observed changes in neighboring AZs on the whole cell calcium currents (Figure S1a, Supplementary Methods) and corrected for this when estimating Ca^{2+} -influx in the synapse under study. On average, the total signal change in neighboring AZs was $12 \pm 4\%$ of that of the targeted AZ."

This reduced the average count of synaptic Ca^{2+} -channels from 144 to 125 for this method, which is still greater than the 78 estimated by the fluctuation analysis. Therefore, in addition to presenting this corrected approach we added further discussion of the potential shortcomings of the two methods to the MS:

"In summary, both experimental approaches reported somewhat different estimates for the mean number of Ca^{2+} -channels per AZ (125 and 78), suggesting either an overestimation of the number of

channels after application of EGTA (e.g. due to additional effect on extrasynaptic Ca^{2+} -channels that is difficult if not impossible to quantify from our experiments), or an underestimation in the optical fluctuation analysis experiments (e.g. due to activity of synaptic channels outside the microscope's PSF), or a combination of both. Nevertheless, the observation of a large range of Ca^{2+} -channels per AZ by both approaches confirms that the previously reported heterogeneity of Ca^{2+} -signaling among the AZs of a given IHC¹⁵ results from different numbers of Ca^{2+} -channels per AZ. Indeed, the coefficients of variation found for Ca^{2+} -channel number per AZ by both approaches (0.5 and 0.64) were comparable to that previously reported for the maximal synaptic Ca^{2+} -influx (0.65, ref. 15) and to that which we found for the integrals of the 2D Gaussian fits to $\text{Ca}_v1.3$ immunofluorescence of Ca^{2+} -channel clusters (0.67, data from Fig. 1). This AZ heterogeneity might be related to functional diversity of the spiking behavior of postsynaptic spiral ganglion neurons^{15,44}. Finally, we can determine the total number of synaptic Ca^{2+} -channels by multiplying the average number of Ca^{2+} -channels per AZ with the count of synapses per IHC (typically 12 in the area of the organ of Corti used in our measurements), arriving at a value between 950–1500. Relating this to the total number of Ca^{2+} -channels per IHC (~1900, see above) indicates that approximately 20%–50% of all channels localize extrasynaptically. “

Additionally, we now report the average decrease in fluorescence by application of EGTA as follows:

“We monitored the reduction of synaptic Ca^{2+} -influx during a 150 ms long depolarization of the IHC to -14 mV by the decrease of the Fluo-8FF fluorescence at the synapse ($\Delta(\Delta F/F_0)$, Figure 3c, top, on average down to 52%) and of the total IHC Ca^{2+} -influx (ΔI_{Ca} , Figure 3c, bottom).”

2) It is surprising that the authors did not estimate Ca^{2+} channel numbers in their Bassoon mutant. They rely on the mutant for comparison of structural details of Ca^{2+} channels organization but little in the way of functional details (apart from the width of a Ca^{2+} hotspot in 5e). This is not only pertinent (they previously published reduced Ca^{2+} channel number following loss of Bassoon [see Frank et al. 2010]) but also would help validate their techniques (e.g., the expectation is that average number of Ca^{2+} channels per active zone will decrease and there will be an accompanying increase in the extrasynaptic population).

An estimate of the number of synaptic Ca^{2+} -channels in bassoon mutant IHCs would be very difficult to achieve using either of the two methods introduced in this MS. The method of suppression of Ca^{2+} -influx at individual synapses using local application of EGTA heavily relies on localization of synaptic ribbons using the RIBEYE binding peptide. Since bassoon mutant IHCs lose most of their synaptic ribbons (Khimich et al., 2005), this is not possible here. Moreover, the large spread of Ca^{2+} -channels at the IHC synapses of bassoon mutants would make the estimation less precise, as then selective extinction of Ca^{2+} -influx at the targeted active zone is expected to be more challenging. Finally, the reduced amplitude of the Ca^{2+} -hotspots would further reduce the validity of the approach. While the optical fluctuation analysis does not require knowledge about neighboring synapses, the large size of the depolarization-evoked Ca^{2+} -hotspots in bassoon mutant IHCs would mean that a large amount of the presynaptic Ca^{2+} -influx would occur outside the PSF, leading to a severe underestimation of the number of synaptic Ca^{2+} -channels. Therefore, we have refrained from experimentally addressing this interesting question and refer to our previous data that indicated an overall reduction of the number of Ca^{2+} -channels (by analysis of whole-cell Ca^{2+} -current fluctuation analysis).

Reviewer #4 (Remarks to the Author):

This review is mostly interested in the large number of delicate optical techniques that the authors use to achieve this study. In this manuscript, two dimensional and three dimensional super-resolution STED imaging and fluorescence lifetime measurements were performed to study the morphology and the calcium concentration in the vicinity of the active zones of synapses of sensory hair cells. Super-resolution STED imaging appeal careful characterizations that the authors rigorously manage and expose in a quite clear manner.

We thank the reviewer for the appreciation of our work and the valuable comments that helped us to further improve our MS. Below, please find our point-by-point responses.

I have a few comments and questions:

- The manuscript first present morphological images using a STED system lacking super-resolution along the longitudinal axis before using a other system achieving isotropic super-resolution. The purpose of this first study is unclear to us. If the goal of the first part is to conclude that biological results are irrelevant with 2D STED, I would suggest placing these inaccurate data in supplementary materials. Moreover, the introduction stresses on the results about Ca imaging but do not motivate the need for a morphological study.

We did not mean to imply that biological results are irrelevant with 2D-STED. Instead, we see that both techniques have pro's and con's. Therefore, we employed them in a complementary purpose. At least in our realization we achieved greater xy resolution in 2D-STED (50 nm) than in 3D-STED (160 nm). Hence the categorization of cluster shapes as provided needs to rely upon 2D-STED imaging of synapses that ideally stratify in the xy-plane. This condition, however, is not always met and hence, 3D-STED is required, e.g. to acquire the real cluster length, as the 2D-imaging can only report an "apparent length". Moreover, the category of small spot-like Ca²⁺-channel clusters, might instead represent "stripes" that localized vertically relative to the xy-plane. We have now strengthened this notion in the MS:

"If we assume that the imaged AZs were not oriented perfectly parallel to the imaging plane, but at an angle, then measuring the cluster sizes using 2D-STED (only lateral: xy) super-resolution without axial (z) super-resolution would lead to an underestimation of the true cluster dimensions. We therefore turned to 3D-STED imaging³³ to complement our 2D-STED data. While sacrificing some lateral resolution for increased axial resolution, we could obtain a near-isotropic point spread function (PSF). We acquired stacks of images with 40 nm pixel-size in the x-, y-, and z-axis (Figure 2a'-a''') and, to reduce bleaching, lowered the power of the STED laser, then achieving a 3D-resolution of ~160 nm (<20% the focal volume of a confocal PSF). This way, we could no longer reliably differentiate the cluster types, but using these stacks we estimated the true length of the long axis of linear clusters by measuring the full width at half maximum of the cluster signal in a line profile of fluorescence intensity in 3D (Figure 2b), which amounted to 461 ± 18 nm (median: 432 nm) and 450 ± 13 nm (median: 462 nm) for Ca_v1.3 and bassoon clusters, respectively (Figure 2c; n = 33 AZs; measurements of clusters tagged with both Abberior STAR635p and STAR580 were pooled, since due to the lower STED efficiency the resolution was not notably different). This indicates that the apparent length measured by 2D-STED microscopy underestimated the true length by approximately

45% for both Ca²⁺-channel and bassoon clusters. This was most likely due to the inclusion of clusters that were tilted relative to the imaging plane and thus appeared shorter in the 2D-STED data, since performing the 2D-analysis on z-projections of the 3D-data resulted in average apparent lengths of 308 ± 11 nm and 286 ± 10 nm for Ca_v1.3 and bassoon clusters, respectively, again shorter than the true lengths measured in the 3D-STED data (see above)."

- I. 142: Why does some antibodies give higher resolution than others? Is it about the antibody (as stated) or the dye (as given in parenthesis)?

We have clarified that the difference in resolution is caused by the dye used for tagging the antibody:

"For the most faithful 2D-STED measurements of the cluster, we used a secondary antibody tagged with the dye providing the highest resolution (Abberior STAR 635P) for either Ca_v1.3 channels or bassoon."

- I. 184: The Ca imaging is described in detail but a short description of the electrophysiology/stimulation is missing before mentioning a "depolarization-evoked fluorescence".

We have added the following to the MS to clarify:

"We applied EGTA in a ramp-like manner after $\Delta F/F_0$ at the AZ had reached a steady state during a 150 ms long depolarization of the IHC to -14 mV (Figure 3b,c)."

- I. 194: the pA/a.u. unit is irrelevant and useless. Furthermore $\Delta F/F$ is unitless and not an arbitrary unit. Same remark for labels of scale bars in Fig. 3b,c.

We have corrected this in the manuscript.

- I.291: The argument in the sentence is unclear to me: "By searching at the basal pole of the IHCs we favored synapses that seemed to lay en face to the image plane"

We have reworded the MS as follows:

"The first task after patching an IHC was then to locate AZs, typically near the basal pole of the IHCs, where most of the membrane, hence the AZs, ran perpendicular to the optical axis."

- The stripe-like arrangement of channels at synapses was already observed with STORM imaging for instance and is not something so surprising. A citation to Dani et al. Neuron Neurotechniques 68 , 843–856, (2010) may here make sense at least from the instrumental point of view. At lines 294-297, it is thus valuable to observe this structure under live-conditions but the conclusion the authors draw is weak. Moreover, the reference to a possible "artifact in immunocytochemistry" should be made explicit if the authors wish to keep this conclusion. Finally, a single example of STED image of OGB is given in Fig. 5a' in which the elongated structure is not so clear as in Fig. 1 and 2: it would be appreciated by the reader to have other illustrations in a supplementary figures.

We thank the reviewer for pointing us to this publication, which is now cited in the MS.

We expand on the possible fixation artifact in the $BSN^{r/-}$ immunostainings as follows:

“STED imaging did not reveal any further details within the large Ca^{2+} -domains, leading us to suspect that the small spot-like appearance of $Ca_v1.3$ immunofluorescence²⁴ was not physiological but artificial, e.g. resulting from aggregation of Ca^{2+} -channels during precipitation by the fixative. We furthermore suspect that the incapability of the previous study to visualize the size differences of the presynaptic Ca^{2+} -hotspots in both genotypes can be explained by the imaging conditions of that study, including a larger confocal PSF and higher $[Ca^{2+}]_e$ than [buffer]). We further addressed this question in additional experiments using conditions that enhanced Ca^{2+} -influx (10 mM $[Ca^{2+}]_e$) and also involved higher OGB-5N (300 μ M) and EGTA (10 mM; “intensified conditions”; Figure 5f). We still observed the above described wide-spread Ca^{2+} -signals in bassoon-deficient IHCs in confocal and STED Ca^{2+} -imaging. Interestingly, we found larger Ca^{2+} -hotspots for wildtype synapses in the “intensified conditions”, where [OGB-5N] was much greater than the expected presynaptic $[Ca^{2+}]$, than described above for more physiological conditions, where [OGB-5N] was more similar to $[Ca^{2+}]$ at the synapse.”

We have now added further examples of depolarization-evoked hotspots of OGB-5N fluorescence increase at STED resolution to Figure 5.

- l. 299 - 301. The authors deduce the intrinsic size of structures from the sted-power independence of the imaged structures. We feel that this is not a foregone conclusion since other imaging artifacts, especially regarding the dye photo-dynamic, could result in the same power-independent behaviour.

We have further expanded on the MS to show that the sizes of the Ca^{2+} -hotspots we measured here are compatible with the mathematical modeling we have done, which is based solely on the Ca^{2+} -binding/-unbinding properties and the diffusion of Ca^{2+} and its fluorescent and non-fluorescent buffers in the cell (as well as on the geometry of the synaptic active zone, as described earlier).

Furthermore, after discussing this specific problem with numerous experts within the STED field, we are not aware of imaging artifacts which could cause a dye to behave in a STED-power dependent manner until a certain threshold, upon which it deviates from the square-root depletion law. In our specific case, we observed in calibration measurements within a homogenous fluorescence “lake” of OGB-5N that the fluorescence intensity continued to drop with ever increasing STED power, up until the maximum available STED power was utilized (Fig. S8e). The de-excitation of the fluorophores by the STED beam is a simple statistical process, and the increased STED beam intensity just increases the probability of a given fluorophore to be switched off. Moreover, as we discussed earlier (and have since added to the supplements), diffusion is too slow to noticeably affect the STED PSF during an individual excitation/detection cycle of 25 ns. The only possible explanation we could think of, where increasing the STED power does not result in an improved spatial resolution (besides the obvious explanation that the observed feature is of the same size or larger than the spatial resolution of the STED nanoscope), would be that the increasing STED power would diminish the remaining fluorescence signal so far, that the signal-to-noise ratio becomes so unfavorable, that any increase in spatial resolution is lost in the increased noise. This could conceivably be possible, if for example tissue-induced aberrations have degraded the STED donut minimum so far, that the fluorescence in

the donut center is switched off, as well. Given the high number of recordings taken at high STED powers and the signal-to-noise ratios observed, we can however rule out this explanation in most cases.

Therefore, we believe that our original conclusion is the most probable explanation, and have thus maintained this in the manuscript.

- I. 306: "re-excitation by the STED beam" is jargon and should be rephrased and explained.

We have removed this from the MS.

- I. 325: The authors dismiss earlier results in Ref 23. This needs more explanation especially since the senior author is the same in the present manuscript as in Ref 23.

We suspect that the inability of the previously published experiments to detect a difference in size of the depolarization-evoked Ca^{2+} -hotspots between wildtype and bassoon-deficient AZs resulted from the imaging conditions used there. Indeed, we found that mathematical modeling of the conditions used in the previous study and convolution of the data with an appropriately-sized PSF resulted in a distribution of Ca^{2+} -bound dye with a FWHM of 830 x 870 nm, even from "wildtype" Ca^{2+} -channel clusters of 67 nm x 430 nm size:

Figure 2: Mathematical modeling of the imaging conditions from Frank et al. 2010.

A mathematical model of the AZ, set up as in the MS but with 400 μM of Fluo-5N, 2 mM EGTA and Ca^{2+} -currents reflecting 5 mM $[\text{Ca}^{2+}]_e$ resulted in a distribution of Ca^{2+} -bound dye that could be fitted with a 2D-Gaussian function with FWHMs of 830 nm x 880 nm. Scale bar: 500 nm.

We have now added the following sentences to the MS:

“We furthermore suspect that the incapability of the previous study to visualize the size differences of the presynaptic Ca^{2+} -hotspots in both genotypes can be explained by the imaging conditions of that study, including a larger confocal PSF and higher $[\text{Ca}^{2+}]_e$ than [buffer]). We further addressed this question in additional experiments using conditions that enhanced Ca^{2+} -influx (10 mM $[\text{Ca}^{2+}]_e$) and also involved higher OGB-5N (300 μM) and EGTA (10 mM; “intensified conditions”; Figure 5f). We still observed the above described wide-spread Ca^{2+} -signals in bassoon-deficient IHCs in confocal and STED Ca^{2+} -imaging. Interestingly, we found larger Ca^{2+} -hotspots for wildtype synapses in the

“intensified conditions”, where [OGB-5N] was much greater than the expected presynaptic $[Ca^{2+}]$, than described above for more physiological conditions, where [OGB-5N] was more similar to $[Ca^{2+}]$ at the synapse.”

-l. 349: What is a "highly monoexponential decay"?

As described above, the fluorescence decay of the free OGB-5N cannot be fully fitted with a monoexponential decay, hence the derived lifetime is only an approximation. Nevertheless, our method of biexponential fitting proved to be highly effective in estimating the $[Ca^{2+}]$, as could be shown in calibration measurements. This was probably because, despite the imperfections in fitting the fast lifetime, the quality of the fit to the long-lifetime component was high enough and the difference in lifetime between the Ca^{2+} -bound and -free OGB-5N (of more than one order of magnitude) was large enough, so that both states could be readily discriminated. In the manuscript we have changed this statement to:

“The fluorescence decay of both the Ca^{2+} -bound and -free state is very well described by a monoexponential function, allowing us to assume a true two-state system (Figure S5a–e, see also Supplementary Methods).”

-l. 514: It is not clear why do the pipet solutions contain Calcium in addition to the Calcium Chelator. Wouldn't it be more efficient omitting the Ca?

Here, 1 mM Ca^{2+} was added to the solution in order to reach a basal level of Ca^{2+} which is closer to the concentration naturally occurring in IHCs without sacrificing too much of the Ca^{2+} -buffering capacity of the 10 mM EGTA. With this combination, we expect about 12 nM of free intracellular $[Ca^{2+}]$.

- In Fig. 5b-c, the presented cluster sizes seem to significantly differ from the results presented in Fig. 1 and Fig. 2. We understand that morphological images in Fig. 1 and 2 were carried with a better dye but the discrepancy in the results should at least be pointed out and discussed in the manuscript.

This is correct, but the features measured in Figs. 1&2 and 5, respectively, are not the same. In Figs. 1&2 we measured the actual antibody-labeled Ca^{2+} -channels as well as the presynaptic scaffolding protein bassoon. In Fig. 5 we imaged (indirectly) the distribution of Ca^{2+} -ions within the cell at the synaptic AZ, more specifically the Ca^{2+} -sensitive dye OGB-5N. The distribution of the Ca^{2+} -bound dye signal within the cell will, of course, be heavily influenced by the lateral diffusion of both the Ca^{2+} -ions and the Ca^{2+} -indicator (as also shown in our simulations) as well as by the sensitivity of the dye. We therefore do not expect the same sizes, but rather that the Ca^{2+} -distributions are larger than the Ca^{2+} -channel (and the bassoon-) distribution. We have expanded the manuscript by adding the following statement:

“The dimensions of the observed hotspots matched well with reaction/diffusion simulations of presynaptic Ca^{2+} -signals we performed using the CalC modeling software⁴⁶, based on the number and distribution of channels as measured before (430 x 67 nm cluster of 120 Ca^{2+} -channels, Figure 6g–i, S14, Table 1), which resulted in a presynaptic Ca^{2+} -domain with a FWHM of 190 x 343 nm (as

estimated by fitting of a 2D Gaussian function to the distribution of Ca²⁺-bound OGB5-N), considerably larger than the size of the Ca²⁺-channel cluster due to lateral diffusion of dye and Ca²⁺-ions (Figure S13). These functional observations therefore corroborate the stripe-like arrangement of presynaptic Cav1.3 immunofluorescence at wildtype AZs (Figure 1a–e).”

- Fluorescence lifetime measurements with STED were performed after 300ps to wait for depletion by the STED beam while the fluorescence lifetime of the free OGB is only 230ps. This means that data fittings are performed on a very small fraction of fluorophores. This fraction is not clearly given and in my opinion, the quick component of the double exponential fitting in Fig. S5e does not work. Is this remark somehow related to the "additional correction factor" mentioned at line 627? These latter sentences were unclear to me.

This an important concern and we have undertaken substantial efforts to further test the validity of our analysis and to clarify our methodology in the manuscript. First, to clarify: the “additional correction factor” (dubbed the “STED efficiency correction”) mentioned here is necessary to correct for the different STED efficiencies of the Ca²⁺-bound and -free states of OGB-5N and is described in detail in Fig. S8 and in the supplemental methods. It and is unrelated to the “blinking” of the first 300ps during the fitting procedure, which requires no additional correction to be applied. We have clarified this both in the manuscript and in the supplements.

Regarding both the “blinking” method and the fitting of the shorter lifetime component in general: Even though the time course of stimulated fluorophore depletion is not much faster than the fast lifetime of OGB-5N, this component can still be estimated with good reliability. For one, the effect of the STED beam ends abruptly with the cessation of STED pulse, whereas the fluorescence decay of the short-lifetime component continues even past the fluorescence lifetime (albeit with diminished amplitude). Additionally, even though the excitation of the fluorophores is fast, it is not instantaneous. This convolution of the fluorescence decay with the instrument response function leads to a “slowing effect”: when convolving an exponential of 230 ps lifetime with the instrument response function of our microscope, we found that still about 63% of the peak fluorescence were left outside the blanking window. Furthermore, due to the much higher brightness of the Ca²⁺-bound OGB-5N (30-fold), an increase in Ca²⁺-binding is much more noticeable in the long-lifetime component, which is much easier to fit and almost unaffected by blanking the first 300 ps of the recorded signal.

Most importantly, we were able to directly test the impact of blanking on the estimation of the fast component of fluorescence decay by applying it to confocal lifetime measurements, where the analysis of the fast component is not complicated by the STED-associated fluorescence decline (Fig. S6). We examined both in situ calibration data, as well as experimental data from IHC recordings, considering measurements with both varying brightness and [Ca²⁺] levels. In essence, the exponential fits using the “blanked” data range were highly similar to the fits using the full data range. No systematic bias was detectable, and the deviations between the “blanked” and the “full-range” fits were minor: expressed in photons, only 0.4–2.8% of the photons were assigned to a different channel (i.e. slow instead of fast, or vice versa); for experimental data, the average (absolute) Δ[Ca²⁺] was between 0.4–1 μM, or between 2–3% of the maximum [Ca²⁺] measured in that specific experiment (Figure S6). This systematic analysis is detailed in Supplemental Figures S5 and S6, as well as in the supplemental methods. Furthermore, we now state on page 18:

“Finally, the STED beam itself impacts the measured fluorescence lifetime (Figure S5, S6): by its very nature, STED causes an additional fast decay of fluorescence ($\tau = 0.19$ ns, Figure 6b, S6a, S8a), which needs to be considered to avoid overestimating the free (fast) dye contribution during STED measurements. When imaging with STED, we therefore excluded the data recorded in the first 300 ps (during the STED laser pulse) from the fitting routine. To ensure that this “blinking” of the first 300 ps did not significantly alter the results of the fitting routine, we fitted fluorescence decay traces recorded in confocal mode using both the full data range and the blanked data (Figure S6). Indeed, both methods produced highly similar results: the differences in the number of photons assigned to the fast and slow lifetime channel were as small as 0.4%–2.8% for $[Ca^{2+}]$ between 10 nM – 40 μ M.”

- Fig. S7, the inefficient STED effect on free OGB is solely attributed to the fluorescence life-time. What about the stimulated emission cross-section?

OGB-5N has (as far as we can tell) identical spectroscopic properties in both the Ca^{2+} -bound and -unbound states. What does change, however, is the fluorescence lifetime due to quenching (by a factor of ~ 14) and the extinction coefficient of the excitation (~ 2 -fold). Importantly, the fluorescence lifetime in the unbound state drops to less than the STED pulse duration, meaning that many excited fluorophores have already spontaneously emitted before the last of the STED photons have even arrived. Due to the exponential decay of fluorescence, this is presumed to be the main contributing factor, although we cannot completely rule out other minor contributors. Luckily, the different STED efficiencies on the Ca^{2+} -bound & -unbound dye (thus allowing us to deduce a correction factor) are accurately determinable, regardless of the underlying effect.

- In the discussion, the limit of the performance of STED calcium imaging in the light of optical effects such as resolution and the influence of the binding states of the OGB (which is nicely discussed). However, the effect of diffusion of ions (extremely fast at the scale of a STED PSF or even a confocal PSF) should be discussed. A detailed and quantitative discussion and comparison of diffusion to imaging speed is missing.

We have carefully addressed this point of the reviewer. Generally, we note that, indeed, one expects the size of the Ca^{2+} -domains to exceed that of the Ca^{2+} -channel clusters as the free and dye-bound Ca^{2+} -ions spread from the cluster. This is treated in the model (Figure 6g-i) and the corresponding results section (as stated above):

“The dimensions of the observed hotspots matched well with reaction/diffusion simulations of presynaptic Ca^{2+} -signals we performed using the CalC modeling software⁴⁶, based on the number and distribution of channels as measured before (430 x 67 nm cluster of 120 Ca^{2+} -channels, Figure 6g–i, S14, Table 1), which resulted in a presynaptic Ca^{2+} -domain with a FWHM of 190 x 343 nm (as estimated by fitting of a 2D Gaussian function to the distribution of Ca^{2+} -bound OGB5-N), considerably larger than the size of the Ca^{2+} -channel cluster due to lateral diffusion of dye and Ca^{2+} -ions (Figure S13).”

In response to the reviewer’s comment we further added the following considerations on how diffusion affects the Ca^{2+} -imaging. There are two relevant time scales when considering what influence diffusion of the Ca^{2+} -dye or -ions might have on our measurements: (i) the larger scale

diffusion of ions and dye over the course of an image, i.e. over (tens of) milliseconds, or until a steady state sets in, or (ii) the possible movement of the dye during one excitation/detection-cycle (~25ns), which could conceivably cause a “blurring of the PSF” as excited fluorophores could diffuse out of the effective focal area. The first should impact the distribution of ions and Ca²⁺-bound dye away from the synaptic active zone, thus determining the size of the observed Ca²⁺-domain. This diffusion should cause no distortions of the image, as the size of the hotspots reached a steady state within <1 ms (as shown by line scans of OGB-5N fluorescence and modeling, see Figure S13), well faster than the ~80ms duration of the depolarization). The second time range is determined by the temporal separation of the individual laser pulses (25 ns), as well as the fluorescence lifetime of the dye. In this brief time window, a “blurring of the PSF” could occur, if excited dye molecules from inside the (effective) focal area diffuse outside, causing the PSF to “smear out”. Considering the limited time frame (max. 25 ns) and the limited diffusion speed (diffusion constant of ~0.1 μm²/ms), dye molecules or ions should not be able to travel more than 2.5 nm away from their previous location. Given an effective focal area (during STED measurements) between 50nM—100nM, this effect should be negligible.

Both these effects have little influence on our data. In our measurements, we do not attempt to measure the fluorescence of a fixed population of dye molecules, as would be the case for imaging morphological structures, but are rather examining a steady state of Ca²⁺-influx, dye binding & unbinding, as well as binding to non-fluorescent buffers.

We have added Supplementary Figure S13 and the following sentence to the supplement:

“Diffusional spread of Ca²⁺-bound OGB-5N from the AZ

There are two relevant time scales when considering what influence diffusion of the Ca²⁺-dye or – ions might have on our measurements. First, the diffusion of ions and dye away from the AZ over the course of an image, i.e. over (tens of) milliseconds or until a steady state sets in, and second, the possible movement of the dye during one excitation/detection-cycle (as determined by the temporal separation of the individual laser pulses of ~25ns), which could conceivably cause a “blurring of the PSF” as excited dye molecules from inside the (effective) focal volume of the STED nanoscope diffuse outside before spontaneously fluorescing.

We performed line-scan recordings of the depolarization-evoked fluorescence increase at IHC AZs in order to measure the speed of the lateral increase of fluorescence away from the AZ. Even though we measured at scan rates of 1.5 kHz, we still could only detect a smaller-sized presynaptic Ca²⁺-domain during the first line of the scan (< 1ms, see Fig. S13a,b). This finding is in line with modeling data indicating that the lateral spread of Ca²⁺-bound OGB-5N reaches a steady state after as little as 400 μs (Fig. S13c). Since this steady state is reached early on during the depolarization, we can assume the Ca²⁺-domain to be stable during our image acquisition, such that lateral diffusion of the dye does not impair the imaging conditions.

Additionally, diffusional movement of individual excited dye molecules out of the effective focal volume of the STED nanoscope is not expected to have any noticeable impact on the spatial resolution, given the limited time frame. Within the 25 ns elapsing between two consecutive laser pulses, a single dye molecule is expected to move on average as little as 2.3 nm (calculated from $l = \sqrt{t \cdot 2D}$, with l the average diffusion length, t the time available for diffusion [here 25 ns], and D

the diffusion coefficient [here $0.1 \mu\text{m}^2/\text{ms}$]. Given an effective focal area (during STED measurements) between 50 nm–100 nm, this effect should be entirely negligible.”

From the optical techniques point of view, I recommend the manuscript for publication after addressing these remarks.

Thank you for supporting this MS.

Reviewers' comments:

Reviewer #1 (Remarks to the Author):

The revised version of the ms by Neef et al., is improved and has adequately addressed most of the points I raised in my review. Although the authors have partially addressed point 14, by making small changes to the introduction, returning to these aspects in the discussion would strengthen the paper and make it of broader appeal. Specifically, as mentioned in my previous review I would like to see the current results set in the context of previous work in .."Holderith et al 2012, Nakamura et al., 2015 and Keller et al., 2015 a more focused discussion on the parallels between findings at non-ribbon and ribbon synapses (e.g. clustering of Ca channels with highly variable numbers, nanodomain coupling (10-20 nm), predicted [Ca²⁺] in the tens of micromolar and vesicles arranged at the perimeter of the channel cluster) in the context of the results presented in the paper would significantly strengthen the paper." In addition, since the paper was last reviewed a highly relevant paper examining the relationship between Ca channel clusters and release sites was published (Miki et al PNAS 2017). The authors may want to also include this if there is space.

Lastly, the revised manuscript has some errors. These include

P24 4.000 should be 4000

Suppl. Fig. S1 legend labels a), b), b), c) should be a), b),c),d).

Reviewer #2 (Remarks to the Author):

In this revised version of their manuscript the authors addressed some of my concerns. The study was also improved regarding several technical aspects by addressing the other reviewers' comments. However, the main problems, expressed in my previous review, remain unsolved: the manuscript fails in successfully combining its two main parts: "physiology of AZ calcium dynamics" and "establishment of STED Ca imaging" into a manuscript that describes a substantial new finding. The physiological part conveys not enough new, ground-breaking information. The second, technical part remains vague, difficult to digest and below the standards established in the field.

For STED-based lifetime imaging a large and complex array of experiments comparing confocal and STED imaging, simulations and calibrations is presented. However, lifetime curves are not displayed as they are subjected to the final determination of intracellular Ca levels. Diverse correction factors, blanking times, etc. are introduced without providing the reader with a clear-cut description of the basic requirements for STED-based lifetime imaging. Why had two gated channels been used for recording the lifetimes? What is the rationale for showing traces from "reverse-pulse experiments" (fig. S4)? Why are the traces of the apo-forms of dyes faster in the STED mode than in confocal mode, while the Ca-bound forms are not affected (fig. S4c)? Why is there no illustration of actual recordings in which the STED artefact was blanked? How do the traces shown in S8c,d compare if the STED artefacts are blanked? How is the fact that the PSFs of apo and Ca-bound dyes differ reflected in the numerical simulations of Ca dynamics?

I appreciate the reply to my previous comments on the technical details of STED imaging. Nonetheless, the "mixed-bag" description of STED-based Ca imaging is still confusing and not to the point. STED-based Ca imaging is an important part of this study, both in the mere coverage of figures and text as well as regarding the data on AZ Ca dynamics. This study is, to my knowledge, the first on STED-based fluorescence lifetime calcium imaging. Consequently, the methods need to be described adequately.

Other issues were more or less solved, I list them for completeness:

Previous point 1), types of AZ as determined by 2D STED.

The manuscript now partly addresses the difficulty in sorting AZ according to their 2D appearance

into distinct classes. Nonetheless, the authors still discuss “different types of AZs” (e.g., legend to fig. 1i). These are, however, only “apparent types”, most likely created by imaging AZ at various angles. The text, figures and legends should clearly say so.

Regarding the length estimates it remained unclear, how well the Bassoon and Cav clusters are described by 2D Gaussian functions. Illustrating the fit for the CaV cluster shown in 1b would clarify this issue. Similarly, it should be illustrated how line profiles were generated for the 3D images. The Bassoon cluster shown in fig. 2a''' could serve as a good candidate. In general, splitting the Bassoon and CaV channels would help inspecting the clusters.

Previous point 2) Differences in estimates of Ca channels per AZ.

The authors addressed this discrepancy now in the text. Accordingly, they should clearly state in the text (not buried in the methods) that the different experiments were done on different age groups. They should also provide the reader with the median and interquartile values shown in fig. 1 of the response letter.

Previous point 8): While p values had been included for statistical tests, power values are still missing.

Previous point 9): I still think that fig. 7 (a scheme based on published work, not on data from the present study) should be removed or exchanged for an illustration of the core messages / the core data of this study.

Previous point 5): More experiments were included, response is accepted.

Previous points 6) and 7): Solved

Additional points, which came up while reading the revised manuscript.

a) It remained unclear why fig. 3d includes so many data points with $d(dF/F_0)$ values close to zero. The text states that only recordings with an EGTA-induced decrease in fluorescence were included. Why had traces been included that show an increase or a minimal decrease only? The same applies for fig. S1c,d.

b) In all plots the axis should include zero values (e.g., figs. 1j, 2c, 3b,c).

c) The grids in fig. 2a are barely visible.

d) Legend to fig. 2c: “length of the line profiles” instead of “width”.

e) Are the data shown in fig. 4c from the recording shown in 4b?

f) In fig. 5c,d,g please clarify the STED power that was used.

g) In fig. 5 add the estimated 2D Gaussians to the images.

h) The numbering of the supplementary figure does not follow their mentioning in the main text.

Reviewer #3 (Remarks to the Author):

The authors have adequately addressed all of my concerns.

Reviewer #4 (Remarks to the Author):

Carefully reading the manuscript and the authors' responses, all my comments have been satisfactorily addressed.

I recommend publication of the manuscript

**Reviewer #1 (Remarks to the Author):**

The revised version of the ms by Neef et al., is improved and has adequately addressed most of the points I raised in my review. Although the authors have partially addressed point 14, by making small changes to the introduction, returning to these aspects in the discussion would strengthen the paper and make it of broader appeal. Specifically, as mentioned in my previous review I would like to see the current results set in the context of previous work in .."Holderith et al 2012, Nakamura et al., 2015 and Keller et al., 2015 a more focused discussion on the parallels between findings at non-ribbon and ribbon synapses (e.g. clustering of Ca channels with highly variable numbers, nanodomain coupling (10-20 nm), predicted [Ca²⁺] in the tens of micromolar and vesicles arranged at the perimeter of the channel cluster) in the context of the results presented in the paper would significantly strengthen the paper." In addition, since the paper was last reviewed a highly relevant paper examining the relationship between Ca channel clusters and release sites was published (Miki et al PNAS 2017). The authors may want to also include this if there is space.

We thank the reviewer for the positive feedback on our manuscript. We have now enhanced the presentation of the relevant literature and included the interesting new paper by Miki et al., 2017 (highlighted in yellow for the sake of this response below).

We have revised the introduction as follows:

"However, understanding synaptic transmission requires information on the number, biophysical properties, and topography relative to vesicular release sites of the presynaptic Ca²⁺-channels that determine the AZ Ca²⁺-signaling. Indeed, localization and quantification of presynaptic Ca²⁺-channels by electron microscopy^{10,13,14} or cell-attached recordings^{15,16} has recently fueled the progress."

and the discussion as follows:

"IHC ribbon-type AZs achieve the high average density of 4000 Ca²⁺-channels per μm² (~120 channels on 0.03 μm²) – considerably higher than typically found in CNS neurons where it ranges from 500-1500^{10,13,14 10,13,14} – likely through a protein network organized by bassoon^{27,59,60} and including the synaptic ribbon^{27,61}, RIM⁶²⁻⁶⁵, and RIM-binding protein^{66,67}. Given this very high density it seems likely that the Ca²⁺-influx at IHC AZs serves additional roles besides the triggering of synaptic release, such as increasing [Ca²⁺] to facilitate vesicle resupply⁶⁸. The distance between Ca²⁺-channels and synaptic vesicles appears to be comparable between IHC AZs and some CNS AZs⁶⁹⁻⁷², which in some cases employ a placement of synaptic vesicles around the perimeter of Ca²⁺-channel clusters¹⁰. Thus, it seems possible that similar mechanisms are used in these presynaptic preparations, even though IHCs make use of a different set of proteins for the release of synaptic vesicles⁷³. Our Ca²⁺-imaging data illustrates

MAX-PLANCK-GESELLSCHAFT

how disruption of bassoon disintegrates this sophisticated supramolecular machinery at the AZ in a manner reminiscent of what was seen at drosophila neuromuscular junctions upon mutation of the scaffold *bruchpilot*^{74,75}. In addition, our study validated the previously assumed presence of extrasynaptic channels in IHCs^{9,25,29}, which form a considerable fraction of 20%–50%. A subdivision into smaller clusters has been described in CNS AZs, where a 1:1 relationship between the number of Ca²⁺-channel clusters and the number of vesicular release sites was reported¹⁴. Future analysis, e.g. employing MINFLUX microscopy⁷⁶ or electron microscopy of immunolabeled SDS-replica¹⁰, will be required to test such a scenario for IHCs. Interestingly, despite the different complement of Ca²⁺-channels per AZ, the number of individual channels per release site, interestingly, appears comparable: 9 channels/site reported by ref. 14 in parallel fiber synapses vs. 80-120 channels for 10-15 release sites⁷⁷ at IHC AZs.”

Lastly, the revised manuscript has some errors. These include P24 4.000 should be 4000

This has been changed.

Suppl. Fig. S1 legend labels a), b), b), c) should be a), b), c), d).

We fixed this mistake.

MAX-PLANCK-GESELLSCHAFT

Reviewer #2 (Remarks to the Author):

We thank the reviewer for the feedback on our manuscript. We have itemized the comments for the sake of our response.

In this revised version of their manuscript the authors addressed some of my concerns. The study was also improved regarding several technical aspects by addressing the other reviewers' comments. However, the main problems, expressed in my previous review, remain unsolved: the manuscript fails in successfully combining its two main parts: "physiology of AZ calcium dynamics" and "establishment of STED Ca imaging" into a manuscript that describes a substantial new finding. The physiological part conveys not enough new, ground-breaking information. The second, technical part remains vague, difficult to digest and below the standards established in the field.

We respectfully disagree. We provide substantially new information on the organization and function of inner hair cell synapses using innovative techniques that have been developed for this project. The manuscript reports on the novel results of this quantitative study as well as on the newly developed techniques, which were necessary to achieve the results. Both techniques and results are validated by the mathematical modelling that was based on the estimates e.g. of Ca^{2+} -channel number and channel cluster size of the present study. In response to the reviewers' comment, to better communicate this point we enhanced the introduction as follows:

"Here, we made use of the good experimental accessibility individual AZs in IHCs to analyze the number, distribution, and activity of Ca^{2+} -channels at individual synapses using innovative methods. We combined confocal Ca^{2+} -imaging with selective inhibition of Ca^{2+} -influx at individual AZs and optical fluctuation analysis for estimating the number of Ca^{2+} -channels per AZ. Employing 3D-STED we quantified the structure of AZ Ca^{2+} -channel clusters in IHCs. We established 2D-STED Ca^{2+} -imaging and 2D-STED fluorescence lifetime measurements of AZ [Ca^{2+}] to analyze the Ca^{2+} -signaling at the AZ and validate the method by mathematical modeling based on quantitative structural and functional characterization of presynaptic Ca^{2+} -channels."

Furthermore, we have done a major reorganization of the part of the Results describing the modeling. We have now prepared a new part and figure (new Fig. 5) dedicated exclusively to modeling, where we describe our efforts to examine the feasibility of STED Ca^{2+} -imaging and find the optimal imaging conditions (concentrations of Ca^{2+} -indicator and non-fluorescent Ca^{2+} -buffers). This part makes use of the results from the previous experiments (size of Ca^{2+} -channel cluster and number of Ca^{2+} -channels) to prepare for the later Ca^{2+} -imaging experiments and thus hopefully provides a better link. Additionally, as before, it serves as an important validation of our STED Ca^{2+} -imaging results. The new part includes some data already presented in the previous version of the manuscript (free Ca^{2+} , physiological conditions, effect of spatial

averaging by STED PSF) as well as some data not shown before (effect of increased concentrations of OGB-5N and EGTA) and now reads:

“Modeling Ca^{2+} -influx at IHC AZs

A previous study had estimated the presynaptic $[\text{Ca}^{2+}]$ to reach $\sim 3 \mu\text{M}$ at IHC AZs during depolarization¹⁹, a value that appears surprisingly low considering the large density of Ca^{2+} -channels established here. Most likely this low value was due to spatial averaging by the PSF of the microscope used there and thus not representative of the true $[\text{Ca}^{2+}]$ at the AZ. We postulated that STED superresolution imaging with its greatly reduced PSF size would provide measurements that were closer to the actual $[\text{Ca}^{2+}]$ at the AZ, allowing us to establish what concentrations would occur *in vivo*. First, we screened several Ca^{2+} -indicator dyes for compatibility with STED nanoscopy and found Oregon Green BAPTA-5N (OGB-5N) to be the best suited Ca^{2+} -indicator with low affinity (Supplementary Fig. S5, S6; reported $K_D = 32.5 \mu\text{M}$ ⁴⁶). In order to better understand what $[\text{Ca}^{2+}]$ can be expected at the IHC AZ and to find out whether superresolution imaging might supply us with a better estimate of the local concentration, we performed reaction/diffusion simulations of the IHC AZ using the CalC modeling software⁴⁷. The model was based on the number and distribution of channels experimentally determined above (430 x 67 nm cluster of 120 Ca^{2+} -channels, Table 1, Supplementary Fig. S7) and used OGB-5N as a Ca^{2+} -indicator. Seeking imaging conditions for optimal super-resolution Ca^{2+} -imaging, we generated several model implementations, changing the concentrations of Ca^{2+} -indicator (from 25 μM to 1000 μM) and non-fluorescent intracellular Ca^{2+} -chelators (from 0.8 mM EGTA and 0.4 mM BAPTA, emulating “physiological” Ca^{2+} -buffering in IHCs with both fast and slow binding kinetics²⁰ to 10 mM of EGTA).

The model predicted that in “physiological” conditions, free $[\text{Ca}^{2+}]$ (Ca^{2+} not bound to Ca^{2+} -indicator or non-fluorescent buffers) can reach up to 100 – 150 μM in close proximity (10-20 nm distance) to the channel mouth (Fig. 5a), but quickly drops as the distance to the channel increases, which is in agreement with previous studies^{3,9,20}. Ca^{2+} -bound indicator could be found at greater distances from the channel cluster due to lateral diffusion of indicator and free Ca^{2+} , such that the Ca^{2+} -domains visible in Ca^{2+} -imaging were larger than the Ca^{2+} -channel cluster itself and did not permit resolution of nanodomains near Ca^{2+} -channels. Still, when local $[\text{Ca}^{2+}]$ was calculated from the ratio of bound to unbound Ca^{2+} -indicator in order to simulate Ca^{2+} -imaging, we found concentrations of up to $\sim 45 \mu\text{M}$ (Fig. 5b), giving a fairly close approximation of $[\text{Ca}^{2+}]$ close to the AZ.

Testing different concentrations of Ca^{2+} -indicator, we found that the best approximations of local $[\text{Ca}^{2+}]$ indeed resulted from lower indicator concentrations (Fig. 5b,c) and thus settled on the lowest feasible concentration of 25 μM . Increasing the non-fluorescent Ca^{2+} -buffers did not appreciably decrease the size of the Ca^{2+} -domain (Fig. 5d). Therefore, we decided for the “physiological” imaging conditions which would also more closely describe $[\text{Ca}^{2+}]$ at IHC AZs *in vivo*. To find out whether STED imaging allows accurate measurements of local $[\text{Ca}^{2+}]$ at the AZ we then convolved the simulated dye distributions with a 3D Gaussian function mimicking the PSF of a 2D-STED microscope (64 x 64 x 542 nm FWHM), and calculated $[\text{Ca}^{2+}]$ from these values. The spatial averaging imposed on the measurements by the PSF (especially in the z axis) lead to a marked underestimation of $[\text{Ca}^{2+}]$, with maximum values of $\sim 10 \mu\text{M}$, but the lateral extent of the Ca^{2+} -signal was strongly decreased compared to confocal

measurements, with a FWHM of 184 x 376 nm (Fig. 5e), indicating the potential of STED Ca^{2+} -imaging to better resolve the Ca^{2+} -domains at IHC AZs.“

The new Fig. 5 is as follows:

Figure 5: Theoretical reaction/diffusion model of Ca^{2+} at the IHC AZ

(a) Theoretical model of Ca^{2+} -influx at a 430 nm x 67 nm cluster of 120 Ca^{2+} -channels (blue symbols, width of 10 nm) shows that the local increase in free $[\text{Ca}^{2+}]$ near the channel mouth, before the Ca^{2+} -ions bind to the Ca^{2+} -indicator or the non-fluorescent buffers, can reach values as high as 150 μM . Blue symbols indicate the positions of simulated Ca^{2+} -channels. Inset shows a magnification of the area marked by the green square. Scale bar: 100 nm.

(b) $[\text{Ca}^{2+}]$, as calculated from the simulated distribution of OGB-5N at the synapse (“reported $[\text{Ca}^{2+}]$ ”), reaches peak values of 45 μM . The lateral diffusion of Ca^{2+} -ions and buffers makes it impossible to acquire the $[\text{Ca}^{2+}]$ at the channel mouth and results in an elongated Ca^{2+} -domain. Numbers indicate the $[\text{Ca}^{2+}]$ at the contour lines.

(c) Increasing the simulated [OGB-5N] (here to 400 μM) results in a lower reported peak $[\text{Ca}^{2+}]$ of ~40 μM .

(d) Increasing the simulated [EGTA] (here to 2 mM) results in a reported peak $[\text{Ca}^{2+}]$ of ~44 μM .

(e) Same data as in (b), but additionally convolved with a Gaussian PSF with a FWHM of 64 x 64 x 542 nm. The reported $[\text{Ca}^{2+}]$ after convolution reaches peak values of up to 10 μM , still considerably lower than the actual concentration near the channels.

Still, we would like to note that, in our view, in particular the technical aspects of the STED Ca^{2+} -imaging are in no means vague, but are described in explicit detail. The manuscript entails not only an exhaustive description of the setup, equipment and measurement details, which are necessary to replicate the experiments, but also a comprehensive list of specific considerations and tricky details that need to be addressed in order to be able to obtain meaningful results, as well as the rationale behind these considerations and all of the correction factors and calibrations measurements required. Especially the TCSPC-STED Ca^{2+} -imaging experiments are highly challenging and require much attention to detail, which by its very nature is “difficult to digest”; this can be considered to be inherent to the difficulty of the problem and less to the quality of the manuscript offered, which nonetheless we consider worth the effort.

MAX-PLANCK-GESELLSCHAFT

For STED-based lifetime imaging a large and complex array of experiments comparing confocal and STED imaging, simulations and calibrations is presented. However, lifetime curves are not displayed as they are subjected to the final determination of intracellular Ca levels.

Representative lifetime curves from actual recordings in inner hair cells were already shown in Figure 6d (now Figure 7d), as well as more extensively in Figure S12 where we additionally show the lifetime fits and the blanking window. Of course, the lifetime curves represent but the first step in obtaining the actual intracellular Ca^{2+} -levels, and aside from determining whether there is more or less Ca^{2+} following depolarization, they cannot be interpreted in a quantitative way without the consecutive fitting, calibrations, corrections and calculations. These are the curves as they are subjected to the final determination of intracellular Ca^{2+} -levels. We have now updated Fig. 6d to additionally indicate the blanking window for the STED-data, but would like to avoid further elaboration, as, in our view, all relevant information is provided and space limitations apply.

(d) Fluorescence lifetime traces from single pixels in the same recordings shown in (c), acquired with confocal (top) and STED imaging (bottom). Red traces show the fluorescence data during depolarization of the IHC, black traces at rest. For analysis of the STED imaging traces we employed a blanking range (grey box). Data points inside this range were excluded from the fitting procedure (used to establish the ratio of Ca^{2+} -bound and free dye) to avoid influence of the artificial STED-evoked short lifetime component.

Diverse correction factors, blanking times, etc. are introduced without providing the reader with a clear-cut description of the basic requirements for STED-based lifetime imaging.

Again, we respectfully disagree. We enumerate and explain all the necessary considerations required for calculating $[\text{Ca}^{2+}]$ using STED early on in the manuscript, and then elaborate in more detail further in the manuscript and in even more detail in the supplement. None of the “diverse correction factors [...] etc.” are extraneous in our view, as disregarding any one of them would lead to a misrepresentation of the actual $[\text{Ca}^{2+}]$ values. As such, all of them can be regarded to be “basic requirements” for acquiring quantitative $[\text{Ca}^{2+}]$ data with STED imaging. The correction and calibration factors represent no more than a detailed understanding of the used Ca^{2+} -indicator dye, its buffering properties, its response to STED imaging and the actual experimental setup itself. Furthermore, the equipment and hardware we used for STED-based Ca^{2+} -imaging is listed in detail, allowing immediate replication of the experimental setup.

MAX-PLANCK-GESELLSCHAFT

Why had two gated channels been used for recording the lifetimes?

This appears to be a misunderstanding, since gating was not used for the recording of lifetime traces. In order to make the reference to xy-scanning even more clear, we have changed the methods description to read:

“For the analysis of the spatial extent of the Ca^{2+} -domains in xy-scans, the detector signal was electronically time-gated by diverting the first 450 ps (equaling twice the fluorescence lifetime of unbound OGB-5N) of the signal into a separate detection channel...”

Regarding the criticism of the description of the STED-based lifetime imaging: we respectfully disagree with the notion of reviewer. With the main MS and supplemental material we have come a long way making this MS a comprehensive resource for this exciting new method of STED. As mentioned in the acknowledgement we had already communicated our approach to peers in the field and incorporated their feedback. In response to the reviewers continued concern we have thoroughly considered all aspects and also asked Drs. Silvio Rizzoli and Christian Vogl, two biologists heavily applying STED nanoscopy, to read the MS and provide feedback primarily on the aspect of intelligibility. They have suggested some useful edits for further improved readability that we implemented.

What is the rationale for showing traces from “reverse-pulse experiments” (fig. S4)?

As stated in the main manuscript, “we screened several Ca^{2+} -indicator dyes for compatibility with STED nanoscopy and found Oregon Green BAPTA-5N (OGB-5N) to be the best suited Ca^{2+} -indicator with low affinity (Figure S5, S6; reported $K_D = 32.5 \mu\text{M}^{45}$)”. As stated in the caption, Supplementary Fig. S6 (previously Supplementary Fig S4) examines the “Photostability of fluorescent Ca^{2+} -indicators in STED microscopy”. Further explanation is then given in the figure legend for S6b: “STED switched off the fluorescence of the Ca^{2+} -indicators OGB-1, OGB-5N, Fluo-4, and Fluo-5F but had no or very little effect on the fluorescence in reverse pulse mode when the fluorophores were in the ground state. Fluo-4FF and Fluo-5N, however, showed an increase in fluorescence in reverse pulse mode in low $[\text{Ca}^{2+}]$ which might result from photo-damage of the BAPTA chelator from the dye by the STED light.”

To further clarify the rationale for the reverse-pulse experiments, we have now additionally updated the legend for Supplementary Fig. S6a: “In reverse mode the temporal alignment was changed; the excitation pulse of $\sim 110\text{ps}$ full-width half-maximum was delayed and followed the $\sim 300\text{ps}$ long STED pulse. Here, no change in fluorescence intensity would be expected due to the STED pulse if the only effect of the STED laser on the dye would be stimulated emission depletion. Effects of the STED laser in reverse mode, however, would point to potential photo-damage.”

This shows that Fluo-4FF and Fluo-5N are not suitable for STED microscopy, underlining that such experiments were indeed necessary.

MAX-PLANCK-GESELLSCHAFT

Why are the traces of the apo-forms of dyes faster in the STED mode than in confocal mode, while the Ca-bound forms are not affected (fig. S4c)?

The traces of the apo-forms of Fluo-4FF and Fluo-4N are not faster in reverse-STED mode, but rather slower. As we had stated in the legend “Both indicators showed a change towards longer fluorescent lifetime with STED light in low $[Ca^{2+}]$ (dotted and straight blue line) while there was no change in high $[Ca^{2+}]$ (red). The increase in lifetime of the Ca^{2+} -free dye might reflect photo-damage of the Ca^{2+} -chelator moiety of the dye which normally quenches the fluorescence.”

We would need to speculate as to why the Ca^{2+} -bound form of the dyes is not affected, but we note that if indeed the Ca^{2+} -chelator moiety of the dyes were damaged by the STED laser, this could lead to an increase in lifetime and brightness for the apo-form of the dye, since the chelator quenches the fluorescence when not bound to Ca^{2+} . Therefore, damage to it could lead to a reduction in quenching. In the Ca^{2+} -bound form we would not see this damage because, being unquenched, it will have the same or very similar lifetime as damaged dye. The Ca^{2+} -bound form can be regarded as a reference while the apo form shows that these indicators are not usable for STED.

Why is there no illustration of actual recordings in which the STED artefact was blanked?

The “STED artefact” was not blanked from the recordings themselves but merely from the fitting range. That is, when the biexponential function convolved with the instrument response function was fitted to the dataset, those points falling into the blanking range were not included in the fit. This is an important distinction: we did not physically truncate the incoming data, but instead recorded the entire fluorescence trace. We believe that it is important to show the STED artefact and had already illustrated the blanking range in actual recordings in Supplementary Fig. S12a as a transparent grey box, but as stated above we have now additionally indicated the blanking range in the display of actual recordings in Fig. 6d.

How do the traces shown in S8c,d compare if the STED artefacts are blanked?

In this supplementary figure we show that Ca^{2+} -bound and $-Ca^{2+}$ -free OGB-5N shows very different behavior under STED illumination, namely that Ca^{2+} -bound dye is much more STED-efficient than the apo-form (Ca^{2+} -free) of OGB-5N. For this we used in situ samples of OGB-5N in either Ca^{2+} -free or Ca^{2+} -saturated buffer solution, so we would expect (and in fact observed) that all the Ca^{2+} -indicator dye was either in the apo-form (Ca^{2+} -free solution) or in the Ca^{2+} -bound state (saturated solution). These measurements were the basis of the “STED-efficiency correction factor s ”, which was used in our later calculations of $[Ca^{2+}]$. The fitting algorithm was not necessary here (except for calibration purposes), as the fit itself is used to discriminate the photons emanating from Ca^{2+} -bound/ Ca^{2+} -free dye – but in each measurement the dye is either all Ca^{2+} -bound or Ca^{2+} -free. Instead, we compared the absolute number of detected photons for increasing STED powers from 0 (confocal) to maximal STED powers, and so could determine the

8/18

MAX-PLANCK-GESELLSCHAFT

percentage of dye molecules that had been switched off (and so determine the STED efficiency), which is shown in Supplementary Fig. S11 c,d as the shaded areas.

The reviewer's question is phrased rather ambiguously, hence we are not sure what exactly s/he wants to know – but we will try to address all possible readings of the question.

- (1) How do the determined dye ratios compare, if the STED artifact is blanked?

When using the fits to determine the amount of bound or free dye, we determined >99% of the dye to be Ca^{2+} -free/ Ca^{2+} -bound in Ca^{2+} -desaturated (c)/ Ca^{2+} -saturated (d) solution when examining the confocal recordings. When examining the STED-data, we obtained very similar results when fitting to the blanked data region (>97%—99% Ca^{2+} -free/ Ca^{2+} -bound). If the data was not blanked during the fit, then the results from the STED recordings showed incorrect ratios of only 60% in one state and 40% in the other.

- (2) How do the fluorescence decay traces compare, if the blanked data regions are disregarded?

If we truncate the data by cutting off the blanked region, then the individual data curves (using different STED powers) can be rescaled and superimposed to compare whether the fluorescence decay of the dye was altered by the STED beam. If we do this, then the (rescaled) decay curves look practically identical, indicating that once the STED beam subsides, there is no remaining influence on the dye, which decays naturally from that moment on.

- (3) How does the calculated STED-efficiency change when the blanked data regions are excluded from consideration?

Because the truncated data region is so short compared to the entire observed fluorescence decay trace (<3% of the recorded time), and the fluorescence especially of the Ca^{2+} -bound dye is emitted over a very long time period (>8ns), the impact of ignoring the first 300ps changes the calculated STED-efficiency only by a few percent. The small effect this would have on any final [Ca^{2+}] determinations could be compensated for when allocating the number of fluorescence photons to each channel following the fitting algorithm step, and could thus be safely ignored. We tested for this and decided to use our original approach.

- (4) How do the fluorescence traces of panel (c) compare to panel (d) when the STED-artifact is blanked?

If the first 300ps are discarded, then it is clearly evident that the decay measured in the Ca^{2+} -free solution is incredibly short, whereas the decay in the Ca^{2+} -saturated solution remains much longer. Both decay curves are almost purely monoexponential when disregarding the blanked data region, implying that the indicator dye is either entirely in the bound or the unbound state.

This should cover any possible readings of the question. Regardless, however, we did not alter the Supplementary Fig. S11 (previously S8), as the data presented in this Figure did not need to be fitted to determine the STED efficiency correction factor which relies on the comparison of the detected photons.

MAX-PLANCK-GESELLSCHAFT

How is the fact that the PSFs of apo and Ca-bound dyes differ reflected in the numerical simulations of Ca dynamics?

We have not used differently sized PSFs to convolve the simulation data for Ca²⁺-free and -bound dye. As stated in the last point-by-point response *“We cannot directly measure the exact dimensions of the PSFs of Ca²⁺-bound and -free dye, since we would need a sub-resolution-sized sample of dye for this (e.g. a dye-filled 10 nm bead or droplet), which is hard if not impossible to generate. Moreover, encasing the dye within a bead or attaching it to a surface might well significantly change its properties. The confocal PSFs of the Ca²⁺-bound and -free dye should be practically identical, as the spectral properties of the dye do not change upon Ca²⁺-binding”*.

Therefore, as stated in the methods part, *“The Ca²⁺-concentration derived from the ratio of Ca²⁺-bound and -free OGB-5N was calculated as*

$$[Ca^{2+}] = K_D \cdot \frac{[OGB-5N]_{bound}}{[OGB-5N]_{free}}$$

i.e. we have not applied the STED correction factor to the calculation of [Ca²⁺] from the mathematical model. If we were to make some assumptions, we could see two approaches to estimate the “larger PSF” of the free OGB-5N from the difference in the photon count, then requiring application of a STED correction factor of e.g. 0.491:

i) a PSF of free OGB-5N with an increased size of 91x91x542 nm, such that the integral of this PSF is 1/0.491 times as large as that of the 64x64x542 nm PSF used to convolve the Ca²⁺-bound OGB-5N distribution.

ii) a combination of a 64x64x542 nm PSF, identical with the one used to convolve the Ca²⁺-bound OGB-5N distribution, with a 243x243x542 nm “confocal” PSF such that the integral of the combined PSF is 1/0.491 times as large as that of the 64x64x542 nm PSF used to convolve the Ca²⁺-bound OGB-5N distribution. To reach the 1/0.491 value of the original STED-PSF the combination would need to be 7.7% “confocal” PSF and 92.3% “STED” PSF.

We have calculated the Ca²⁺-concentrations from the modeling results, convolved with PSFs following either of the above approaches and then again corrected using the STED correction factor of 0.491 and, unsurprisingly, since we used the same factor as basis of the PSF change, found only very little (max. 0.13 μM) difference:

PSF: Gaussian with FWHM of 64x64x542 nm (unchanged)

Max $[Ca^{2+}] = 10.37 \mu M$

PSF: Gaussian with FWHM of 91.34x91.34x542 nm

Max $[Ca^{2+}] = 10.34 \mu M$

PSF: 2 Gaussians with FWHMs of 64x64x542 nm, amplitude 0.923, and 243x243x542 nm, amplitude 0.077

Max $[Ca^{2+}] = 10.24 \mu M$

Figure L1: Effect of different changes in convolution to simulate the effect of differently sized PSFs due to different STED efficiencies for Ca^{2+} -bound and free dye

a) Plotted are Ca^{2+} -concentrations calculated from the distribution of dye predicted by the mathematical model of the IHC AZ, convolved with a 64 x 64 x 542 nm Gaussian “PSF” and calculated as $[Ca^{2+}] = K_D \cdot \frac{[OGB-5N]_{bound}}{[OGB-5N]_{free}}$.

b) Ca^{2+} -concentration calculated from the same data as in (a), with the same convolution of the distribution of Ca^{2+} -bound dye. However, the distribution of free dye was now convolved with a 91.34 x 91.34 x 542 nm Gaussian “PSF”, and the Ca^{2+} -concentration was calculated as $[Ca^{2+}] = K_D \cdot \frac{[OGB-5N]_{bound}}{[OGB-5N]_{free} \cdot s}$ with s the STED correction factor of 0.491.

c) Same as in (b), this time the distribution of free dye convolved with a “PSF” consisting of the summation of two Gaussians, one with “STED-dimensions” of 64 x 64 x 542 nm and an amplitude of 0.923, the second with “confocal dimensions” of 243 x 243 x 542 nm and an amplitude of 0.077. $[Ca^{2+}]$ was calculated using the same equation as in (b).

In conclusion, we would like to leave the data presented in the manuscript as it is, since we do not want to further complicate the manuscript with additional assumptions that do not seem to impact the results: the differences in $[Ca^{2+}]$ were so low that they would not be visible in the manuscript when rounding.

I appreciate the reply to my previous comments on the technical details of STED imaging. Nonetheless, the “mixed-bag” description of STED-based Ca imaging is still confusing and not to the point. STED-based Ca imaging is an important part of this study, both in the mere coverage of figures and text as well as regarding the data on AZ Ca dynamics. This study is, to my knowledge, the first on STED-based fluorescence lifetime calcium imaging. Consequently, the methods need to be described adequately.

MAX-PLANCK-GESELLSCHAFT

We had already expanded the description of the methods to cover 6 pages (1699 words) and 1 figure in the results section, 1 page (261 words) in discussion, 10 pages (3617 words) in the Supplementary Methods, and 12 Supplementary Figures. As outlined above and below we have made further changes in response to specific comments of the reviewer as well as those of two additional peers. We hope the current state of our description of STED-based Ca^{2+} -imaging in this revised version of the MS will please the reviewer, as we do not see how we can further improve on this in the absence of specific criticism of what exactly remains vague about our presentation of the methods.

Other issues were more or less solved, I list them for completeness:

■ Previous point 1), types of AZ as determined by 2D STED. The manuscript now partly addresses the difficulty in sorting AZ according to their 2D appearance into distinct classes. Nonetheless, the authors still discuss "different types of AZs" (e.g., legend to fig. 1i). These are, however, only "apparent types", most likely created by imaging AZ at various angles. The text, figures and legends should clearly say so.

We have now changed the manuscript to read:

"One major drawback of a 2D-STED analysis is that the orientation of a given AZ relative to the imaging plane can have a marked effect on its apparent shape and size estimate." Further, during the introduction of the morphological 3D STED imaging and its more limited resolution, we state "This way, we could no longer reliably differentiate the the apparent shapes we had previously defined in 2D-STED images, (...)". Additionally, we changed the legend to figure 1 to read "(i) Summary of the distribution of different types of AZs as defined by the apparent shape of the individual AZs (a-h) measured with 2D-STED microscopy in our sample of n=138 AZs from N=4 mice."

Regarding the length estimates it remained unclear, how well the Bassoon and Cav clusters are described by 2D Gaussian functions. Illustrating the fit for the CaV cluster shown in 1b would clarify this issue.

Including fit profiles into Figure 1 would cover up the clusters shown in the figure, making it almost impossible to discern the shape of the cluster under the fit profile. We have therefore now included a new Supplementary Figure 1 to illustrate the 2D Gaussian fitting of the clusters and refer to it in the Results part:

"We then approximated the apparent size of the clusters by fitting a 2D Gaussian function to the STED images (Supplementary Fig. S1)."

Supplementary Figure S1: Measuring the size of Ca²⁺-channel- and bassoon-clusters using 2D Gaussian fits

The lengths and widths of Ca_v1.3 Ca²⁺-channel- and bassoon-clusters were measured by fitting 2D-STED images of the Ca_v1.3 (green) and bassoon (magenta) immunofluorescence at IHC AZs using 2D Gaussian functions. The resulting fit profiles are indicated by line contours, with green lines indicating the half-maximum. The resulting full-widths-at-half-maxima of the long and short axes are indicated below the fits. Scale bar: 100 nm.

Similarly, it should be illustrated how line profiles were generated for the 3D images. The Bassoon cluster shown in fig. 2a''' could serve as a good candidate. In general, splitting the Bassoon and CaV channels would help inspecting the clusters.

We understand that the Figure 2a''' can be confusing, since here we indeed show two Ca²⁺-channel clusters which are very close to each other. We had originally chosen this example since it shows a set of clusters which would be hard to tell apart in 2D-STED (we would not have been able to resolve them since they are very close in z). We have thus chosen to replace this panel with a more appropriate example. Additionally, we now show the Ca_v1.3 clusters displayed in Fig. 2a''-a'''' in isolation, have added line profile plots for these clusters, and have included the corresponding measures in the figure legend:

“Volumetric displays of representative examples of $Ca_v1.3$ (green) and bassoon (magenta) immunofluorescence at IHC AZs, acquired with 3D-STED microscopy. In the right panels of a'' - a''' , $Ca_v1.3$ immunofluorescence is shown in isolation, blue lines indicate the 3D line profile plots used to estimate the length of the clusters by measuring the full width at half maximum of the line profile, here 455 nm (a''), 389 nm (a'''), and 535 nm (a''''). Grid distance: 500 nm.”

For the reviewer’s information, we additionally show here the panel as it would have looked with the originally used clusters. In this case, the full width at half maximum of the two line profiles would have been 647 nm for the top line profile and 423 nm for the bottom line profile.

Previous point 2) Differences in estimates of Ca channels per AZ. The authors addressed this discrepancy now in the text. Accordingly, they should clearly state in the text (not buried in the methods) that the different experiments were done on different age groups. They should also provide the reader with the median and interquartile values shown in fig. 1 of the response letter.

In response to the comment, we have revisited this point and have now included a dataset on maximum $\Delta F/F_0$ changes from IHC AZs of mice of different age groups, acquired on the same microscope used in the experiments making use of iontophoretic application of EGTA, thereby adding another author to the paper. Notably, this is not identical to the data shown in the last letter to the reviewers, since these data are to be used in a different study. In response to the reviewer’s comment we now show the new data in Supplementary Figure S4. The data indicate a slight decrease in the maximal $\Delta F/F_0$ of IHC AZs in mice of older age and, hence, mention age-dependent differences as potential contributing mechanism. We do, however, still believe that the different number of Ca^{2+} -channels established using the two different methods we used in this project cannot be explained by the age difference of the mice used in either method and say so in the manuscript, now explicitly mentioning the age difference:

“In summary, both experimental approaches reported somewhat different estimates for the mean number of Ca^{2+} -channels per AZ (125 and 78), suggesting either an overestimation of the number of channels after application of EGTA (e.g. due to additional effect on extrasynaptic Ca^{2+} -channels that is difficult if not impossible to quantify from our experiments), or an underestimation in the optical fluctuation analysis experiments (e.g. due to activity of synaptic channels outside the microscope’s PSF), or a combination of both. As the two approaches made use of mice of different age (P15-17 for EGTA application and P21-28 for fluctuation analysis) for which we find subtle differences in the $\Delta F/F_{0, \max}$ of the Ca^{2+} -indicator (Supplementary Fig. S4), we cannot exclude a small contribution of an age-dependence of the AZ Ca^{2+} -channel complement. However, we do not think that this fully accounts for the observed differences.”

We further explain this notion in the legend of the new Supplementary Fig. S4:

“Supplementary Figure S4: Age-dependence of maximal presynaptic Ca^{2+} -increase

Synaptic Ca^{2+} -signals imaged by spinning-disk confocal microscopy as described for the selective suppression of Ca^{2+} -influx at individual AZs by EGTA-iontophoresis (Figure 3). The box plot of the background-normalized maximal Fluo-8FF fluorescence increase ($\Delta F/F_{0, \max}$) in response to ramp-depolarization of IHC active zones from wildtype mice shows a slight difference between age groups P15-17 (red, $\Delta F/F_{0, \max}=1.17 \pm 0.05$, median 1.02) and P21-28 (black, $\Delta F/F_{0, \max}=1.12 \pm 0.09$, median 0.88, $p=0.03$, Mann Whitney U test, statistical power: 0.07).”

Previous point 8): While p values had been included for statistical tests, power values are still missing.

We have now included power values as well.

Previous point 9): I still think that fig. 7 (a scheme based on published work, not on data from the present study) should be removed or exchanged for an illustration of the core messages / the core data of this study.

At the reviewer’s request we have removed Fig. 7 and any reference to it from the manuscript.

Previous point 5): More experiments were included, response is accepted.

Previous points 6) and 7): Solved

Additional points, which came up while reading the revised manuscript.

a) It remained unclear why fig. 3d includes so many data points with $d(dF/F_0)$ values close to zero. The text states that only recordings with an EGTA-induced decrease in fluorescence were included. Why had traces been included that show an increase or a minimal decrease only? The same applies for fig. S1c,d.

As stated in the figure legend, the data in Fig. 3d is the plot of $\Delta(\Delta F/F_0)$ against ΔI_{Ca} from the representative recording shown in Fig. 3c. Therefore, the data points plotted here do not show the maximal $\Delta(\Delta F/F_0)$ of individual cells but rather the development of the increase in $\Delta(\Delta F/F_0)$

and ΔI_{Ca} during the course of one recording, while ramping up the iontophoretic application of EGTA. Here, each data point corresponds to a point in time during the measurement from one individual synapse. Since in the beginning and the end of the recording no EGTA is applied, no change in fluorescence or Ca^{2+} -current is observed at these time points, resulting in data points showing $\Delta(\Delta F/F_0)$ and ΔI_{Ca} close to zero. These data are

then used to establish the relationship of $\Delta(\Delta F/F_0)$ and ΔI_{Ca} , shown in Supplementary Figure S2c,d (previously S1c,d) in more detail. In order to avoid confusion we now exclude data points before and after the iontophoretic application of EGTA from the graph.

b) In all plots the axis should include zero values (e.g., figs. 1j, 2c, 3b,c).

While we do not share the reviewer's view that all axes should include zero values (e.g. the distance to zero is irrelevant when comparing hotspot sizes, since we are looking at differences only; $\Delta F/F_0$ data are background-subtracted, therefore the zero value should be evident from the baseline of the trace), we have updated Figures 1j, 2c, 3b,c, 4b,c, 6c,d,g, (previously 5c,d,g) S10a, and S16b to include zeros on all axes. We did not update Figure S10c as this would have made it much more difficult to read the figure.

c) The grids in fig. 2a are barely visible.

We have improved visibility of the grids.

d) Legend to fig. 2c: "length of the line profiles" instead of "width".

We have changed the legend to read (...) "as measured from the full width at half maximum of the line profiles.", since the length of the line profile simply depends on the manually defined start and end points of the measurement. The data plotted in 2c, while showing the lengths of the clusters, is measured as the full width at half maximum of the line profiles, so we would like to keep the term "width".

MAX-PLANCK-GESELLSCHAFT

e) Are the data shown in fig. 4c from the recording shown in 4b?

Yes. We have updated the figure legend as follows:

“(c) Plot of baseline-subtracted variance of Ca^{2+} -indicator fluorescence (open circles) or of whole cell Ca^{2+} -current (filled diamonds) against mean fluorescence or current, respectively, **from the exemplary cell shown in (b).**”

f) In fig. 5c,d,g please clarify the STED power that was used.

Here, we pooled recordings using different STED powers. We have now updated the figure legend for 6c-d (previously 5c-d) to read:

“Both the short (c) and the long axis (d) of the Gaussian were significantly smaller when measured with STED microscopy (red, n = 51 synapses, **STED power of 12.3 to 35 mW**) compared to confocal microscopy (black, n = 77 synapses)”

and that for 6g (previously 5g) to read

“The size of hotspots of $Bsn^{\Delta Ex4/5}$ mice (as measured by fitting of a 2-dimensional Gaussian function) is unchanged when using STED (red, n = 15 synapses, **STED power of 12.3 to 24.3 mW**) compared to confocal recordings (black, n = 23 synapses)”

g) In fig. 5 add the estimated 2D Gaussians to the images.

We would like to refrain from doing so since, as is apparent in Fig. 6a''' (previously 5a'''), adding the 2D Gaussian fits makes it much harder to see the shape of the synaptic Ca^{2+} -hotspot.

h) The numbering of the supplementary figure does not follow their mentioning in the main text.

We have re-sorted the Supplementary Figures according to their first mention in the main text.

MAX-PLANCK-GESELLSCHAFT

Reviewer #3 (Remarks to the Author):

The authors have adequately addressed all of my concerns.

We thank the reviewer for the positive feedback on our manuscript.

Reviewer #4 (Remarks to the Author):

Carefully reading the manuscript and the authors' responses, all my comments have been satisfactorily addressed.

I recommend publication of the manuscript

We thank the reviewer for the positive feedback on our manuscript.

REVIEWERS' COMMENTS:

Reviewer #1 (Remarks to the Author):

The authors have now adequately addressed my concerns in the revised manuscript.

Reviewer #2 (Remarks to the Author):

In the second revision my technical concerns have been adequately addressed.

My general concerns, however, have not been addressed: i) there is no ground-breaking new biological information and ii) the new methodology is not described in a clear and comprehensible style. This has nothing to do with the complexity of the measuring system but with a poorly structured manuscript (at least regarding the incorporation and description of the STED-based calcium imaging). The paper is ok but, in my view not suitable for publication in a prestigious journal.

MAX-PLANCK-GESELLSCHAFT

Reviewer #1 (Remarks to the Author):

The authors have now adequately addressed my concerns in the revised manuscript.

We thank the reviewer for the positive feedback on our manuscript.

Reviewer #2 (Remarks to the Author):

In the second revision my technical concerns have been adequately addressed.

■ My general concerns, however, have not been addressed: i) there is no ground-breaking new biological information and ii) the new methodology is not described in a clear and comprehensible style. This has nothing to do with the complexity of the measuring system but with a poorly structured manuscript (at least regarding the incorporation and description of the STED-based calcium imaging). The paper is ok but, in my view not suitable for publication in a prestigious journal.

We are glad that we have managed to address the reviewer's technical concerns. As stated before, we respectfully disagree regarding the lack of new biological information. We have however taken additional steps to further clarify the description of STED-based Ca^{2+} -imaging by including details from the Supplementary Methods in the main MS Methods part, streamlining the description, and adding a new Figure 7 which gives a schematic overview of the different correction factors used to calculate $[\text{Ca}^{2+}]$ from the fluorescence lifetime data. We hope that these measures helped to make the STED methods more easily understandable.